# Metabolic fingerprinting on retinal pigment epithelium thickness for individualized risk stratification of type 2 diabetes mellitus

Shaopeng Yang [1,2,3,4], Zhuoting Zhu[5], Shida Chen[1,2,3,4], Yixiong Yuan[1,2,3,4], Mingguang He[1,6] ✉ & Wei Wang [1,2,3,4] ✉

The retina is an important target organ of diabetes mellitus, with increasing evidence from patients and animal models suggesting that retinal pigment epithelium (RPE) may serve as an early marker for diabetes-related damages. However, their longitudinal relationship and the biological underpinnings remain less well understood. Here, we demonstrate that reduced in vivo measurements of RPE thickness (RPET) represents a significant risk factor for future type 2 diabetes mellitus (T2DM) and its microvascular phenotypes. After performing systematic analyses of circulating plasma metabolites using two complementary approaches, we identify a wide range of RPET metabolic fingerprints that are independently associated with reduced RPET. These fingerprints hold their potential to improve predictability and clinical utility for stratifying future T2DM and related microvascular phenotypes beyond traditional clinical indicators, providing insights into the promising role of retinas as a window to systemic health.

Type 2 diabetes mellitus (T2DM) represents a chronic metabolic disorder that poses a significant global health challenge, with an estimated 783.2 million projected to suffer from the disease by 2045[1–3]. Although seminal risk factors and models have been established to improve risk assessments and prevention of T2DM[4–7], their predictive abilities are imperfect, highlighting the necessity for research into novel biomarkers that enhance our understanding of the disease process and bridge the current gap in diabetes stratification[7].

The retina is an important target organ of diabetic damage, which has exceptional susceptibility to diabetes-related metabolic stresses[8–10]. Using state-of-the-art in vivo retinal optical coherence tomography (OCT), changes of retina were captured even before evident diabetes[11–13]. Retinal pigment epithelium (RPE) is the outer layer of the retina and plays an important role in maintaining retinal function by providing 60–80% of the retina's glucose supply through its high-capacity transport system[14]. Evidence from patients and animal models

have suggested RPE structural damages and dysfunction in prediabetes and early stages of diabetes without retinopathy, indicating that RPE may serve as an early marker for diabetes-related damages[15–22]. However, to our knowledge, there are currently no studies evaluating the association between in vivo RPE measurements and the risk of diabetes in cohorts. Therefore, their longitudinal relationship and the biological underpinnings remain unknown.

We hypothesized that RPE alterations precede the onset of diabetes, possibly owing to the exuberant nature of RPE metabolism, which predisposes RPE to subtle metabolic perturbations that may exist well before the manifestation of evident diabetes. Under this hypothesis, RPE-associated metabolic alterations may provide a distinctive glimpse into diabetic metabolic disturbances at a much earlier stage. Herein, we first scrutinized the prospective association of in vivo OCT gauged RPE thickness (RPET) with future risk of developing T2DM in the general population of Europe. Subsequently, we performed a

¹State Key Laboratory of Ophthalmology, Zhongshan Ophthalmic Center, Sun Yat-sen University, Guangzhou, China. ²Guangdong Provincial Key Laboratory of Ophthalmology and Visual Science, Guangzhou, China. ³Guangdong Provincial Clinical Research Center for Ocular Diseases, Guangzhou, China. ⁴Hainan Eye Hospital and Key Laboratory of Ophthalmology, Zhongshan Ophthalmic Center, Sun Yat-sen University, Haikou, China. ⁵Centre for Eye Research Australia, Royal Victorian Eye and Ear Hospital, Melbourne, Australia. ⁶Experimental Ophthalmology, The Hong Kong Polytechnic University, Hong Kong, China. ✉e-mail: mingguang.he@polyu.edu.hk; wangwei@gzzoc.com

systematic analysis of circulating plasma metabolomics to identify RPET-associated metabolites and to explore the potential of these metabolic fingerprints to inform T2DM risk. Finally, a deeper-penetrating OCT device and a more sensitive metabolomic assay were deployed in a southern Chinese diabetic population to further assess the association of RPET with future diabetic microvascular phenotypes, and the role of RPET metabolic fingerprints in facilitating risk stratification for these complications.

## Results

### Baseline characteristics
The workflow of the study design is shown in Fig. 1. A total of 7,824 participants in population-I with 3,913 right eyes and 3,911 left eyes were eligible for phase-I and -II analyses. In addition, 84,224 eligible participants in population-II were included for phase-III analysis, of which baseline diabetes were excluded prior to all analyses. Baseline characteristics of the study population are summarized in Supplementary Table S1. Compared to population-II, participants in population-I were typically younger, male, more educated, had higher income, lower BMI, smoked less, and less likely to be hypertension (all $P < 0.05$). Participants in the training and testing sets shared similar distributions of characteristics (all $P > 0.05$).

### Association between RPET and risk of T2DM
A greater average RPET significantly reduced the risk of future T2DM, both in the overall population-I (hazard ratio [HR] = 0.856, 95% confidence interval [CI]:0.768, 0.954; $P = 0.005$) and in the subgroup with participants in the lower three quartiles of AMD-PRS (HR = 0.835, 95% CI:0.734, 0.950; $P = 0.006$) after adjusting for a wide range of covariates. These associations persisted across both sexes and 8 ETDRS subfields of RPET (Fig. 2, Supplementary Tables S2, S3). However, no association was observed when analyzing participants in the highest quartile of AMD-PRS (HR = 0.900, 95% CI:0.731, 1.107; $P = 0.319$). Similar results were obtained in sensitivity analyses, including those

that (1) adjusted for more comprehensive ethnicity classifications with further subdivisions for South Asian, East Asian, Black, and Mixed races; (2) further adjusted for biological age[23]; (3) further adjusted for frailty score[24]; and (4) excluding cancer, cardiovascular diseases, and renal diseases (Fig. 2, Supplementary Table S4).

### RPET-associated metabolites and risk of T2DM
A total of 64 metabolic biomarkers were independently associated with average RPET after multiple testing correction (false discovery rate [FDR] $P < 0.05$). These included 60 with negative associations, encompassing total lipids, triglycerides, phospholipids, cholesterol, free cholesterol, and cholesteryl esters in very low-density lipoproteins (VLDL) and low-density lipoproteins (LDL) particles, as well as apolipoprotein B and linoleic acid, with adjusted β values ranging from −0.204 to −0.104 per 1-standard deviation (SD) change. Additionally, 4 biomarkers showed positive associations with RPET, including leucine, isoleucine, tyrosine, and phospholipids to total lipids ratio in small HDL, with adjusted β values ranging from 0.112 to 0.132 per 1-SD change (Fig. 3, Supplementary Table S5).

After a median time of 12.3 (interquartile range: 11.6, 13.0) years of follow-up, a total of 5,714 participants in population-II developed T2DM, including 4,038 in the training set and 1,676 in the testing set. Figure 3 and Supplementary Table S6 demonstrate that after adjusting for potential confounders and correcting for multiple testing (FDR $P < 0.05$), 57 of the 64 RPET-associated metabolic biomarkers were significantly associated with the risk of T2DM, including 32 with positive associations, with adjusted HRs ranging from 1.072 to 1.439 per 1-SD change; and 25 with negative associations, with adjusted HRs ranging from 0.775 to 0.968 per 1-SD change.

### Performance of RPET-associated metabolites for T2DM stratification
The RPET metabolic state model is a stepwise Cox proportional hazard (CPH) model trained on RPET-associated metabolites. We have

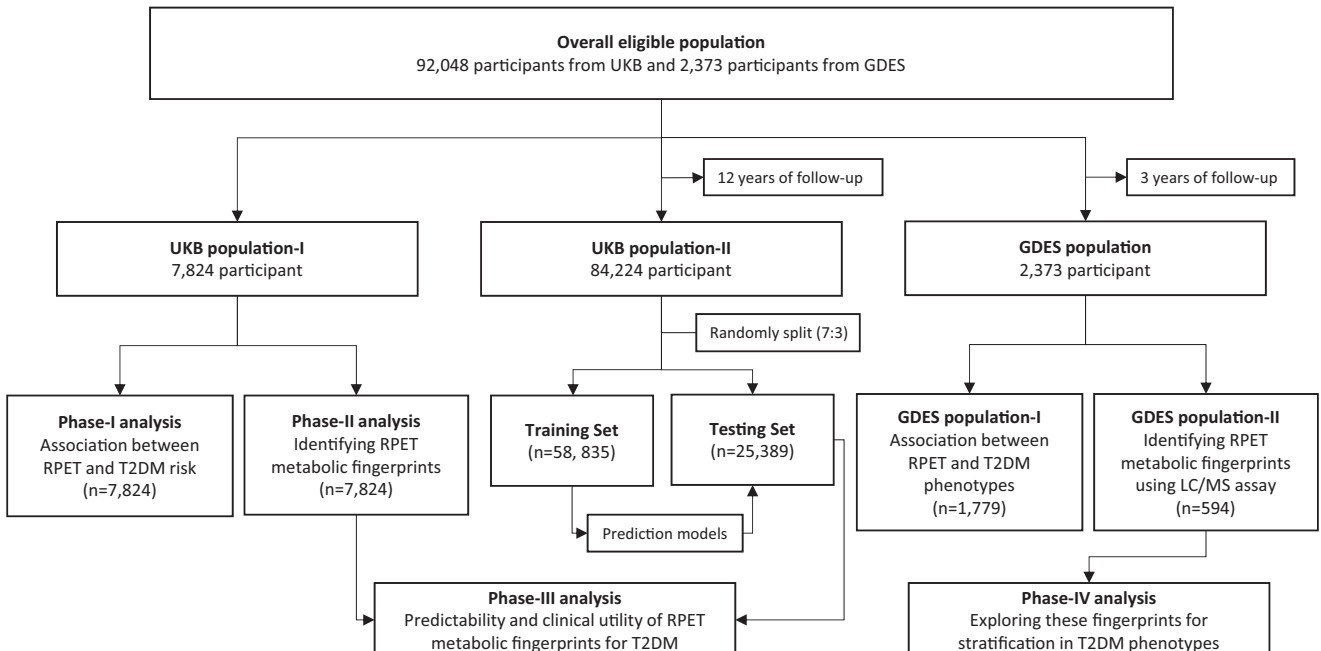

**Fig. 1 | The analytic framework of the study.** Eligible UKB participants were split into population-I for evaluating the prospective association of RPET with future T2DM (phase-I analysis) and identifying RPET metabolic fingerprints (phase-II analysis); and population-II for evaluating the prospective association of the RPET metabolic fingerprints with future risk of T2DM, as well as their incremental predictive value and clinical utility (phase-III analysis). Additional GDES population was

employed to investigate the association between average RPET with future diabetic microvascular phenotypes, and the potential of RPET metabolic fingerprints in facilitating risk stratification for these outcomes (phase-IV analysis). UKB UK Biobank, GDES Guangzhou Diabetes Eye Study; RPET retinal pigment epithelium thickness; T2DM type 2 diabetes mellitus; LC/MS Liquid Chromatography/Mass Spectrometry.

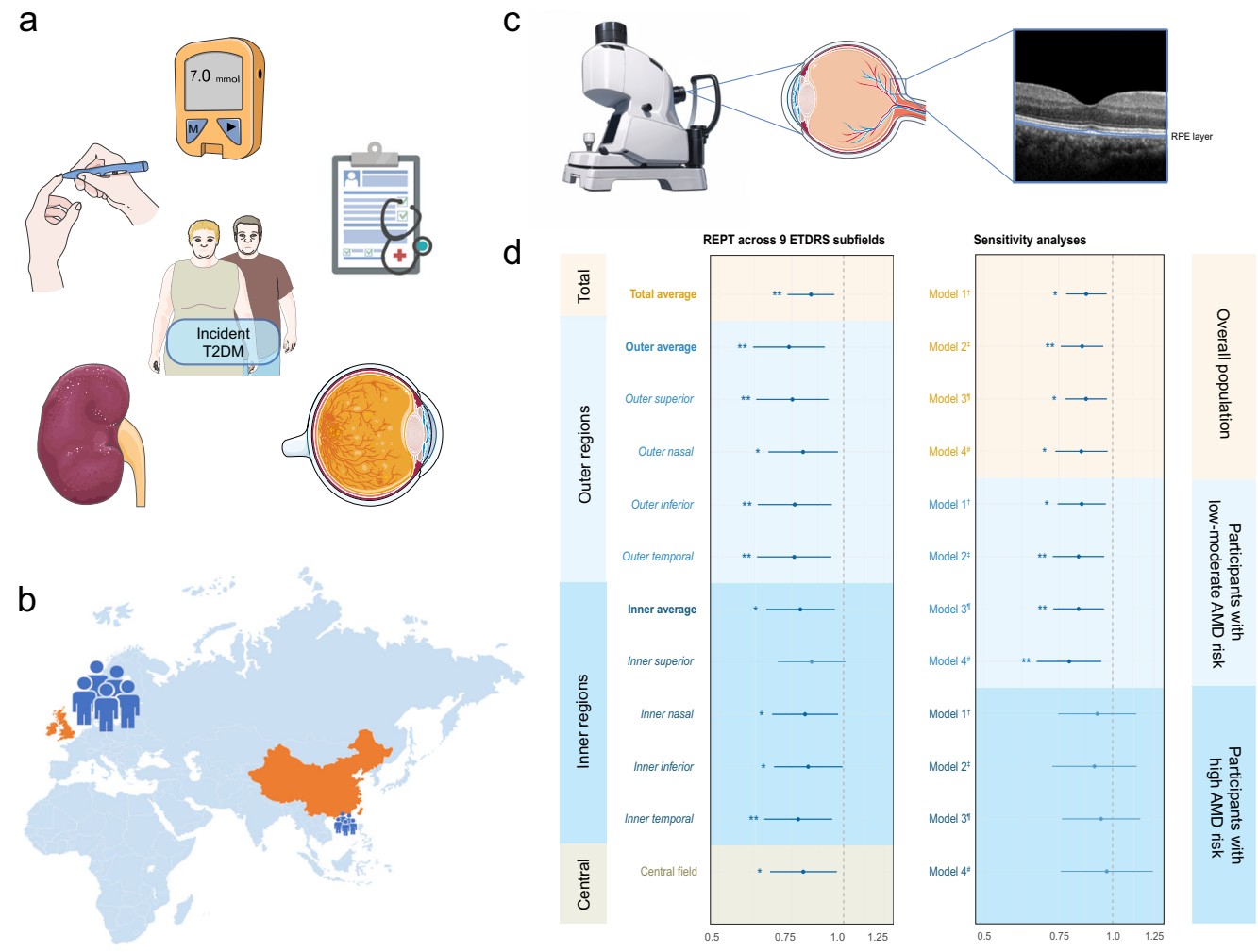

**Fig. 2 | Study overview and prospective association of RPET measurements with T2DM risk. a** Schematic representation of study endpoints. **b** Multinational participants were recruited from the UKB cohort in the UK and the GDES cohort in southern China. **c** Schematic representation of in vivo RPET measurement using optical coherence tomography. **d** Hazard ratios for incident T2DM ($n = 5,714$) per 1-SD increment in RPET across ETDRS subfields estimated with CPH models. Multiple sensitivity analyses were conducted to confirm the robustness of the results. The solid dots represent the estimated hazard ratios, with 95% CIs indicated. Asterisks and steel blue indicates significant; light blue indicates insignificant. Two-

sided statistical tests were conducted, and no adjustments were made for multiple tests. * $P < 0.05$; ** $P < 0.01$. Source data are provided as a Source Data file. Parts of **a**, **b**, and **c** were modified from Servier Medical Art (smart.servier.com) licensed under a Creative Commons Attribution 3.0 Unported License and BioRender (bio-ender.com). T2DM type 2 diabetes mellitus, RPET retinal pigment epithelium thickness, ETDRS Early Treatment of Diabetic Retinopathy Study, AMD age-related macular degeneration, UKB UK Biobank, GDES Guangzhou Diabetes Eye Study, SD standard deviation, CI confidence interval.

observed increasing T2DM event rate over RPET metabolic states and significant differences in the cumulative hazard across the four quartile-risk trajectories ($P_{trend} < 0.0001$), suggesting the rich information these fingerprints hold to distinguish individuals at different T2DM risk (Fig. 4).

The Harrell's C-statistic of the metabolic state model (C-statistic = 0.805, 95% CI:0.794, 0.817) for discriminating T2DM was significantly higher than that of all individual clinical indicators, including age (C-statistic = 0.589, 95% CI:0.575, 0.603; $P < 1.0 \times 10^{-8}$), sex (C-statistic = 0.572, 95% CI:0.559, 0.585; $P < 1.0 \times 10^{-8}$), ethnicity (C-statistic = 0.530, 95% CI:0.522, 0.538; $P < 1.0 \times 10^{-8}$), smoking (C-statistic = 0.536, 95% CI:0.522, 0.549; $P < 1.0 \times 10^{-8}$), drinking (C-statistic = 0.503, 95% CI:0.494, 0.513; $P < 1.0 \times 10^{-8}$), BMI (C-statistic = 0.714, 95% CI:0.701, 0.726; $P < 1.0 \times 10^{-8}$), WHR (C-statistic = 0.718, 95% CI:0.704, 0.731; $P < 1.0 \times 10^{-8}$), hypertension (C-statistic = 0.642, 95% CI:0.628, 0.655; $P < 1.0 \times 10^{-8}$), and use of antihypertensive (C-statistic = 0.639, 95% CI:0.626, 0.652; $P < 1.0 \times 10^{-8}$) and lipid-

lowering medication (C-statistic = 0.681, 95% CI:0.668, 0.694; $P < 1.0 \times 10^{-8}$).

The integration of these metabolic fingerprints to the model based on clinical indicators yielded a significant improvement in the C-statistic (from 0.809, 95% CI: 0.798, 0.819, to 0.837, 95% CI: 0.827, 0.847, $P < 1.0 \times 10^{-8}$) (Fig. 4). Statistically significant improvements in net reclassification index (NRI, 0.287, 95% CI:0.259, 0.324; $P < 0.0001$) and integrated discrimination index (IDI, 0.051, 95% CI:0.046, 0.066; $P < 0.0001$) were also observed for the model combining clinical indicators and RPET metabolic fingerprints. The goodness of model fit was confirmed ($P_{Hosmer-Lemeshow} > 0.9$), and decision curve analysis revealed further improvement in the clinical utility with the addition of these fingerprints (Fig. 4). Similar improvements were observed in the sensitivity analyses of constructing the RPET metabolic state model using only: (1) biomarkers associated with both thinner RPET and increased T2DM risk; and (2) biomarkers independent of ageing (Supplementary Figure S1, S2, Supplementary Tables S7, S8).

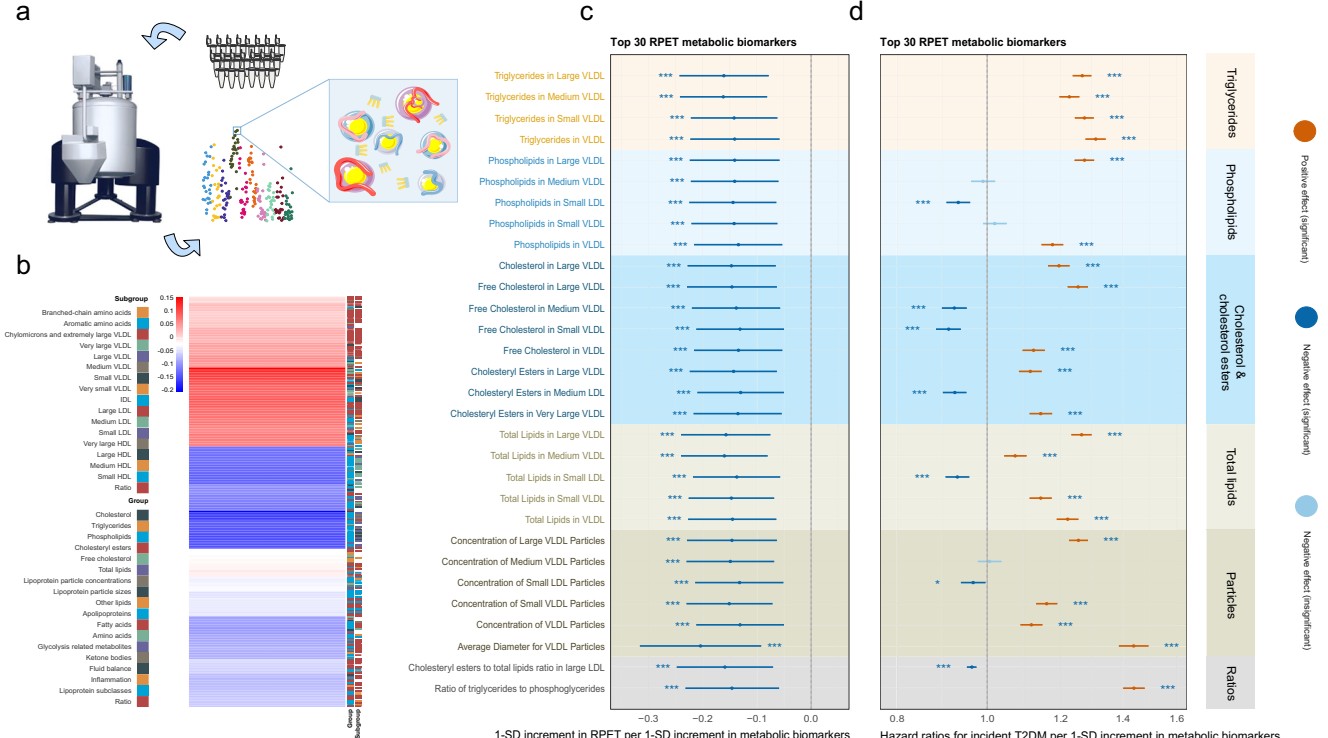

**Fig. 3 | RPET-associated metabolites and risk of T2DM. a** Schematic representation of the quantification of metabolic biomarkers in the UKB cohort. **b** Heatmap illustrating the overall associations between RPET and all 249 metabolic biomarkers. Red indicates positive effects; blue indicates negative effect. **c, d** The top 30 metabolic biomarkers associated with RPET are displayed, along with their hazard ratios for incident T2DM (*n* = 5,714) per 1-SD increment estimated with multiple linear regression models and CPH models. The solid dots represent the estimated hazard ratios, and the horizontal lines represent the 95% CIs. Asterisks and steel blue indicates significant negative effects; asterisks and brick red indicates significant positive effects; light blue indicates insignificant. Two-sided statistical tests were conducted, and BH method was employed to reduce false discovery rate for multiple tests. * *P* < 0.05; ** *P* < 0.01; *** *P* < 0.001. Source data are provided as a Source Data file. Parts of **a** were modified from Servier Medical Art (smart.servier.com), licensed under a Creative Commons Attribution 3.0 Unported License VLDL very low-density lipoproteins; IDL intermediate-density lipoproteins, LDL low-density lipoproteins, T2DM type 2 diabetes mellitus, RPET retinal pigment epithelium thickness, UKB UK Biobank, SD standard deviation, CI = confidence interval.

## RPET-associated metabolites and T2DM microvascular phenotypes

In the GDES cohort, a total of 2,373 T2DM participants underwent swept-source OCT (SS-OCT) scanning and 593 of them with Liquid Chromatography Tandem Triple Quadrupole Ion Trap Mass Spectrometry (LC-QTRAP-MS/MS) metabolomic profiling at baseline assessment were eligible. The baseline characteristics are summarized in Supplementary Table S9. After adjusting for age, sex, duration of diabetes, HbA1c, BMI, SBP, smoking, drinking, and hyperlipidemia, a lower baseline average RPET was significantly associated with an increased risk of diabetic microvascular phenotypes, including development and progression of diabetic retinopathy, rapid decline of renal function, and fast retinal capillary rarefaction in both macula and optic nerve head (ONH) (Fig. 5, Supplementary Table S10–S12). As shown in Fig. 5 and Supplementary Table S13, 58 metabolites were independently associated with average RPET after multiple test correction (all FDR *P* < 0.05), including 57 with positive associations, encompassing amino acids, FAs, benzene, nucleotides, organic acids, heterocyclic compounds, and their derivatives, with adjusted β values ranging from 1.120 to 2.856 per 1-SD change; and 1 with negative associations (quinmerac, β = −1.460, 95% CI: −2.306, −0.614). Finally, the incorporation of these metabolites into the clinical indicators-based model resulted in improvements in discriminative power and clinical utility, further confirming the role of RPET metabolic fingerprints in facilitating risk stratification for T2DM-related damages (Supplementary Fig. S3).

## Discussion

This study presents evidence for the prospective association of in vivo RPET measurements with T2DM risk in a large population-based cohort. To probe the biological underpinnings, we performed a systematic analysis of circulating metabolomics and identified 64 plasma metabolic fingerprints associated with RPET, many of which were also associated with the risk of T2DM. These fingerprints exhibited the potential to improve predictability and clinical utility for T2DM beyond traditional risk factors. Moreover, by utilizing a deeper-penetrative OCT device and a more sensitive metabolomic assay, we provided further evidence for the association of RPET with future adverse microvascular phenotypes, and the role of RPET metabolic fingerprints in facilitating risk stratification for these outcomes in a southern Chinese community-based diabetic cohort. These findings provide insights into the promising role of retinas as a window to systemic health and suggest the potential value of RPET metabolic fingerprints for individualized risk stratification for diabetes and related target organ damages.

There is growing evidence that structural and functional damage to the RPE have already taken place in patients and experimental animals with pre-diabetes and early stages of diabetes[15–17,19]. Karaca et al.[15] found RPE thinning in patients with pre-diabetes, suggesting that damage to the RPE associated with diabetes-related metabolic disorders precedes the onset of evident diabetes. Enzoly et al.[16] showed that while the overall retinal thickness of rats with early diabetes remains unchanged, decreases in RPET and RPE65 protein

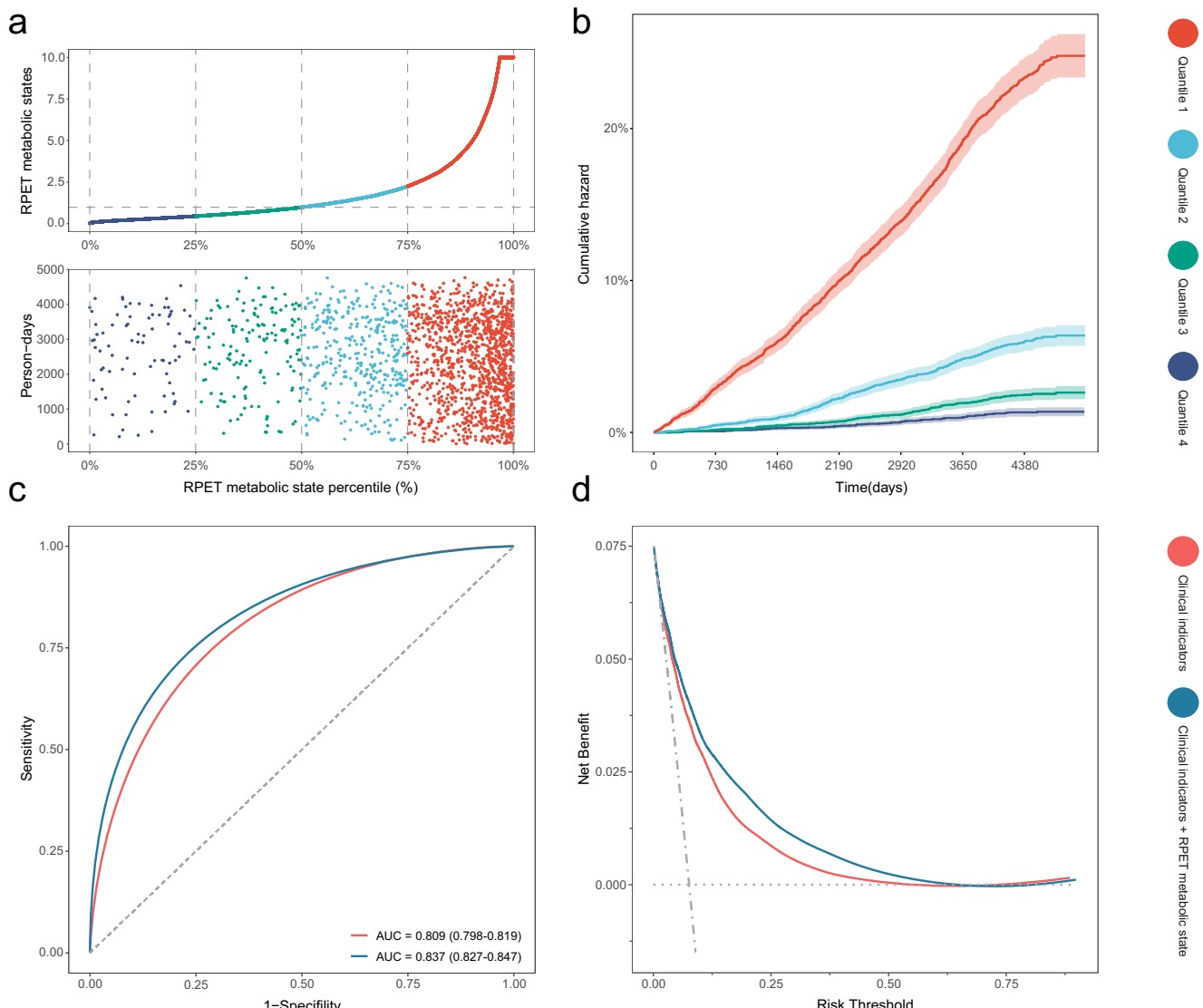

**Fig. 4 | Performance of RPET metabolic fingerprints for T2DM stratification.**
**a** Progressive increase of T2DM events across RPET metabolic states. **b** Cumulative T2DM hazard over the follow-up period, stratified by RPET metabolic state quartiles. Data are presented as observed event frequency and 95% confidence intervals. Red indicates top quartiles; sky blue indicates second quartile; green indicates third quartile; navy indicates fourth quartile. **c** Receiver operating characteristic curves of clinical indicators alone versus the combination of clinical indicators and RPET metabolic states. **d** Net benefit curves of clinical indicators alone versus the combination of clinical indicators and RPET metabolic states. Orange indicates clinical indicators only; blue indicates the combination of clinical indicators and RPET metabolic states; horizontal dotted gray line indicates treat none; and vertical dot-dash gray lines indicates treat all. Source data are provided as a Source Data file. RPET retinal pigment epithelium thickness, T2DM type 2 diabetes mellitus; AUC area under receiver operating characteristic curve.

immunoreactivity are evident. Consistently, Hammoum et al.[19] also observed a significant reduction in RPE65 staining intensity in high-fat diet-induced diabetic *Meriones shawi* at three months. In addition, RPE dysfunction concomitant with the onset of hyperglycemia has also been reported, suggesting RPE dysfunction as an early hallmark of diabetes[17]. Consistent with these studies, our analysis revealed that for every 1-SD decrease in RPET, the 12-year risk of T2DM increases by approximately 15%.

Although diabetes-related damages may collectively affect the retina and other target organs in the body, the exceptional susceptibility of the retina to the metabolic stresses may permit early manifestation of these damages. Previous studies have indicated that insulin resistance can stimulate triglyceride synthesis and VLDL production, leading to an excessive accumulation of VLDL and LDL[25–27]. Studies on NMR metabolomic profiling have also linked VLDL and LDL particle sizes and concentrations to an increased risk of T2DM[28–31].

These findings are in keeping with the adverse RPET metabolic state contributing to the increased T2DM risk observed in this study. Studies on retina have established that diabetic retinopathy and its severity are associated with multiple VLDL and LDL particles, and lipid-lowering therapy has been shown to have significant benefits in preventing and mitigating diabetic retinopathy[32,33]. Modified LDL can have toxic effects on retinal cells, including RPE cells, and their excessive accumulation in these cells may result in oxidative stress and endoplasmic reticulum stress[34], potentially leading to dysfunction and structural damages of the RPE. Consistently, in vitro studies have demonstrated that RPE cells cultured under diabetic-like conditions produce elevated levels of reactive oxygen species compared to those incubated under non-diabetic conditions[35]. All these lines of evidence indicate that a specific metabolic state that prioritizes damage to the retina before evident T2DM has already taken place, and while these subtle metabolic alterations may not have a noticeable effect on the body yet, they

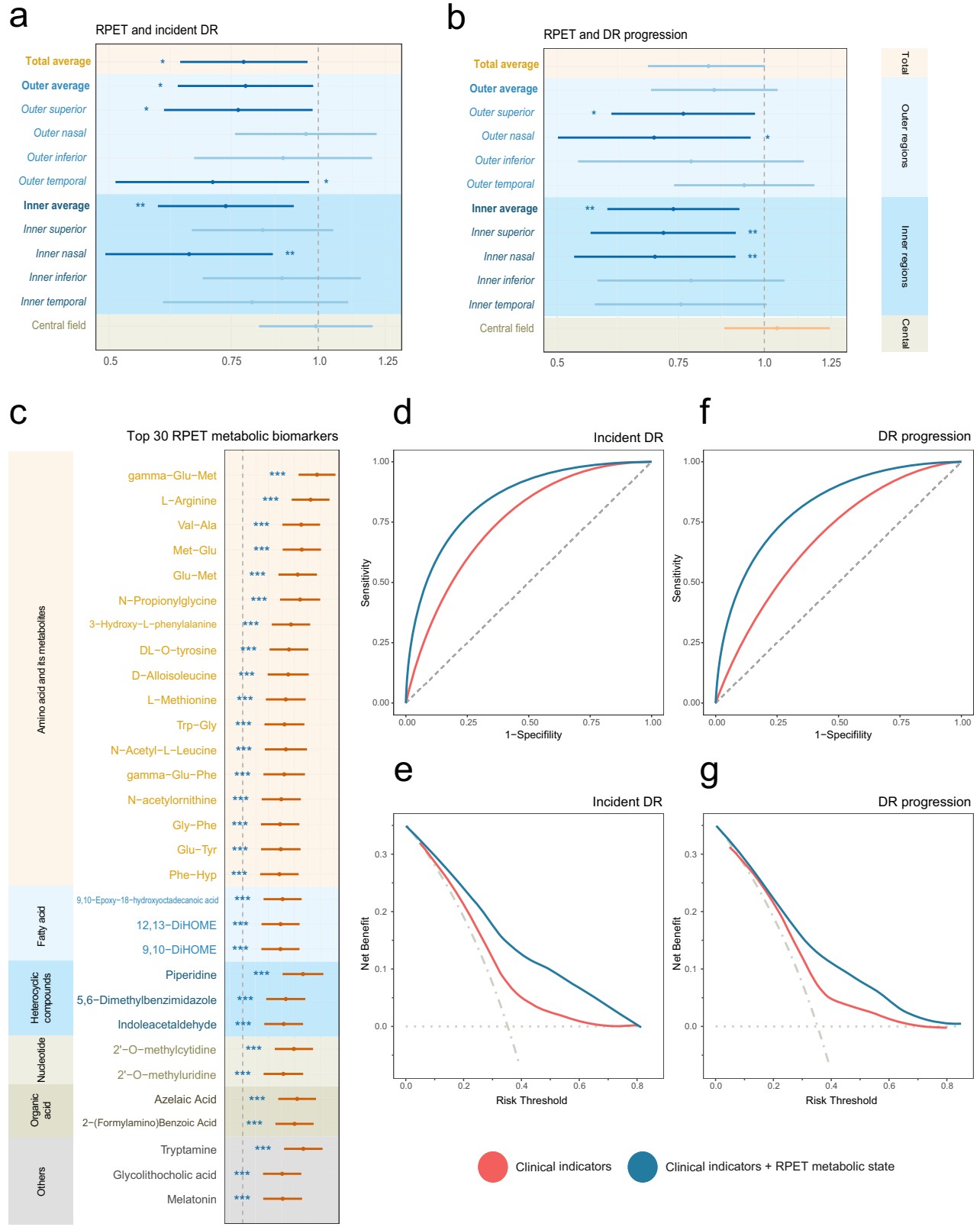

are already detectable through in vivo OCT scanning and metabolomic profiling.

To investigate the predictive value of RPET metabolic fingerprints for stratifying future T2DM and related target organ damages, we conducted receiver operating characteristic and decision curve analyses. The inclusion of these fingerprints in clinical indicators-based models led to improvements in C-statistic and clinical utility, revealing that RPET metabolic fingerprints captured residual risk that eluded quantification by traditional risk factors. Considering the proposition that T2DM might be a manifestation of premature aging[36], we conducted further investigations to compare RPET metabolomics in relation to aging and diabetes. Remarkably, similar improvements in

**Fig. 5 | RPET-associated metabolites and T2DM microvascular phenotypes.**
**a**, **b** Hazard ratios for (**a**) incident DR ($n = 161$) and (**b**) DR progression ($n = 160$) per 1-SD increment in RPET across ETDRS subfields. Asterisks and steel blue indicates significant negative effects; light blue and orange indicates insignificant. **c** The top 30 metabolic biomarkers that are associated with RPET estimated with multiple linear regression models. Data are presented as hazard ratios and 95% confidence intervals. Asterisks and brick red indicate significant. Two-sided statistical tests were conducted, and BH method was employed to reduce false discovery rate for multiple tests. **d**–**g**, Receiver operating characteristic curves and net benefit curves of clinical indicators alone versus the combination of clinical indicators and RPET metabolic states for stratifying incident DR (**d**–**e**) and DR progression (**f**–**g**). Orange indicates clinical indicators only; blue indicates the combination of clinical indicators and RPET metabolic states; horizontal dotted gray line indicates treat none; and vertical dot-dash gray lines indicates treat all. * $P < 0.05$; ** $P < 0.01$; *** $P < 0.001$. Source data are provided as a Source Data file. RPET retinal pigment epithelium thickness, DR diabetic retinopathy.

predictability and clinical utility were observed when utilizing only age-independent fingerprints for model construction. These findings suggest that the RPET metabolic fingerprints was specific markers of T2DM that transcend the confounding of the aging process. All these lines of evidence highlight the potential of these fingerprints to augment individualized T2DM risk stratification. Since the retina is highly susceptible to metabolic damage, an exploration on RPET-mediated biological changes may help further unravel the early pathogenesis of T2DM and provide insights into biological changes in early T2DM stages. More importantly, intensive interventions aimed at modifiable risk factors in identified high-risk individuals, including RPET-associated metabolites, hold the promise for reversing or interrupting the onset of T2DM and minimizing target organ damages in the future.

Given the potential limitations on SD-OCT imaging and 1H-NMR metabolomic assays, we employed more advanced SS-OCT scanning and more sensitive ultra-performance LC/MS analysis in the GDES cohort to explore the stratification value of in vivo measurement-based RPET metabolic fingerprints for diabetic target organ damages. Although SD-OCT can already achieve satisfactory in vivo retinal imaging, the major limitation of SD-OCT is the loss of resolution with increasing depths, due to sensitivity roll-off effect as the light travels deeper into the tissue being imaged[37]. With the use of longer wavelengths, SS-OCT reduces attenuation from ocular opacities and provides deeper penetration and lower scattering from the RPE[37,38], thereby facilitating more accurate RPET measurements. Moreover, SS-OCT achieves faster scan rates, which substantially reduces the impact of motion artifacts on imaging[39]. Employing more accurate RPET measurements, we provided further evidence for the independent associations of this retinal biomarker with the risks of diabetic microvascular phenotypes.

1H-NMR has been established to be a robust and reliable tool for comprehensive metabolic profiling, necessitating minimal sample preparation and providing high levels of automatability and reproducibility[40]. This approach offers a unique opportunity to investigate the intricate link underlying metabolic states and diseases in large-scale studies, including those on diabetes and its complications[40,41]. However, one of the primary limitations of NMR is its relatively low sensitivity when compared to MS-based approaches. To address this limitation, we performed LC/MS assay in an independent cohort to further investigate the value of RPET metabolic fingerprints for stratifying diabetic adverse microvascular phenotypes. Within this approach, an expanded range of RPE-associated metabolites were identified, particularly those present at low concentrations. NMR and LC/MS assays offer complementary coverage of metabolic markers, providing a more comprehensive RPET metabolic landscape. Future metabolomics studies with higher coverage were poised to improve our system-level understanding of the link underlying RPE with diabetes risks.

In analyzing the association of RPE with T2DM, it is imperative to not neglect the significant impact of AMD on RPE[42]. Numerous studies have shown that progressive degeneration and impairment of RPE function are the early and key events in AMD, which not only confounds the association of RPET with T2DM but also compromises the accuracy of macular segmentation of the OCT system[42–44].

Furthermore, AMD is considered greatly attributable to heredity, with an estimated 46%–71% of AMD variation likely to be explained by genetic factors[45,46]. Recent studies revealed RPET thinning in clinically normal individuals with AMD risk single-nucleotide polymorphisms, suggesting that AMD-related retinal changes may take place much earlier in the life course[47]. To further exclude AMD from confounding the results, we used the upper quartile of the AMD-PRS constructed based on meta-analyzed summary statistics GWAS data as a cut-off point to divide population-I into two subgroups for separate analyses, and no association was observed when analyzing participants with the highest AMD-PRS[48,49]. We speculate that RPET in participants with higher AMD-PRS may have been confounded by other complicated mechanisms associated with AMD pathologies, thus leading to the absence of the associations. In view of this, we excluded participants with high AMD-PRS when identifying RPET-associated metabolites in phase-II analyses to avoid potential confounding. However, we did not exclude individuals with high AMD-PRS in phase-III analysis, given that genomic data are unlikely to be available when applying the RPET metabolomic fingerprints as a screening modality for T2DM, though doing so might further improve the predictability.

The present study boasts several strengths. First and foremost, it benefits from a sizable sample size and extensive longitudinal follow-up, both of which serve to enhance the robustness of the study's findings. Second, this study collected a wide range of covariates to adjust for potential confounders. Third, to obviate the impact of pre-clinical AMD on RPET measurements, the AMD-PRS was incorporated, and participants were stratified accordingly for separate analyses. This approach has yielded evidence for the prospective association of RPET with T2DM risk in a large, population-based cohort. Fourth, to probe the underlying link between the association, we identified RPET metabolic fingerprints and evaluated their added predictability and clinical utility for T2DM in an even larger population. Fifth, by using longer wavelengths, more accurate RPET measurements were acquired with the state-of-the-art SS-OCT technology in the GDES cohort to further investigate the association between RPET and diabetic microvascular phenotypes. Finally, the complementary LC/MS assay further broadened the metabolic landscape of RPET and provided additional evidence for their value in stratifying diabetic organ damages, even in a different race.

Certain limitations of this study should also be acknowledged. First, some participants lacked first diagnosis data in their inpatient records, so the date of diagnosis for these participants was defined based on self-reported questionnaires, which could lead to potential recall bias. Second, participants in population-I were younger, male, more educated, had higher income, lower BMI, smoked less, and less likely to be hypertension. Therefore, caution may be warranted when generalizing the RPET metabolomic fingerprints to other populations. However, the concept was confirmed in both UK and Chinese populations. Third, our results showed that a decreased RPET based on in vivo OCT imaging was associated with diabetes and its organ damages; however, it is important to acknowledge that the measurement of thickness was inferred by the hyperreflective band situated on the outer layer of the retina. Consequently, it must be recognized that in certain instances, a dissociation may arise between alterations in optical reflectance and actual changes in thickness, thus posing

challenges in establishing a direct functional relationship thus far. Nevertheless, we bolstered the robustness of our results by extending the association between RPE thinning and diabetes-related organ damages in an independent cohort of a distinct ethnic background, utilizing the more advanced and higher-resolution SS-OCT. Fourth, the metabolomics profiling was based on a single sample collection at baseline in both cohorts and therefore may not reflect the fluctuations of these metabolites over time. Fifth, we did not account for the potential diurnal change of RPE. Finally, although a comprehensive range of confounders was adjusted for in the current study, potential residual confounders that could not be excluded may still exist.

In summary, we provide evidence for the prospective association of RPET with risks of T2DM and related microvascular phenotypes. Our evidence suggests that alterations of in vivo measurements of RPET may represent a specific metabolic state of the human body that indicates an elevated risk of future T2DM and target organ damages. RPET metabolic fingerprints improve individualized risk stratification for T2DM and its microvascular phenotypes beyond traditional risk factors, providing insights into the promising role of the retina as a window to systemic health.

## Methods

### Study design and population

Our study adheres to the Guidelines of the Ministry of Science and Technology (MOST) for the Review and Approval of Human Genetic Resources. From 2006 to 2017, UK Biobank (UKB) recruited over 500,000 men and women aged 40–69 years old from 22 assessment centers in the UK. The detailed study design is described elsewhere[50]. Baseline assessments were conducted through questionnaires, medical interviews, physical examinations, and blood tests (2006–2010). Health outcomes were linked to hospital and primary care records as well as cancer and death registries. The Guangzhou Diabetes Eye Study (GDES) is a community-based cohort study that recruits over 2,300 T2DM patients aged 35–85 in Guangzhou, China[51]. Baseline assessments were conducted from 2017 to 2019. The study was approved by the Northwest Multicenter Research Ethics Committee (11/NW/0382) and the Ethics Committee of Zhongshan Ophthalmic Center (2017KYPJ094), with written informed consent obtained from all participants. The study was performed in accordance with the STROBE statement[52] and the STARD guidelines[53] (Supplementary Tables S14, S15).

The overall design of the study consists of four parts (Fig. 1). UKB participants were divided into three non-overlapping groups[54]: (1) population-I with both OCT measurements and metabolomics data; (2) population-II with metabolomics data only and without OCT measurements; and (3) the remaining others excluded from the analysis. Population-I was used to evaluate the prospective association of RPET with future T2DM (phase-I analysis) and identify RPET metabolic fingerprints (phase-II analysis). Population-II was used to evaluate the prospective association of the RPET metabolic fingerprints with future risk of T2DM, as well as their incremental predictive value and clinical utility (phase-III analysis). Additional analyses on diabetic microvascular phenotypes in the GDES population were described in a separate section below (phase-IV analysis).

### Nuclear magnetic resonance metabolomic profiling

High-throughput nuclear magnetic resonance (NMR) platform (Nightingale Health, Finland) was used to obtain 249 metabolic metrics from the plasma samples of the participants[55–57]. Sample collection was undertaken at baseline in 22 local assessment centers across the UK between 2007 and 2010. Cryopreserved EDTA plasma samples were thawed and centrifuged and then the supernatant was mixed with phosphate buffer. The samples were then loaded onto a cooled sample changer, and two NMR spectra of each plasma sample were recorded using a 500 MHz NMR spectrometer (Bruker AVANCE IIIHD). One spectrum characterized resonances produced mainly by proteins and lipid lipoprotein particles, whereas the other detected low-molecular-weight metabolites. After accredited quality controls[58], metabolic metrics, including 168 presented in absolute levels (i.e., fatty acids, glycolytic metabolites, ketone bodies, amino acids, lipids, and lipoproteins) and 81 presented as ratio values, were quantified using the Nightingale Health Biomarker Quantification Library 2020.

### Spectral-domain OCT imaging

Spectral-domain OCT was performed in an enclosed darkroom using a Topcon 3D OCT-1000 Mk II (Topcon, Inc., Oakland, NJ, USA) using a 3D 6 × 6 mm macular volume scan mode centered at the fovea with a scan density of 512 A-scans × 128 B-scans within 3.7 seconds. The Topcon Advanced Boundary Segmentation algorithm[59] (version 1.6.1.1) automatically segmented the retinal layers and delineated RPE-Bruch's membrane complex[60,61]. The inner boundary corresponds to the photoreceptor outer segment-RPE boundary, while the outer boundary corresponds to the BM-choroid boundary. Measurements were acquired across 9 Early Treatment of Diabetic Retinopathy Study (ETDRS) subfields. The parafoveal average RPET was determined by calculating the average RPET of the four inner-ring quadrants, while the perifoveal average thickness was calculated by averaging the RPET of the four outer-ring quadrants. The overall average RPET was computed as a weighted average of the sectoral thickness measurements. This was done using the following formula: $(1/36 \times \text{center}) + (1/18 \times \text{sum}$ of the four inner-ring quadrants' thickness) + $(3/16 \times \text{sum}$ of the four outer-ring quadrants' thickness)[62]. Image quality scores, inner limiting membrane metrics, validity count, and motion metrics were recorded for quality control, whereby images with low signal strength (Q < 45) or poor segmentation or centration (the worst 20% of each indicator) were excluded. If both eyes were eligible, then one eye was randomly selected for further analysis.

### Ascertainment of T2DM

The Hospital Episode Statistics database, Scottish Morbidity Record, and Patient Episode Database were used to obtained inpatient records in England, Scotland, and Wales, respectively. The determination of T2DM was based on ICD-10 code E11. Death dates were obtained from national datasets with the NHS Digital (England and Wales) and NHS Central Register (Scotland). The follow-up period was from March 16, 2006, to March 31, 2021. Person-days for each participant were calculated from the date of baseline assessment to the date of disease onset, death, or end of follow-up, whichever came first.

### Assessment of covariates

Face-to-face interviews, detailed touchscreen questionnaires, and physical measurements were conducted on all participants at baseline (2006–2010). The self-administered questionnaire covered demographic and socioeconomic factors (age, gender, race, education, Townsend deprivation index, and household income), lifestyle factors (smoking and drinking status), and medical history, including use of lipid-lowering medications, antihypertensives. Baseline diseases were determined by a combination of inpatient records, touchscreen questionnaires, and verbal interviews. Baseline body mass index (BMI) was calculated as weight divided by height squared, and waist-to-hip ratio (WHR) as waist circumference divided by hip circumference. Visual acuity was tested with traditional LogMAR charts. The refractive error was measured using an autorefractor (Tomey, Japan), and spherical equivalent (SE) were calculated based on autorefraction results. Intraocular pressure (IOP) was measured using an Ocular Response Analyzer (Rerchert, USA). Genotyping was conducted using the UK BiLEVE Axiom Array or the UKB Axiom Array, and quality control and imputation of the genetic data were described elsewhere[63]. Polygenic risk score (PRS) of age-related macular degeneration (AMD) (AMD-PRS) was computed using alleles and their effect based on meta-analyzed summary statistics GWAS data across 26 studies[48,49].

## Additional analyses on diabetic microvascular phenotypes

We conducted additional analysis in the GDES cohort to investigate the association between average RPET with future diabetic microvascular phenotypes, and the potential of RPET metabolic fingerprints in facilitating risk stratification for these outcomes. A total of 2,373 GDES participants with type 2 diabetes mellitus, who met similar eligibility criteria as UKB participants, were included in the phase-IV analysis, and were further divided into two groups: (1) GDES population-I, which included OCT measurements only; and (2) GDES population-II, which included both OCT measurements and metabolomics data. GDES population-I was used to analyze the association of average RPET with the risks of T2DM microvascular phenotypes, while GDES population-II was used to identify RPET-related metabolites and explore their value in stratifying these outcomes.

Swept Source OCT (SS-OCT, DRI OCT Triton; Topcon, Japan) was used to measure RPET across nine ETDRS subfields at baseline, with a 3D Macula Cube 7 × 7 mm scan mode centered at the fovea. This instrument is characterized by high scanning speed and deep penetration, which is especially suitable for imaging of deep structures. Each OCT scan is performed through an internal fixator, and the fixation is monitored by the instrument's built-in fundus camera. Additionally, blood flow angiograms of the microvasculature in macula and ONH region were acquired using SS-OCT angiography, with an Angio Macula 6 × 6 mm scan mode centered on the fovea and an Angio Disc 6 × 6 mm scan mode centered on the disc, respectively. Automatic image segmentation was conducted using a built-in software (IMA-GEnet 6, Version 1.22). Metabolomic profiling was conducted using LC-QTRAP-MS/MS (LC, ExionLC AD, SCIEX, USA; MS, QTRAP® System, SCIEX, USA)[64,65]. During instrumental analysis, one quality control sample is analyzed for every ten actual samples to monitor the reproducibility of the analytical process. To further ensure the accuracy of the results, blank samples are also included in the experiment. These samples contain no analyte, and their peaks reflect whether there were any residuals during the assay. Internal standard samples with known concentrations were included, and their coefficients of variation are controlled at <5%. DR diagnosis, including its severity, was graded by the same ophthalmologist using the modified Airlie House classification system with a severity score[66]. Overnight fasting venous blood and mid-stream urine samples were collected to assess HbA1c, creatinine, total cholesterol, triglycerides, HDL and LDL cholesterol, and urine microalbumin.

## Statistics

R (version 4.2.2) and Stata/MP (version 17.0) was used for all data analyses and presentation of results. Continuous variables were presented as mean (SD), and categorical variables were presented as number (percentage). Student's t-test and chi-square test were used to compare continuous and categorical variables, respectively. The z-score normalization was performed for all metabolic measures to ensure comparability across metabolites. No statistical method was used to predetermine sample size. No eligible data meeting criteria for inclusion were excluded from the analyses. The experiments were not randomized, and the investigators were not blinded to allocation during experiments and outcome assessment.

In the phase-I analysis, the association of in vivo RPET measurements with the risk of T2DM was evaluated using CPH models. Considering the significant impact of AMD on RPE, we divided population-I into two subgroups according to the upper quartile of AMD-PRS for separate analyses. Participants with high AMD-PRS were at higher genetic risk for AMD and may therefore be susceptible to preclinical AMD pathologies. The covariates included age, sex, ethnicity, assessment center, household income, Townsend deprivation index, education, smoking, drinking, BMI, hypertension, hyperlipidemia, IOP, SE, and use of antihypertensive and lipid-lowering medication. Variables that reached a threshold of $P < 0.1$ were adjusted in the multivariate

models. Sex-specific analyses were conducted, using the same approach without the inclusion of sex as a covariate. Sensitivity analyses was performed by: (1) adjusted for more comprehensive ethnicity classifications with further subdivisions for South Asian, East Asian, Black, and mixed races; (2) further adjusted for biological age[23]; (3) further adjusted for frailty score[24]; and (4) excluding participants with cancer, cardiovascular, and pulmonary diseases. HRs with 95% CI showed the association of baseline RPET with the risk of T2DM.

In the phase-II analysis, associations between 249 metabolic biomarkers and average RPET were assessed using multiple linear regression models after adjusting for age, sex, ethnicity, assessment center, household income, Townsend deprivation index, education, smoking, drinking, BMI, IOP, SE, use of lipid-lowering medications, and anti-hypertensive medications. Participants with highest AMD-PRS were excluded. β values and 95% CIs were used to represent the change in RPET cause by 1-SD change for each biomarker. The Benjamini-Hochberg (BH) method was employed to reduce FDR for multiple tests.

In the phase-III analysis, participants from population-II were randomly divided into training and testing set at a ratio of 7:3 (Fig. 1). The association between RPET metabolic fingerprints and the risk of T2DM was evaluated using CPH models, which was adjusted for the same covariates as in the phase-I analysis, with the BH method for multiple testing correction. The RPET metabolic state model is a stepwise CPH model trained on (1) all RPET-associated metabolic biomarkers; (2) only biomarkers that showed negative associations with RPET and positive associations with T2DM risk; and (3) only biomarkers independent of ageing. Participants in the testing set were divided into four quartiles based on the calculated RPET metabolic states, and the risks of developing T2DM were compared among groups. To assess the predictivity of the RPET metabolic states for T2DM, the Harrell's C-statistics was calculated, and their predictive value for T2DM was compared with those of individual clinical indicators. The added predictability of these metabolites for the risk of T2DM was also evaluated compared to the clinical indicators-based model. The NRIs and IDIs were also computed. The goodness of model fit was assessed using Hosmer-Lemeshow test. Finally, decision curve analyses were conducted to estimate the benefits in clinical utility.

In the GDES cohort, the associations of average RPET with the risk of diabetic adverse microvascular phenotypes were evaluated using logistic regression models, after adjusting for age, sex, duration of diabetes, HbA1c, smoking, drinking, BMI, SBP, and hyperlipidemia. The associations between each metabolite and RPET were assessed using multiple linear regression models after adjusting for the same covariates as above, with the BH method for multiple testing correction. C-statistics were calculated, and decision curve analyses were conducted to investigate the role of these metabolites in facilitating risk stratification for these outcomes. A $P$ value < 0.05 was statistically significant, with exceptions for where specified.

## Reporting summary

Further information on research design is available in the Nature Portfolio Reporting Summary linked to this article.

## Data availability

All the data utilized in this study, including imaging, NMR, and geno-typing data from the UKB, are available via data access procedures (http://www.ukbiobank.ac.uk). Permission to use the UKB Resource was obtained via a material transfer agreement as part of Application 62443, 62489, 62491 and 62525. Raw data from the GDES analyzed in the current study are not publicly available due to HIPAA compliance and were used with Zhongshan Ophthalmic Center institutional permission for the purposes of this project. All requests for access to in-house data will be addressed to the corresponding authors, Dr. Wei Wang (Email: wangwei@gzzoc.com), and will be processed in accordance with Zhongshan Ophthalmic Center guidelines. Guangzhou

Diabetic Eye Study Group will assess all requests based on the purpose of data request, and it may take up to 90 days to process the request. A material-transfer or data-usage agreement will be required between Zhongshan Ophthalmic Center and the receiving organization, and the requesting organization must state the intended purpose of the data transfer and provide assurances that the transferred data will only be used for non-commercial academic and educational purposes in compliance with Zhongshan Ophthalmic Center institutional guidelines. Source data are provided with this paper.

## Code availability

R Scripts used for analyses are available at GitHub Repository https://github.com/Yangshp5/RPEMet[67].

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

## Acknowledgements
We thank Zhuoyao Xin, Mei Kei Chan, Jiaqing Chu, and Shawn Zhang for their valuable contributions in figure presentation. Illustrations were created with biorender.com and smart.servier.com. This study was funded by the Hainan Province Clinical Medical Center, the National Natural Science Foundation of China (82371086, W.W.), the Natural Science Foundation of Guangdong Province (2020A1515011282, S.C.), and Global STEM Professorship Scheme (P0046113, M.H.). We would also like to thank all the participants and staff in the UKB and GDES.

## Author contributions
These authors contributed equally: S.Y., Z.Z., S.C. Study concept and design: W.W., Z.Z., M.H.; Acquisition, analyses, or interpretation: All authors; Drafting of the manuscript: S.Y., W.W., Y.Y.; Critical revision of the manuscript for important intellectual content: All authors; Statistical analyses: W.W., S.Y., Y.Y.; Obtained funding: W.W., S.C., M.H.; Administrative, technical, or material support: S.Y., W.W.; Study supervision: W.W., Z.Z., M.H.

## Competing interests
The authors declare no competing interests.
