## [Peer Review File · Nature Communications]

REVIEWER COMMENTS

Reviewer #1 (Remarks to the Author):

The authors have identified that the apparent thickness of the RPE (not clear in which subfield), measured by SD-OCT is a predictive imaging biomarkers of diabetes mellitus type 2 at 12 years. They have identified metabolomics markers associated with the RPE thinning and have then validated these lipidomic markers with the risk of diabetes in a very large population. They conclude that RPE thinning helps predict the risk of developing diabetes. While the study is interesting and original, based on large cohort from the UKB, the main parameter, which is RPE thickness in poorly reliable. To strengthen their results they need to include also the thickness of the outer retina (from ELM to RPE) and the inner retina (ganglion and fiber layers), since RPET might be modified by the other layers.

Explanations on lipidomics finding, RPET and possible RPE alteration are not convincing.

Introduction

L. 78 The authors should moderate their statement that there is no effective biomarkers available for early diagnosis. They should cite several reviews on several omics as biomarkers of prediabetes and early diabetes.

L. 84 The authors cite papers showing that reduction of retinal layers thickness is observe before diabetes is diagnosed. But in there study, they do not include the measurement of the different retinal layers, although these data are available. These data should be included.

The authors should also discuss the impact of retinal thinning on the reflectivity of the RPE/ Bruch complex and even measure its potential impact.

L. 87 The authors did not cite the first papers showing that outer retinal barrier and RPE alterations are the first abnormalities in types 2 diabetes models [2013. doi: 10.1016/ and j.ajpath.2011.04.018. Epub 2011 Jun 2].

These papers should be included since they cite RPE observations on type 2 diabetic models.

L. 91-L.94 The authors should moderate their statement here. They do not show any sign of RPE alteration in the present study. They show a reduction in thickness measured by OCT, which cannot be correlated so far to any functional consequence.

Methods

Populations: Could the authors specify the inclusion/ exclusion criteria in the different populations? This information is crucial to understand the study design and should not require to consult another reference.

There are insufficient details in the method used to quantify RPET using SD-OCT.

It is not clear where RPE thickness was measured (which subfields?)

and how it was measured. Did the author measure the RPE and Bruch membrane complex or only the RPE? Did they include the interdigitations in the RPE measurement? How could the author ascertain that retinal thinning and pigments have no influence on the RPE reflectivity?

The other retinal layer thickness should be included in the dataset since the data are available.

Several papers showed that the subfoveal RPE/ Bruch membrane complex thickness decreases with aging and hypertension. These papers must be cited and their methodology must be compared to the methodology used in the present paper [doi: 10.1177/11206721221101372. Epub 2022 May, doi: 10.1038/s41433-021-01911-5. Epub 2022 Jan 20.]. Since diabetes can be considered as a premature aging, it could be interesting to compare metabolomics associated with RPET reduction during aging and in diabetes to ensure specificity to diabetes.

L. 328 Is there information of cardiovascular/ kidney complication at the time of diabetes diagnosis?

Results

The authors should include the thickness of other retinal layers (particularly the ganglion cell and nerve fiber layers) and compare their relevance to the RPET. How can they ascertain that RPET is a better biomarker for early diabetes detection as compared to other retinal layer thickness, that can be measured with more precision.

Reduction in ganglion cell layer/ Nerve fiber layer thickness was shown to be associated with diabetes [doi: 10.1097/OPX.0000000000001650., doi: 10.1016/j.oftale.2022.02.009, DOI: 10.1159/000444498, doi: 10.1038/s41433-020-1020-z].

Discussion

L.181

It seems that the authors establish a direct link between the thickness of a monolayer of cells and functional alterations of this cell layer. The volume of the pigmented epithelium varies with multiple factors including osmotic factors or the circadian rhythm. The OCT does not allow to measure with precision the height of this epithelium and measures in reality all the interdigitations between the villi of the PE and the external segments of the photoreceptors until the choriocapillaris. It is enough that the phagocytosis activity is reduced so that this height can vary without being associated with any cell layer assay. The pigmentation of the PE can also influence its reflectivity and it is likely that the thickness appears greater if there is more pigment and thinner in OCT if there is less pigment. But this does not mean that PE is pathological.

The article cited by the authors shows transient functional abnormalities of the electrical function of the RPE at the time of the installation of hyperglycemia which is understandable since this activity depends on ionic transport. The other cited paper reports changes in the RPE and their thinning observed in the models of diabetes, as event secondary to degeneration of the outer segments of the cones.

Morphological studies of the epithelium in type 2 diabetics show changes in the cytoskeleton, cellular edema and recruitment of microglial cells to the cell contact, which should result in an increase in thickness measured in OCT. Moreover, several studies in humans have confirmed the presence of microglial cells in the outer retina of diabetic patients without retinopathy.

The apparent thinning of the RPE/Bruch layer could be due to a dilution of the pigment due to RPE proliferation. The skin color of the patients should also be taken into account and the role of pigmentation on the measured thickness should be analyzed.

Since metabolomics identified lipid pathways, the thickness of the outer segments and of all retinal layers is necessary to ensure that the RPET change is not secondary to abnormal cone function due to change in vesicular transports of lipoproteins from the choroid to the RPE and the photoreceptors.

Reviewer #2 (Remarks to the Author):

This is an interesting study. The reviewer has some comments for the authors to consider.

1. The authors only considered the effect of age-related macular degeneration on the retinal pigment epithelium. To our knowledge, retinitis pigmentosa and Stargardts disease, among others, are retinal degenerative diseases that are accompanied by RPE dysfunction. I would like the authors to provide information on other diseases or to add the limitation in the discussion.

2. Typically, adjustment variables are derived from screening processes such as univariate analysis or based on disease research experience. In the first phase of analysis, the investigators increased the number of adjustment variables based on model I. I note that the adjustment variables in both model I and model II contain population characteristics. What is the rationale for adjusting the models in this order?

3. In the third stage of the analysis, the authors randomly divided the participants from Population-II into training and validation groups in a 1:1 ratio, for which we only provide suggestions that trying to increase the grouping of the training set sample ratio may yield better model training results.

4. The study title "Metabolic Fingerprinting on Retinal Pigment Epithelium Thickness for Individualized Risk Prediction of Type 2 Diabetes Mellitus" might be inaccurate. Individualized risk prediction requires clear prediction results and evaluation criteria. We believe that the results of this study can only show that metabolomic fingerprinting can help improve individualized risk stratification in T2DM.

5. The main results of the study include the identification of metabolic fingerprints associated with reduced RPET and metabolic fingerprints to improve risk stratification, and I suggest expanding the discussion section to include validation of the results of metabolic fingerprint-related features or previous studies.

Reviewer #3 (Remarks to the Author):

This paper provide evidence that the reduced RPE thickness (RPET) is a predictor of type 2 diabetes development. In addition, the authors found that serum metabolic fingerprints (mainly lipid-related) associated with RPET were also predictors of future type 2 diabetes development beyond the traditional risk factors.

This is a well-written and well-designed paper.

Major point:

There authors should give some explanation regarding the potential reasons why the metabolites (all lipid-related) detected in serum are related to the RPE thinning, as well as their role in the pathogenesis of type 2 diabetes.

Minor point:

Please specify the number of patients in whom the metabolomic study was performed.

Reviewer #4 (Remarks to the Author):

Herein Yang et al. conducted a large population-based cohort study and found reduced retinal pigment epithelium thickness (RPET) drastically increased the risk of onset and development of type 2 diabetes mellitus (T2DM) after adjusting for covariates like age, sex, ethnicity, income, education, smoking and others. Then they identified several metabolite biomarkers for RPET via an NMR-based plasma metabolomics method. Furthermore, meta-RPET score calculated from RPET-related metabolomics changes improved T2DM predictability and achieved individualized risk stratification for T2DM. Overall, this study is an observational study, but lack of mechanisms. The causal association between lipids/lipoproteins and RPET remains obscure. Moreover, I am also worried about the quantitative analysis of so many lipoprotein parameters using NMR spectroscopy. Therefore, I do not recommend this work to be published in Nature Communications. Major comments or suggestions as follows:

(1) The absolute quantification of lipoprotein parameters is of great importance in this study, but the detailed key information is insufficient. Although the authors cited the detailed protocol on quantitative serum NMR metabolomics, plasma sample was used in this study, so which type of anticoagulants was used? For example, EDTA seriously affects NMR metabolic signals. In addition, the main NMR parameters should be provided for users.

(2) The authors reported that metabolic metrics including 168 metabolites in absolute levels and 81 as ratio values were identified in plasma sample, which is probably a little bit difficult using 500 MHz NMR spectrometer. Several macromolecules such as lipids, proteins or lipoproteins in plasma will overlay small molecular metabolites and make their quantifications impossible. The authors must show the characteristic NMR spectrum of plasma and provide the detailed information of metabolite assignments.

(3) Lipoproteins-associated metabolites were found to be more related to RPET, but what is the causal relationship between them? Additional in vivo and in vitro studies need to explore the potential mechanisms.

Responds to the reviewer's comments:

Replies to Reviewer #1

Comment #1. The authors have identified that the apparent thickness of the RPE (not clear in which subfield), measured by SD-OCT is a predictive imaging biomarkers of diabetes mellitus type 2 at 12 years. They have identified metabolomics markers associated with the RPE thinning and have then validated these lipidomic markers with the risk of diabetes in a very large population. They conclude that RPE thinning helps predict the risk of developing diabetes. While the study is interesting and original, based on large cohort from the UKB, the main parameter, which is RPE thickness is poorly reliable. To strengthen their results, they need to include also the thickness of the outer retina (from ELM to RPE) and the inner retina (ganglion and fiber layers), since RPET might be modified by the other layers. Explanations on lipidomics finding, RPET and possible RPE alteration are not convincing.

Response: We extend our sincere gratitude for the constructive feedback and valuable suggestions provided, which have immensely contributed to enhancing the quality of our manuscript. We have carefully examined all the specific comments and made significant revisions to the manuscript accordingly. It is our hope that these changes have effectively addressed your concerns and improved the overall clarity and cohesiveness of our work.

- In response to your suggestions, we have provided a more detailed description of RPET measurements and conducted additional analyses of other retinal layers to strengthen our results. Moreover, to mitigate the potential impact of skin color on RPET measurements, we carried out a sensitivity analysis that adjusted for more comprehensive ethnicity classifications. Furthermore, we conducted additional decision curve analyses to confirm the clinical utility of these fingerprints across a wide range of medical decision threshold scenarios.
- As diabetes can be viewed as a form of premature aging, we compared diabetes-associated with aging-associated RPET metabolomics and performed a sensitivity analysis by incorporating only RPET metabolomics that are

independent of aging in the RPET metabolic state model.

- Additionally, given the potential limitations of SD-OCT and proton NMR-based metabolomics assays, we conducted additional experiments employing SS-OCT with deeper penetration and higher resolution and LC/MS metabolomics assays with higher sensitivity in a southern Chinese diabetes cohort. Our additional analysis further confirmed the discriminative improvements and incremental clinical utility of RPET-associated metabolic alterations for stratifying diabetic microvascular phenotypes.
- Finally, we have moderated our statements throughout the text and highlighted in the **Discussion** section that our findings are based on measurements derived from *in vivo* OCT scanning, which cannot currently be correlated with any functional consequences. Point-to-point responses to more specific comments are listed below. We hope that the revised manuscript meets your expectations.

Comment #2. Introduction: L. 78 - The authors should moderate their statement that there is no effective biomarkers available for early diagnosis. They should cite several reviews on several omics as biomarkers of prediabetes and early diabetes.

Response: Thank you very much for pointing out this important issue. We have moderated our statement and pointed out that “*seminal risk factors and models have been established to improve risk assessments and prevention of T2DM*” (**Page 4, Line 70**). Several articles and reviews on biomarkers for diabetes, including those on omics, have been cited in the revised manuscript. Hopefully our introduction become more balanced and reasonable.

Comment #3. Introduction: L. 84 - The authors cite papers showing that reduction of retinal layers thickness is observe before diabetes is diagnosed. But in there study, they do not include the measurement of the different retinal layers, although these data are available. These data should be included. The authors should also discuss

the impact of retinal thinning on the reflectivity of the RPE/ Bruch complex and even measure its potential impact.

Response: Thank you for your valuable suggestions. To further unravel the association of retinal thickness with future T2DM, we have conducted additional analyses of other retinal layers in the revised manuscript to strengthen our results (*Supplementary Table S13*). For your convenience, we have provided a summary of the results regarding the associations between other retinal layers and incident T2DM below. Among overall average retinal thickness, average RPE thickness, average external limiting membrane-inner segment outer segment (ELM-ISOS) thickness, average ganglion cell-inner plexiform layer (GC-IPL) thickness, and average retinal nerve fiber layer (RNFL) thickness, only average RPE thickness demonstrated significant association with the risk of future T2DM, after adjusted for age, sex, ethnicity, household income, Townsend deprivation index, education, smoking, drinking, body-mass index, and use of antihypertensive and lipid-lowering medication (all of which reached a threshold of $P < 0.1$ in the univariate models).

In these cited papers you kindly mentioned, Sahin et al.¹ discovered a thinning of ganglion cell complex (GCC) thickness in several ETDRS subfields in participants with prediabetes. However, they only measured the GCC layer and RNFL, without assessing the potential changes in the RPE layer. De Clerck et al.² and Huru et al.³ found significant thinning of macular thickness in patients with prediabetes and/or diabetes without diabetic retinopathy, but no further segmentation of the macular retina was performed. Therefore, their findings did not assess whether the altered thickness of the macular retina is at least partially attributable to the RPE. It's worth noting that these studies were based on small sample sizes and cross-sectional designs. Additionally, all patients had already been diagnosed with diabetes or prediabetes at baseline in these studies, and therefore these studies may not have had the ability to examine earlier changes before then.

Other studies have shown that changes in the retinal photoreceptor cell layer and

RPE are observed simultaneously in early diabetes. Enzsoly et al.⁴ found in STZ-induced Wistar diabetic mice that photoreceptor cells showed signs of outer segment degeneration, along with decreased RPE cell thickness, reduced cytoplasmic volume, reduced smooth endoplasmic reticulum, and decreased amount of isomerohydrolase enzyme (RPE65). Lu et al.⁵ showed a concomitant decrease in the number of photoreceptor cell nuclei layers and a shortening of RPE cell height in Long-Evans Tokushima spontaneously diabetic rats. These results suggest that changes of RPE are at least no later than that of photoreceptor cells in these diabetes models. In the current study, the average ELM-ISOS layer, which represents the photoreceptors, did exhibit a trend of thinning thickness in association with the risk of T2DM (HR = 0.942, 95%CI: 0.850, 1.044, P = 0.257); however, it did not reach statistical significance. Our hypothesis is that alterations in the RPE may precede that of the photoreceptor layer. According to your suggestion, we have added this point in the revised manuscript. An extended discussion is available in ***Supplementary Discussion***. Hopefully our discussion become more balanced and reasonable.

In contrast to the previous studies with small sample sizes and cross-sectional designs, our study is based on an extensive follow-up for over a decade of a large number of participants who did not have a diabetes diagnosis at baseline. By incorporating an extensive range of covariates, we reported on early changes in RPE thickness based on OCT measurements are indicative of future risk of T2DM. Additionally, the RPE-associated metabolites identified in this study significantly improved the discriminatory ability and clinical utility of T2DM, which further strengthens our confidence that this ocular biomarker may be indicative of diabetes-related biological changes. Hopefully this addresses your concerns.

Reference

1. Şahin M, Şahin A, Kılınç F, Karaalp Ü, Yüksel H, Özkurt ZG, Türkcü FM, Çaça İ. Early detection of macular and peripapillary changes with spectralis optical coherence tomography in patients with prediabetes. Arch Physiol Biochem. 2018

- Feb;124(1):75-79. doi: 10.1080/13813455.2017.1361450. PMID: 28780883.
2. De Clerck EEB, Schouten JSAG, Berendschot TTJM, Goezinne F, Dagnelie PC, Schaper NC, Schram MT, Stehouwer CDA, Webers CAB. Macular thinning in prediabetes or type 2 diabetes without diabetic retinopathy: the Maastricht Study. *Acta Ophthalmol.* 2018 Mar;96(2):174-182. doi: 10.1111/aos.13570. Epub 2017 Nov 1. PMID: 29090852.
 3. Huru J, Leiviskä I, Saarela V, Liinamaa MJ. Prediabetes influences the structure of the macula: thinning of the macula in the Northern Finland Birth Cohort. *Br J Ophthalmol.* 2021 Dec;105(12):1731-1737. doi: 10.1136/bjophthalmol-2020-317414. Epub 2020 Oct 7. PMID: 33028576.
 4. Énzöly A, Szabó A, Kántor O, Dávid C, Szalay P, Szabó K, Szél Á, Németh J, Lukáts Á. Pathologic alterations of the outer retina in streptozotocin-induced diabetes. *Invest Ophthalmol Vis Sci.* 2014 May 20;55(6):3686-99. doi: 10.1167/iovs.13-13562. PMID: 24845643.
 5. Lu ZY, Bhutto IA, Amemiya T. Retinal changes in Otsuka long-evans Tokushima Fatty rats (spontaneously diabetic rat) --possibility of a new experimental model for diabetic retinopathy. *Jpn J Ophthalmol.* 2003 Jan-Feb;47(1):28-35. doi: 10.1016/s0021-5155(02)00631-7. PMID: 12586175.

Supplementary Table S13

Multivariable Cox models for evaluating associations between various retinal layer thickness and risk of incident T2DM in the UKB cohort.

Risk of T2DM *†	HR	95% CI	P ‡	
Overall retinal thickness	0.926	0.835	1.027	0.142
Outer retinal thickness				
RPE	0.856	0.768	0.954	0.005
ELM-ISOS	0.945	0.852	1.047	0.276
Inner retinal thickness				
RNFL	1.032	0.931	1.144	0.548
GC-IPL	0.978	0.884	1.082	0.668

* Adjusted for age, sex, ethnicity, household income, Townsend deprivation index, education, smoking, drinking, body-mass index, and use of antihypertensive and lipid-lowering medication.

† Per-SD change of various retinal thickness, measured by SD-OCT centered at the foveal.

‡ Bold indicates statistically significant.

T2DM = type 2 diabetes mellitus; HR = hazard ratio; CI = confidence interval; RPE = retinal pigment epithelium; ELM-ISOS = external limiting membrane-inner segment outer segment; RNFL = retinal nerve fiber layer; GC-IPL = ganglion cell-inner plexiform layer; SD-OCT = spectral-domain optical coherence tomography.

Comment #4. Introduction: L. 87 - The authors did not cite the first papers showing that outer retinal barrier and RPE alterations are the first abnormalities in types 2 diabetes models [2013. doi: 10.1016/ and j.ajpath.2011.04.018. Epub 2011 Jun 2]. These papers should be included since they cite RPE observations on type 2 diabetic models.

Response: Thank you for your valuable suggestions. We have incorporated this paper into the revised manuscript (**Page 4, Line 82**).

Comment #5. Introduction: L. 91-L.94 - The authors should moderate their statement here. They do not show any sign of RPE alteration in the present study. They show a reduction in thickness measured by OCT, which cannot be correlated so far to any functional consequence.

Response: Thank you for bringing this issue to our attention. The conclusions relating to RPE-BM complex thickness rely on interpretation of OCT imaging. In the revised manuscript, we have emphasized that RPET changes were inferred by optical reflectivity changes. If optical reflectivity changes occur within the RPE cell for reasons other than shortening or loss of cells, then there could be decoupling between optical reflectivity changes and true thickness change within the RPE-BM complex. The measurement changes of RPE-BM complex may reflect changes in structural, morphologic, and density indices, which is currently challenging to establish any direct relationship with function.¹ We have moderated our statements throughout the text and have also highlighted this limitation in the **Discussion** section (**Page 13, Line 350**):

[Discussion section, Page 13, Line 350-356]

“Third, our results showed that a decreased RPET based on in vivo OCT imaging was associated with diabetes and its organ damages; however, the thickness was inferred by the hyperreflective band situated on the outer layer of the retina. Consequently, it is important to acknowledge that there could be decoupling between optical

reflectivity changes and true thickness changes under certain circumstances, making it difficult to establish any direct relationship with function so far.”

Comment #6. Methods: Populations - Could the authors specify the inclusion/exclusion criteria in the different populations? This information is crucial to understand the study design and should not require to consult another reference.

Response: We apologize for the lack of adequate information regarding the eligibility criteria. Detailed exclusion criteria of the study have been added to **Supplementary Methods: Eligibility criteria**, read as:

“For the phase-I analysis, only participants who underwent eligible retinal SD-OCT measurements and NMR metabolomic profiling at baseline assessment were included. The eligibility criteria involved the exclusion of participants without NMR metabolomic profiling (391,763 participants) and OCT scanning (96,068 participants), as well as participants with missing thickness values (294 right eyes and 407 left eyes), low signal strength (1,272 right eyes and 1,067 left eyes), poor centration or segmentation (4,291 right eyes and 4,309 left eyes), high refractive error (spherical equivalent [SE] > 6 or < -6 diopters [D]) (341 right eyes and 404 left eyes), visual impairment (> 0.1 logarithm of the minimum angle of resolution [log MAR]) (2,054 right eyes and 2,014 left eyes), or abnormal intraocular pressure (IOP) (≥ 22 or ≤ 5 mmHg) (701 right eyes and 752 left eyes). Furthermore, patients with glaucoma (82 participants), other retinal disorders (162 participants), and neurodegenerative diseases (39 participants) were excluded, owing to the potential for secondary retinal destruction. Ultimately, the phase-I analysis included a total of 7,824 participants, whose medical history was obtained via questionnaires, interviews, and inpatient diagnoses prior to baseline.

Phase-II analysis included 110,730 participants who completed metabolomic profiling at baseline assessment, with the exclusion of participants lacking hospital records

(18,032 participants). In addition, participants who were already included in phase-I analysis (6,684 participants) were further excluded. Participants with baseline T2DM were also excluded (1,790 participants) prior to all analyses. Hence, the participants included in phase-II analysis and those included in phase-I analysis represent two distinct subsets that do not overlap. This culminated in a total of 84,224 participants for phase-II analysis."

We accepted your suggestions, and have incorporated these data into the revised Method section accordingly. Hopefully this addresses your concerns.

Comment #7. Methods: There are insufficient details in the method used to quantify RPET using SD-OCT. It is not clear where RPE thickness was measured (which subfields?) and how it was measured. Did the author measure the RPE and Bruch membrane complex or only the RPE? Did they include the interdigitations in the RPE measurement? How could the author ascertain that retinal thinning and pigments have no influence on the RPE reflectivity? The other retinal layer thickness should be included in the dataset since the data are available.

Response: We apologize for the insufficient details of the methodology. The measurements of RPE-Bruch's membrane (BM) complex were acquired across nine ETDRS subfields. The RPE-BM complex was delineated using the TABS algorithm, which identifies this complex on SD-OCT images as a thick hyperreflective band located on the outer aspect of the retina.¹ The algorithm differentiates between the inner and outer boundaries of this band, with the thickness of the RPE-BM complex corresponding to the separation between these two lines.² Specifically, the inner boundary corresponds to the photoreceptor outer segment-RPE boundary, while the outer boundary corresponds to the BM-choroid boundary. This information has been added to the **Methods** section (**Page 15, Line 420**) and **Supplementary Methods: Acquisition, analysis, and quality control processes for SD-OCT imaging in the UKB cohort**.

The thinning of other layers of the retina as well as pigments does have potential impacts on the thickness measurement of the RPE. To analyze these effects, we included measurements of other layers of the retina in the revised manuscript for additional analyses (**Supplementary Table S13**). For your convenience, we have provided a summary of the results regarding the associations between other retinal layers and incident T2DM below. Among overall average retinal thickness, average RPE thickness, average ELM-ISOS thickness, average GC-IPL thickness, and average RNFL thickness, only average RPE thickness demonstrated significantly association with the risk of future T2DM, after adjusted for age, sex, ethnicity, household income, Townsend deprivation index, education, smoking, drinking, body-mass index, and use of antihypertensive and lipid-lowering medication (all of which reached a threshold of $P < 0.1$ in the univariate models). This is in line with our hypothesis that alterations in the RPE may precede that of other retinal layer in early diabetes.

In order to make more conservative and robust conclusions, we conducted a sensitivity analysis to mitigate the potential impact of differences in pigmentation on RPET measurements across different ethnic groups. We adjusted for more detailed ethnicity classifications, including further subdivisions for White, South Asian, East Asian, Black, and Mixed races. This approach allowed us to better control for the confounding due to pigmentation levels across different racial groups in this study of multi-ethnic population. Additionally, we conducted several other sensitivity analyses based on suggestions from the reviewers. This information has been added to the **Methods** section (**Page 18, Line 502**). Importantly, these sensitivity analyses yielded same results to the main analysis. (**Supplementary Table S3**). For your convenience, we have provided a summary of the results below. Hopefully this addresses your concerns.

Reference

1. Yang Q, Reisman CA, Wang Z, Fukuma Y, Hangai M, Yoshimura N, Tomidokoro A, Araie M, Raza AS, Hood DC, Chan K. Automated layer segmentation of macular OCT images using dual-scale gradient information. *Opt Express*. 2010 Sep 27;18(20):21293-307. doi: 10.1364/OE.18.021293. PMID: 20941025; PMCID: PMC3101081.
2. Staurengi G, Sadda S, Chakravarthy U, Spaide RF; International Nomenclature for Optical Coherence Tomography (IN•OCT) Panel. Proposed lexicon for anatomic landmarks in normal posterior segment spectral-domain optical coherence tomography: the IN•OCT consensus. *Ophthalmology*. 2014 Aug;121(8):1572-8. doi: 10.1016/j.ophtha.2014.02.023. Epub 2014 Apr 19. PMID: 24755005.

Supplementary Table S13

Multivariable Cox models for evaluating associations between various retinal layer thickness and risk of incident T2DM in the UKB cohort.

Risk of T2DM *†	HR	95% CI	P ‡	
Overall retinal thickness	0.926	0.835	1.027	0.142
Outer retinal thickness				
RPE	0.856	0.768	0.954	0.005
ELM-ISOS	0.945	0.852	1.047	0.276
Inner retinal thickness				
RNFL	1.032	0.931	1.144	0.548
GC-IPL	0.978	0.884	1.082	0.668

* Adjusted for age, sex, ethnicity, household income, Townsend deprivation index, education, smoking, drinking, body-mass index, and use of antihypertensive and lipid-lowering medication.

† Per-SD change of various retinal thickness, measured by SD-OCT centered at the foveal.

‡ Bold indicates statistically significant.

T2DM = type 2 diabetes mellitus; HR = hazard ratio; CI = confidence interval; RPE = retinal pigment epithelium; ELM-ISOS = external limiting membrane-inner segment outer segment; RNFL = retinal nerve fiber layer; GC-IPL = ganglion cell-inner plexiform layer; SD-OCT = spectral-domain optical coherence tomography.

Supplementary Table S3

Sensitivity analyses on the association of average RPET with T2DM risk in UKB cohort.

Risk of T2DM *	HR	95% CI	P value †
Overall population-I			
Model 1 ‡	0.856	0.767-0.954	0.005
Model 2 §	0.868	0.779-0.968	0.011
Model 3 ¶	0.849	0.757-0.951	0.005
Model 4 #	0.867	0.775-0.970	0.013
Model 5 **	0.845	0.735-0.973	0.019
Stratified by PRS risk of AMD			
Quartile 2 to Quartile 4 (Participants with low-moderate AMD risk)			
Model 1 ‡	0.835	0.734-0.950	0.006
Model 2 §	0.847	0.744-0.963	0.011
Model 3 ¶	0.833	0.726-0.956	0.009
Model 4 #	0.833	0.727-0.954	0.009
Model 5 **	0.792	0.666-0.941	0.008
Quartile 1 (Participants with highest AMD risk)			
Model 1 ‡	0.900	0.731-1.107	0.319
Model 2 §	0.921	0.747-1.137	0.446
Model 3 ¶	0.907	0.723-1.138	0.400
Model 4 #	0.940	0.762-1.161	0.568
Model 5 **	0.969	0.757-1.241	0.802

* Per-SD change of RPET.

† Bold indicates significant.

‡ Adjusted for age, sex, ethnicity, Townsend deprivation index, household income, education, smoking, drinking, body mass index, and use of antihypertensive and lipid-lowering medication.

§ Further adjusted for more comprehensive ethnicity classifications with further subdivisions for South Asian, East Asian, Black, and Mixed races.

¶ Further adjusted for biological ages.¹

Further adjusted for frailty score.²

** Further excluded cancer, cardiovascular diseases, and renal diseases.

RPET= retinal pigment epithelium thickness; T2DM= type 2 diabetes mellitus; HR= hazard ratio; CI= confidence interval; SD= standard deviation.

Reference

1. Yang G, Cao X, Li X, Zhang J, Ma C, Zhang N, Lu Q, Crimmins EM, Gill TM, Chen X, Liu Z. Association of Unhealthy Lifestyle and Childhood Adversity With Acceleration of Aging Among UK Biobank Participants. *JAMA Netw Open*. 2022 Sep 1;5(9): e2230690. doi: 10.1001/jamanetworkopen.2022.30690. PMID: 36066889; PMCID: PMC9449787.
2. Fried LP, Tangen CM, Walston J, Newman AB, Hirsch C, Gottdiener J, Seeman T, Tracy R, Kop WJ, Burke G, McBurnie MA; Cardiovascular Health Study Collaborative Research Group. Frailty in older adults: evidence for a phenotype. *J Gerontol A Biol Sci Med Sci*. 2001 Mar;56(3):M146-56. doi: 10.1093/gerona/56.3.m146. PMID: 11253156.

Comment #8. Methods: Several papers showed that the subfoveal RPE/ Bruch membrane complex thickness decreases with aging and hypertension. These papers must be cited and their methodology must be compared to the methodology used in the present paper [doi: 10.1177/11206721221101372. Epub 2022 May, doi: 10.1038/s41433-021-01911-5. Epub 2022 Jan 20.]. Since diabetes can be considered as a premature aging, it could be interesting to compare metabolomics associated with RPET reduction during aging and in diabetes to ensure specificity to diabetes.

Response: Thank you for your valuable suggestions. We have systematically searched and reviewed the relevant studies, and have cited these papers in the revised manuscript (**Page 15, Line 422**). A detailed comparison of the methodology among these studies and the current study is added to **Supplementary Discussion: Comparison of RPE measurements with two previous studies**. Since diabetes can be considered as a premature aging, we calculated the biological age and frailty score of the UKB participants using established methods and performed a series of sensitivity analyses. The results of all sensitivity analyses were consistent with the primary results. In addition, we compared metabolomics associated with RPET reduction during aging and in diabetes to ensure their specificity to diabetes.

(1) Comparison of methodology with two previous studies

The definition of RPE thickness used in these two studies is the same as that of the present study, i.e., from the inner boundary of the RPE to the outer boundary of the BM. The main differences between the present study and these two studies were the OCT instrument used, the retinal layer segmentation method (manual segmentation was used in these two studies), and the subfield measured.

Kadhim et al.¹ used RTVue SD-OCT, Shao et al.² used Heidelberg SD-OCT, while Topcon SD-OCT was used in this UKB study. In both studies, manual segmentation of the retina was used to obtain RPE thickness measurements. Kadhim et al.¹ also compared manual segmentation with automated segmentation and found no significant differences among these two techniques. However, it was almost

impossible to manually segment the retina in the current study with massive images at a biobank scale. As a result, an algorithm was used to automatically segmented the retinal layers and delineated RPE-Bruch's membrane complex in this study.

Furthermore, the measured area differed among these two studies and the current study. Kadhim et al.¹ measured the thickness of four areas: (1) fovea minimum, the thinnest foveal thickness at the center point of fovea; (2) central subfield, defined by 500 μm radius from fovea minimum; (3) inner macula, defined by 500–1500 μm radius from fovea minimum; and (4) outer macula, defined by 1500–3000 μm radius from fovea minimum. Shao et al.² measured the RPE-BM thickness at five locations: in the fovea, and in the outer extreme section superior to the fovea, inferior to the fovea, temporal of the fovea and 2 mm nasal to the fovea in direction to the optic disc. In this study, average RPE thickness measurements were obtained from nine ETDRS subfields, including a 1-mm diameter circle and two rings with diameters of 3 mm and 6 mm, respectively. These rings were subdivided into superior, inferior, nasal, and temporal subregions, providing a detailed analysis of RPE thickness across the macula. This approach allowed us to gain a better understanding of regional variations on associations between RPE thickness and T2DM risk (**Supplementary Table S2**). For your convenience, we have provided the summary result table below. The comparison of methodology used in these papers and the current study is added to **Supplementary Discussion: Comparison of RPE measurements with two previous studies**.

(2) Comparison of RPET metabolic fingerprints between T2DM and ageing

We strongly agree that diabetes is also a form of premature aging³. In order to exclude the potential bias due to accelerated aging, we have conducted a more comprehensive investigation into the relationship between RPET and incident T2DM, as well as the association between RPET metabolomics and the aging/diabetes process. Firstly, we collected additional data and calculated biological age⁴ and frailty scores⁵ of the UKB population using established and well-known methods. We then

performed sensitivity analyses in phase-I analysis by (1) further adjusting for biological age and (2) further adjusting for frailty score. Several other sensitivity analyses were also conducted and all of them produced similar and robust results (**Supplementary Table S3**, as provided below for your convenience). These findings suggest that the independent association between RPET and incident T2DM is not influenced by confounding due to aging.

Next, we leveraged biological ages to identify metabolomics that are associated with RPET decline during aging (**Methods** section, **Page 19, Line 523**). We included pertinent variables in our multivariate regression model, such as gender, race, household income, Townsend deprivation index, education, smoking, alcohol consumption, BMI, hypertension, hyperlipidemia, and the use of antihypertensive and lipid-lowering drugs. All these variables were statistically significant in univariate linear regression models. After multiple testing correction, a total of fifty-four RPET-associated metabolic biomarkers were found to be significantly associated with biological age (**Supplementary Table S7**, as provided below for your convenience). Several aging-associated biomarkers overlapped with T2DM-associated ones, which is to be expected given that diabetes is a form of accelerated aging³. Additionally, thirty-one metabolic biomarkers that were associated with T2DM did not show any significant association or displayed an opposite association with aging. This suggests that these biomarkers may be specific to diabetes independent of ageing. Our sensitivity analysis confirmed these results, as constructing an RPET metabolic state model using these fingerprints independent of aging resulted in similar enhancements in predictability and clinical utility for T2DM (**Supplementary Figure S3**). For your convenience, the summary result figure is provided below. The information has been added to **Supplementary Discussion: Comparison of RPET metabolic fingerprints of T2DM and ageing**.

Hopefully this addresses your concerns.

Reference

1. Ghanem Kadhim Z, Mohammad NK. Effect of aging and lifestyle on healthy macular photoreceptors and retinal pigment epithelium-Bruch membrane complex thickness. *Eur J Ophthalmol*. 2023 Jan;33(1):441-447. doi: 10.1177/11206721221101372. PMID: 35585693.
2. Shao L, Zhang QL, Zhang C, Dong L, Zhou WD, Zhang RH, Wu HT, Wei WB. Thickness of retinal pigment epithelium-Bruch's membrane complex in adult Chinese using optical coherence tomography. *Eye (Lond)*. 2023 Jan;37(1):155-159. doi: 10.1038/s41433-021-01911-5. PMID: 35046547; PMCID: PMC9829656.
3. Palmer AK, Gustafson B, Kirkland JL, Smith U. Cellular senescence: at the nexus between ageing and diabetes. *Diabetologia*. 2019 Oct;62(10):1835-1841. doi: 10.1007/s00125-019-4934-x. PMID: 31451866; PMCID: PMC6731336.
4. Yang G, Cao X, Li X, Zhang J, Ma C, Zhang N, Lu Q, Crimmins EM, Gill TM, Chen X, Liu Z. Association of Unhealthy Lifestyle and Childhood Adversity With Acceleration of Aging Among UK Biobank Participants. *JAMA Netw Open*. 2022 Sep 1;5(9): e2230690. doi: 10.1001/jamanetworkopen.2022.30690. PMID: 36066889; PMCID: PMC9449787.
5. Fried LP, Tangen CM, Walston J, Newman AB, Hirsch C, Gottdiener J, Seeman T, Tracy R, Kop WJ, Burke G, McBurnie MA; Cardiovascular Health Study Collaborative Research Group. Frailty in older adults: evidence for a phenotype. *J Gerontol a Biol Sci Med Sci*. 2001 Mar;56(3):M146-56. doi: 10.1093/gerona/56.3.m146. PMID: 11253156.

Supplementary Table S2

Multivariable Cox models for evaluating association between RPET across all ETDRS subfields and T2DM risk in participants with low-moderate PRS risk of AMD.

Risk of T2DM *†	HR	95% CI	P ‡
Total average	0.741	0.609-0.902	0.003
Outer average	0.739	0.606-0.901	0.003
Outer superior	0.753	0.617-0.919	0.005
Outer nasal	0.799	0.660-0.968	0.022
Outer inferior	0.763	0.622-0.936	0.010
Outer temporal	0.761	0.620-0.934	0.009
Inner average	0.787	0.652-0.951	0.013
Inner superior	0.838	0.695-1.010	0.064
Inner nasal	0.808	0.674-0.970	0.022
Inner inferior	0.822	0.681-0.993	0.042
Inner temporal	0.778	0.645-0.938	0.009
Central field	0.800	0.666-0.962	0.018

* Adjusted for age, sex, ethnicity, household income, Townsend deprivation index, education, smoking, drinking, body-mass index, use of antihypertensive, and lipid-lowering medication.

† Per-SD change of RPET, measured by SD-OCT using the ETDRS nine-pane grid.

‡ Bold indicates significant.

RPET = retinal pigment epithelium thickness; ETSRS = Early Treatment Diabetic Retinopathy Study; T2DM = type 2 diabetes mellitus; PRS = polygenetic risk score; AMD = age-related macular degeneration; HR = hazard ratio; CI = confidence interval; SD = standard deviation; SD-OCT = spectral-domain optical coherence tomography.

Supplementary Table S3

Sensitivity analyses on the association of average RPET with T2DM risk in UKB cohort.

Risk of T2DM *	HR	95% CI	P value †
Overall population-I			
Model 1 ‡	0.856	0.767-0.954	0.005
Model 2 §	0.868	0.779-0.968	0.011
Model 3 ¶	0.849	0.757-0.951	0.005
Model 4 #	0.867	0.775-0.970	0.013
Model 5 **	0.845	0.735-0.973	0.019
Stratified by PRS risk of AMD			
Quartile 2 to Quartile 4 (Participants with low-moderate AMD risk)			
Model 1 ‡	0.835	0.734-0.950	0.006
Model 2 §	0.847	0.744-0.963	0.011
Model 3 ¶	0.833	0.726-0.956	0.009
Model 4 #	0.833	0.727-0.954	0.009
Model 5 **	0.792	0.666-0.941	0.008
Quartile 1 (Participants with highest AMD risk)			
Model 1 ‡	0.900	0.731-1.107	0.319
Model 2 §	0.921	0.747-1.137	0.446
Model 3 ¶	0.907	0.723-1.138	0.400
Model 4 #	0.940	0.762-1.161	0.568
Model 5 **	0.969	0.757-1.241	0.802

* Per-SD change of RPET.

† Bold indicates significant.

‡ Adjusted for age, sex, ethnicity, Townsend deprivation index, household income, education, smoking, drinking, body mass index, and use of antihypertensive and lipid-lowering medication.

§ Further adjusted for more comprehensive ethnicity classifications with further subdivisions for South Asian, East Asian, Black, and Mixed races.

¶ Further adjusted for biological ages.¹

Further adjusted for frailty score.²

** Further excluded cancer, cardiovascular diseases, and renal diseases.

RPET = retinal pigment epithelium thickness; T2DM = type 2 diabetes mellitus; HR = hazard ratio; CI = confidence interval; SD = standard deviation.

Reference

1. Yang G, Cao X, Li X, Zhang J, Ma C, Zhang N, Lu Q, Crimmins EM, Gill TM, Chen X, Liu Z. Association of Unhealthy Lifestyle and Childhood Adversity with Acceleration of Aging Among UK Biobank Participants. *JAMA Netw Open*. 2022 Sep 1;5(9): e2230690. doi: 10.1001/jamanetworkopen.2022.30690. PMID: 36066889; PMCID: PMC9449787.
2. Fried LP, Tangen CM, Walston J, Newman AB, Hirsch C, Gottdiener J, Seeman T, Tracy R, Kop WJ, Burke G, McBurnie MA; Cardiovascular Health Study Collaborative Research Group. Frailty in older adults: evidence for a phenotype. *J Gerontol a Biol Sci Med Sci*. 2001 Mar;56(3):M146-56. doi: 10.1093/gerona/56.3.m146. PMID: 11253156.

Supplementary Table S7

Multi-adjusted associations between the RPET-associated metabolites and ageing.

Metabolic biomarkers *	HR	95%CI	P _{raw}	P _{FDR}
VLDL Cholesterol	0.073	-0.006 0.152	7.11×10 ⁻²	8.27×10 ⁻²
LDL Cholesterol	-0.488	-0.570 -0.407	<1.0×10 ⁻⁸	<1.0×10 ⁻⁸
Total Triglycerides	0.123	0.052 0.194	6.34×10 ⁻⁴	8.82×10 ⁻⁴
Triglycerides in VLDL	-0.039	-0.110 0.031	0.276	0.294
Phospholipids in VLDL	0.055	-0.020 0.130	0.148	0.164
Phospholipids in LDL	-0.465	-0.548 -0.382	<1.0×10 ⁻⁸	<1.0×10 ⁻⁸
Cholesteryl Esters in VLDL	0.099	0.018 0.180	1.64×10 ⁻²	2.01×10 ⁻²
Cholesteryl Esters in LDL	-0.469	-0.550 -0.387	<1.0×10 ⁻⁸	<1.0×10 ⁻⁸
Free Cholesterol in VLDL	0.011	-0.066 0.087	0.780	0.781
Total Lipids in VLDL	0.041	-0.032 0.114	0.273	0.294
Total Lipids in LDL	-0.411	-0.493 -0.330	<1.0×10 ⁻⁸	<1.0×10 ⁻⁸
Concentration of VLDL Particles	0.210	0.134 0.286	7.27×10 ⁻⁸	1.79×10 ⁻⁷
Concentration of LDL Particles	-0.215	-0.298 -0.132	4.30×10 ⁻⁷	8.80×10 ⁻⁷
Average Diameter for VLDL Particles	-0.354	-0.425 -0.284	<1.0×10 ⁻⁸	<1.0×10 ⁻⁸
Apolipoprotein B	-0.088	-0.172 -0.004	3.98×10 ⁻²	4.71×10 ⁻²
Linoleic Acid	-0.064	-0.141 0.012	9.80×10 ⁻²	1.12×10 ⁻¹
Isoleucine	0.524	0.459 0.589	<1.0×10 ⁻⁸	<1.0×10 ⁻⁸
Leucine	0.088	0.022 0.155	9.07×10 ⁻³	1.16×10 ⁻²
Tyrosine	1.024	0.960 1.087	<1.0×10 ⁻⁸	<1.0×10 ⁻⁸
Concentration of Very Large VLDL Particles	-0.164	-0.235 -0.093	6.34×10 ⁻⁶	1.13×10 ⁻⁵
Total Lipids in Very Large VLDL	-0.139	-0.211 -0.068	1.27×10 ⁻⁴	1.85×10 ⁻⁴
Phospholipids in Very Large VLDL	-0.141	-0.212 -0.069	1.12×10 ⁻⁴	1.69×10 ⁻⁴
Cholesterol in Very Large VLDL	-0.195	-0.269 -0.122	2.06×10 ⁻⁷	4.55×10 ⁻⁷
Cholesteryl Esters in Very Large VLDL	-0.261	-0.337 -0.186	<1.0×10 ⁻⁸	<1.0×10 ⁻⁸
Free Cholesterol in Very Large VLDL	-0.135	-0.207 -0.063	2.31×10 ⁻⁴	3.28×10 ⁻⁴
Triglycerides in Very Large VLDL	-0.141	-0.212 -0.071	7.68×10 ⁻⁵	1.26×10 ⁻⁴
Concentration of Large VLDL Particles	-0.184	-0.256 -0.113	4.40×10 ⁻⁷	8.80×10 ⁻⁷
Total Lipids in Large VLDL	-0.198	-0.270 -0.126	6.53×10 ⁻⁸	1.67×10 ⁻⁷
Phospholipids in Large VLDL	-0.189	-0.261 -0.117	2.78×10 ⁻⁷	5.93×10 ⁻⁷
Cholesterol in Large VLDL	-0.169	-0.243 -0.096	6.81×10 ⁻⁶	1.18×10 ⁻⁵
Cholesteryl Esters in Large VLDL	-0.117	-0.192 -0.043	2.10×10 ⁻³	2.80×10 ⁻³
Free Cholesterol in Large VLDL	-0.222	-0.295 -0.149	<1.0×10 ⁻⁸	<1.0×10 ⁻⁸
Triglycerides in Large VLDL	-0.241	-0.311 -0.171	<1.0×10 ⁻⁸	<1.0×10 ⁻⁸
Concentration of Medium VLDL Particles	-0.235	-0.314 -0.156	<1.0×10 ⁻⁸	1.47×10 ⁻⁸
Total Lipids in Medium VLDL	-0.187	-0.264 -0.111	1.55×10 ⁻⁶	2.93×10 ⁻⁶
Phospholipids in Medium VLDL	-0.163	-0.243 -0.084	5.88×10 ⁻⁵	9.90×10 ⁻⁵

Cholesterol in Medium VLDL	-0.167	-0.251	-0.084	9.05×10 ⁻⁵	1.41×10 ⁻⁴
Free Cholesterol in Medium VLDL	-0.115	-0.196	-0.033	5.67×10 ⁻³	7.41×10 ⁻³
Triglycerides in Medium VLDL	-0.192	-0.263	-0.121	1.27×10 ⁻⁷	3.01×10 ⁻⁷
Concentration of Small VLDL Particles	0.121	0.047	0.194	1.30×10 ⁻³	1.77×10 ⁻³
Total Lipids in Small VLDL	0.146	0.072	0.221	1.13×10 ⁻⁴	1.69×10 ⁻⁴
Phospholipids in Small VLDL	0.014	-0.065	0.092	0.730	0.753
Cholesterol in Small VLDL	0.037	-0.042	0.116	0.359	0.376
Cholesteryl Esters in Small VLDL	0.097	0.019	0.174	1.45×10 ⁻²	1.83×10 ⁻²
Free Cholesterol in Small VLDL	-0.086	-0.167	-0.005	3.78×10 ⁻²	4.56×10 ⁻²
Triglycerides in Small VLDL	0.218	0.149	0.287	<1.0×10 ⁻⁸	<1.0×10 ⁻⁸
Concentration of Large LDL Particles	-0.203	-0.286	-0.120	1.60×10 ⁻⁶	2.93×10 ⁻⁶
Concentration of Medium LDL Particles	-0.270	-0.351	-0.190	<1.0×10 ⁻⁸	<1.0×10 ⁻⁸
Total Lipids in Medium LDL	-0.562	-0.642	-0.482	<1.0×10 ⁻⁸	<1.0×10 ⁻⁸
Phospholipids in Medium LDL	-0.659	-0.739	-0.578	<1.0×10 ⁻⁸	<1.0×10 ⁻⁸
Cholesterol in Medium LDL	-0.606	-0.687	-0.526	<1.0×10 ⁻⁸	<1.0×10 ⁻⁸
Cholesteryl Esters in Medium LDL	-0.558	-0.638	-0.478	<1.0×10 ⁻⁸	<1.0×10 ⁻⁸
Concentration of Small LDL Particles	-0.067	-0.149	0.015	0.107	0.121
Total Lipids in Small LDL	-0.406	-0.488	-0.324	<1.0×10 ⁻⁸	<1.0×10 ⁻⁸
Phospholipids in Small LDL	-0.161	-0.242	-0.081	8.44×10 ⁻⁵	1.35×10 ⁻⁴
Cholesterol in Small LDL	-0.571	-0.653	-0.489	<1.0×10 ⁻⁸	<1.0×10 ⁻⁸
Cholesteryl Esters in Small LDL	-0.536	-0.618	-0.454	<1.0×10 ⁻⁸	<1.0×10 ⁻⁸
Free Cholesterol in Small LDL	-0.538	-0.615	-0.460	<1.0×10 ⁻⁸	<1.0×10 ⁻⁸
Triglycerides in Small LDL	0.201	0.132	0.271	1.20×10 ⁻⁸	3.20×10 ⁻⁸
Ratio of triglycerides to phosphoglycerides	0.010	-0.062	0.083	0.781	0.781
Ratio of apolipoprotein B to apolipoprotein A1	-0.183	-0.257	-0.108	1.59×10 ⁻⁶	2.93×10 ⁻⁶
Cholesteryl esters to total lipids ratio in large LDL	-0.629	-0.696	-0.562	<1.0×10 ⁻⁸	<1.0×10 ⁻⁸
Cholesteryl esters to total lipids ratio in medium LDL	-0.180	-0.247	-0.112	2.00×10 ⁻⁷	4.55×10 ⁻⁷
Phospholipids to total lipids ratio in small HDL	0.974	0.907	1.040	<1.0×10 ⁻⁸	<1.0×10 ⁻⁸

* Adjusted for age, sex, ethnicity, household income, Townsend deprivation index, education, smoking, drinking, BMI, use of lipid-lowering medications, and antihypertensive medications.

HR = hazard ratio; CI = confident interval; VLDL = very low-density lipoprotein; LDL = low-density lipoprotein; HDL = high-density lipoprotein.

Supplementary Figure S2

Sensitivity analysis of constructing metabolic state model and combined model incorporating only metabolic biomarkers that were independent of ageing.

Comment #9. Methods: L. 328 - Is there information of cardiovascular/ kidney complication at the time of diabetes diagnosis?

Response: Thank you for bringing this important issue to our attention. We performed several additional sensitivity analyses and new experiments to (1) assess the association of RPET with incident T2DM after excluding pre-existing diseases at baseline diabetes diagnosis; and (2) investigate the predictive value and clinical utility of RPET-associated metabolic fingerprints for diabetic complications.

Firstly, we acquired information on participants' diseases at baseline through a combination of inpatient records, touchscreen questionnaires, and verbal interviews. After excluding participants with cardiovascular diseases, kidney diseases, and cancer, we found that the association between RPE thickness and future T2DM remained unchanged (**Supplementary Table S3**). We also conducted several other sensitivity analyses, which are summarized in the table provided below for your convenience.

We also collected information on cardiovascular and kidney complications at or after the diagnosis of incident T2DM.¹ To further evaluate the predictive value and clinical utility of RPET-associated metabolites for diabetic complications, we performed additional receiver operating characteristic analyses and decision curve analyses using cardiovascular complications and kidney complications as follow-up outcomes. Our results indicated that the incorporation of these metabolites significantly improved the performance of the clinical indicators-based models (see **Supplementary Figure S4**, as provided below for your convenience).

Furthermore, we conducted additional experiments using higher resolution SS-OCT² and higher sensitivity LC/MS metabolomics assay³ in a southern Chinese cohort⁴. Detailed information is available in **Supplementary Methods** section. Our results provided further evidence supporting the association between RPET and diabetic microvascular phenotypes, as well as the role of RPET metabolic fingerprints in facilitating risk stratification for diabetic target organ damages (see **Supplementary**

Figure S3, as provided below for your convenience).

Hopefully this addresses your concerns.

Reference

1. Kim DH, Jensen A, Jones K, Raghavan S, Phillips LS, Hung A, Sun YV, Li G, Reaven P, Zhou H, Zhou JJ. A platform for phenotyping disease progression and associated longitudinal risk factors in large-scale EHRs, with application to incident diabetes complications in the UK Biobank. *JAMIA Open*. 2023 Feb 9;6(1): ooad006. doi: 10.1093/jamiaopen/oad006. PMID: 36789288; PMCID: PMC9912368.
2. Laíns I, Wang JC, Cui Y, Katz R, Vingopoulos F, Staurengi G, Vavvas DG, Miller JW, Miller JB. Retinal applications of swept source optical coherence tomography (OCT) and optical coherence tomography angiography (OCTA). *Prog Retin Eye Res*. 2021 Sep; 84:100951. doi: 10.1016/j.preteyeres.2021.100951. Epub 2021 Jan 28. PMID: 33516833.
3. Chen W, Gong L, Guo Z, Wang W, Zhang H, Liu X, Yu S, Xiong L, Luo J. A novel integrated method for large-scale detection, identification, and quantification of widely targeted metabolites: application in the study of rice metabolomics. *Mol Plant*. 2013 Nov;6(6):1769-80. doi: 10.1093/mp/sst080. Epub 2013 May 23. PMID: 23702596.
4. Zhang S, Chen Y, Wang L, Li Y, Tang X, Liang X, He M, Wenyong H, Wang W; GDES group. Design and Baseline Data of the Diabetes Registration Study: Guangzhou Diabetic Eye Study. *Curr Eye Res*. 2023 Feb 27:1-9. doi: 10.1080/02713683.2023.2182745. Epub ahead of print. PMID: 36803011.

Supplementary Table S3

Sensitivity analyses on the association of average RPET with T2DM risk in UKB cohort.

Risk of T2DM *	HR	95% CI	P value †
Overall population-I			
Model 1 ‡	0.856	0.767-0.954	0.005
Model 2 §	0.868	0.779-0.968	0.011
Model 3 ¶	0.849	0.757-0.951	0.005
Model 4 #	0.867	0.775-0.970	0.013
Model 5 **	0.845	0.735-0.973	0.019
Stratified by PRS risk of AMD			
Quartile 2 to Quartile 4 (Participants with low-moderate AMD risk)			
Model 1 ‡	0.835	0.734-0.950	0.006
Model 2 §	0.847	0.744-0.963	0.011
Model 3 ¶	0.833	0.726-0.956	0.009
Model 4 #	0.833	0.727-0.954	0.009
Model 5 **	0.792	0.666-0.941	0.008
Quartile 1 (Participants with highest AMD risk)			
Model 1 ‡	0.900	0.731-1.107	0.319
Model 2 §	0.921	0.747-1.137	0.446
Model 3 ¶	0.907	0.723-1.138	0.400
Model 4 #	0.940	0.762-1.161	0.568
Model 5 **	0.969	0.757-1.241	0.802

* Per-SD change of RPET.

† Bold indicates significant.

‡ Adjusted for age, sex, ethnicity, Townsend deprivation index, household income, education, smoking, drinking, body mass index, and use of antihypertensive and lipid-lowering medication.

§ Further adjusted for more comprehensive ethnicity classifications with further subdivisions for South Asian, East Asian, Black, and Mixed races.

¶ Further adjusted for biological ages.¹

Further adjusted for frailty score.²

** Further excluded cancer, cardiovascular diseases, and renal diseases.

RPET = retinal pigment epithelium thickness; T2DM = type 2 diabetes mellitus; HR= hazard ratio; CI = confidence interval; SD = standard deviation.

Reference

1. Yang G, Cao X, Li X, Zhang J, Ma C, Zhang N, Lu Q, Crimmins EM, Gill TM, Chen X, Liu Z. Association of Unhealthy Lifestyle and Childhood Adversity with Acceleration of Aging Among UK Biobank Participants. *JAMA Netw Open*. 2022 Sep 1;5(9): e2230690. doi: 10.1001/jamanetworkopen.2022.30690. PMID: 36066889; PMCID: PMC9449787.
2. Fried LP, Tangen CM, Walston J, Newman AB, Hirsch C, Gottdiener J, Seeman T, Tracy R, Kop WJ, Burke G, McBurnie MA; Cardiovascular Health Study Collaborative Research Group. Frailty in older adults: evidence for a phenotype. *J Gerontol a Biol Sci Med Sci*. 2001 Mar;56(3):M146-56. doi: 10.1093/gerona/56.3.m146. PMID: 11253156.

Supplementary Figure S4

Discriminative improvements of incorporating RPET metabolic fingerprints for stratifying cardiovascular and kidney complications in UKB.

Supplementary Figure S3

Added discriminative power and clinical utility of RPET metabolic fingerprints for stratifying T2DM adverse microvascular phenotypes in GDES.

Comment #10. Results: The authors should include the thickness of other retinal layers (particularly the ganglion cell and nerve fiber layers) and compare their relevance to the RPET. How can they ascertain that RPET is a better biomarker for early diabetes detection as compared to other retinal layer thickness, that can be measured with more precision. Reduction in ganglion cell layer/ Nerve fiber layer thickness was shown to be associated with diabetes [doi:

10.1097/OPX.0000000000001650., doi: 10.1016/j.oftale.2022.02.009, DOI:

10.1159/000444498, doi: 10.1038/s41433-020-1020-z].

Response: Thank you for this valuable suggestion. We have included the thickness of other retinal layers for additional analyses in the revised manuscript (**Supplementary Table S13**). A summarized result table are provided below for your convenience. Out of the overall average retinal thickness, average RPE thickness, average ELM-ISOS thickness, average GC-IPL thickness, and average RNFL thickness, only the average RPE thickness demonstrated a significantly association with the risk of future T2DM, after adjusted for age, sex, ethnicity, household income, Townsend deprivation index, education, smoking, drinking, body-mass index, and use of antihypertensive and lipid-lowering medication (all of which reached a threshold of $P < 0.1$ in the univariate models). This is in line with our hypothesis that alterations in the *in vivo* measurements of RPE may precede that of other retinal layer in early diabetes.

Previous studies and reviews have evaluated the relationship between retinal thickness and diabetes and/or diabetic retinopathy, as you have kindly mentioned in your comments.¹⁻⁴ However, these studies mainly focused on RNFL and/or GCL (or GCC complex) in patients who already had diabetes and/or diabetic retinopathy. In this study, on the other hand, RPE thickness measurements were obtained from a population without diabetes, with baseline diabetes cases excluded from all analyses. Therefore, the above findings in the current study do not contradict the previously observed alterations in the RNFL and/or GCL, as the current study was conducted years before the onset of diabetes. RNFL and/or GCL changes are very likely to occur after this timepoint. In fact, these reviews also noted that some studies found no

significant difference in RNFL thickness between diabetic patients and healthy individuals.⁵ Thus, even the changes in these layers after diabetes onset remain inconclusive.

Our results suggest that reduced RPE thickness measurements significantly increase the risk of future incident diabetes, whereas other retinal layers did not show significant associations with the disease. These findings align with studies on animal and human studies that have noticed early changes in the outer retina. Hammoum et al.⁶ found that in the Meriones shawi diabetes model, the outer retina (including RPE and photoreceptors) was significantly impaired at three months, while changes in ganglion cells were not significantly different compared to controls. Enzoly et al.⁷ also found early structural and functional photoreceptor and RPE cell damage in streptozotocin-induced diabetic rats, when no apparent changes in overall retinal thickness were observed. In addition, Karaca et al.⁸ found that while detectable thinning of several retinal layers had occurred in patients with metabolic syndrome (a form of prediabetes), the overall thinning of the macular retina was primarily due to thinning of the photoreceptor layer. The authors defined the thickness of this layer as extending from the outer limiting membrane to BM in their study, which also includes the thickness of the RPE-BM complex. These studies all show the importance of thinning of the outer retinal layer in the early stages of diabetes. The ISOS layer, which represents the photoreceptors, did exhibit a trend of thinning thickness in association with the risk of T2DM (HR = 0.942, 95%CI: 0.850, 1.044, P = 0.257) in the current study; however, it did not reach statistical significance. Our hypothesis is that alterations in the RPE may precede alternations of the ISOS layer. An extended discussion on this topic is available in ***Supplementary Discussion***.

Our findings suggest that RPE thickness measurements may serve as an early biomarker for diabetes-related alterations in the retina. This biomarker may be captured through *in vivo* non-invasive OCT scanning prior to evident manifestations of diabetes and precedes changes in other retinal layers. Thus, it is unlikely that the

alterations in the RPE measurements observed in our study were caused by changes in other retinal layers. Additionally, RPE-associated metabolic alterations were found to improve the predictability and clinical utility of T2DM and related microvascular phenotypes beyond traditional risk factors, further demonstrating the potential for RPE to indicate diabetes-related biological changes.

Reference

1. Paulsen AJ, Pinto A, Merten N, Chen Y, Fischer ME, Huang GH, Klein BEK, Schubert CR, Cruickshanks KJ. Factors Associated with the Macular Ganglion Cell-Inner Plexiform Layer Thickness in a Cohort of Middle-aged U.S. Adults. *Optom Vis Sci.* 2021 Mar 1;98(3):295-305. doi: 10.1097/OPX.0000000000001650. PMID: 33771958; PMCID: PMC8007043.
2. Ciprés M, Satue M, Melchor I, Gil-Arribas L, Vilades E, Garcia-Martin E. Retinal neurodegeneration in patients with type 2 diabetes mellitus without diabetic retinopathy. *Arch Soc Esp Oftalmol (Engl Ed).* 2022 Apr;97(4):205-218. doi: 10.1016/j.oftale.2022.02.009. Epub 2022 Mar 2. PMID: 35523467.
3. Jonsson KB, Frydkjaer-Olsen U, Grauslund J. Vascular Changes and Neurodegeneration in the Early Stages of Diabetic Retinopathy: Which Comes First? *Ophthalmic Res.* 2016;56(1):1-9. doi: 10.1159/000444498. Epub 2016 Apr 2. PMID: 27035578.
4. Tang Z, Chan MY, Leung WY, Wong HY, Ng CM, Chan VTT, Wong R, Lok J, Szeto S, Chan JCK, Tham CC, Wong TY, Cheung CY. Assessment of retinal neurodegeneration with spectral-domain optical coherence tomography: a systematic review and meta-analysis. *Eye (Lond).* 2021 May;35(5):1317-1325. doi: 10.1038/s41433-020-1020-z. Epub 2020 Jun 24. PMID: 32581390; PMCID: PMC8182828.
5. Santos AR, Ribeiro L, Bandello F, Lattanzio R, Egan C, Frydkjaer-Olsen U, García-Arumí J, Gibson J, Grauslund J, Harding SP, Lang GE, Massin P, Midena E, Scanlon P, Aldington SJ, Simão S, Schwartz C, Ponsati B, Porta M, Costa MÂ, Hernández C, Cunha-Vaz J, Simó R; European Consortium for the Early Treatment

of Diabetic Retinopathy (EUROCONDOR). Functional and Structural Findings of Neurodegeneration in Early Stages of Diabetic Retinopathy: Cross-sectional Analyses of Baseline Data of the EUROCONDOR Project. *Diabetes*. 2017 Sep;66(9):2503-2510. doi: 10.2337/db16-1453. Epub 2017 Jun 29. PMID: 28663190.

6. Hammoum I, Benlarbi M, Dellaa A, Szabó K, Dékány B, Csaba D, Almási Z, Hajdú RI, Azaiz R, Charfeddine R, Lukáts Á, Ben Chaouacha-Chekir R. Study of retinal neurodegeneration and maculopathy in diabetic Meriones shawi: A particular animal model with human-like macula. *J Comp Neurol*. 2017 Sep 1;525(13):2890-2914. doi: 10.1002/cne.24245. Epub 2017 Jun 7. PMID: 28542922.
7. Énzöly A, Szabó A, Kántor O, Dávid C, Szalay P, Szabó K, Szél Á, Németh J, Lukáts Á. Pathologic alterations of the outer retina in streptozotocin-induced diabetes. *Invest Ophthalmol Vis Sci*. 2014 May 20;55(6):3686-99. doi: 10.1167/iovs.13-13562. PMID: 24845643.
8. Karaca C, Karaca Z. Beyond Hyperglycemia, Evidence for Retinal Neurodegeneration in Metabolic Syndrome. *Invest Ophthalmol Vis Sci*. 2018 Mar 1;59(3):1360-1367. doi: 10.1167/iovs.17-23376. PMID: 29625459.

Supplementary Table S13

Multivariable Cox models for evaluating associations between various retinal layer thickness and risk of incident T2DM in the UKB cohort.

Risk of T2DM *†	HR	95% CI	P ‡	
Overall retinal thickness	0.926	0.835	1.027	0.142
Outer retinal thickness				
RPE	0.856	0.768	0.954	0.005
ELM-ISOS	0.945	0.852	1.047	0.276
Inner retinal thickness				
RNFL	1.032	0.931	1.144	0.548
GC-IPL	0.978	0.884	1.082	0.668

* Adjusted for age, sex, ethnicity, household income, Townsend deprivation index, education, smoking, drinking, body-mass index, and use of antihypertensive and lipid-lowering medication.

† Per-SD change of various retinal thickness, measured by SD-OCT centered at the foveal.

‡ Bold indicates statistically significant.

T2DM = type 2 diabetes mellitus; HR = hazard ratio; CI = confidence interval; RPE = retinal pigment epithelium; ELM-ISOS = external limiting membrane-inner segment outer segment; RNFL = retinal nerve fiber layer; GC-IPL = ganglion cell-inner plexiform layer; SD-OCT = spectral-domain optical coherence tomography.

Comment #11. Discussion: L.181 - It seems that the authors establish a direct link between the thickness of a monolayer of cells and functional alterations of this cell layer. The volume of the pigmentary epithelium varies with multiple factors including osmotic factors or the circadian rhythm. The OCT does not allow to measure with precision the height of this epithelium and measures in reality all the interdigitations between the villi of the RPE and the external segments of the photoreceptors until the choriocapillaris. It is enough that the phagocytosis activity is reduced so that this height can vary without being associated with any cell layer assay. The pigmentation of the PE can also influence its reflectivity and it is likely that the thickness appears greater if there is more pigment and thinner in OCT if there is less pigment. But this does not mean that PE is pathological. The article cited by the authors shows transient functional abnormalities of the electrical function of the RPE at the time of the installation of hyperglycemia which is understandable since this activity depends on ionic transport. The other cited paper reports changes in the RPE and their thinning observed in the models of diabetes, as event secondary to degeneration of the outer segments of the cones. Morphological studies of the epithelium in type 2 diabetics show changes in the cytoskeleton, cellular edema and recruitment of microglial cells to the cell contact, which should result in an increase in thickness measured in OCT. Moreover, several studies in humans have confirmed the presence of microglial cells in the outer retina of diabetic patients without retinopathy. The apparent thinning of the RPE/Bruch layer could be due to a dilution of the pigment due to RPE proliferation. The skin color of the patients should also be taken into account and the role of pigmentation on the measured thickness should be analyzed. Since metabolomics identified lipid pathways, the thickness of the outer segments and of all retinal layers is necessary to ensure that the RPET change is not secondary to abnormal cone function due to change in vesicular transports of lipoproteins from the choroid to the RPE and the photoreceptors.

Response: We sincerely thank you for your thorough and professional review of our manuscript. We total agree with you that any conclusions relating to RPET in the current study rely on the interpretation of OCT imaging, which employed optical

reflectivity changes to identify inner and outer RPE boundaries and to infer changes in its thickness.¹ It is important to note that there could be decoupling between optical reflectivity changes and true thickness changes when optical reflectivity changes occur within the RPE cell for reasons other than shortening or loss of cells. Therefore, changes of RPET measurements could result from changes in structural, morphologic, and density indices, making it challenging to establish a direct relationship with function so far.² We have moderated our statements throughout the text and have also highlighted this limitation in the **Discussion** section (**Page 13, Line 350**).

Since the pigmentation of the RPE can also influence its reflectivity, we conducted a sensitivity analysis to mitigate the potential impact of differences in pigmentation on RPET measurements across different ethnic groups. We adjusted for more detailed ethnicity classifications, including further subdivisions for White, South Asian, East Asian, Black, and Mixed races. Details regarding this analysis have been added to the **Methods** section (**Page 18, Line 502**). We confirmed that all results remained unchanged (**Supplementary Table S3**). Additionally, we acknowledge that circadian rhythms may have influenced the *in vivo* measurement of RPE.³ However, we were unable to fully account for this factor in our study due to the unavailability of information regarding the timing of the OCT scans. Despite this limitation, we believe that the impact of circadian factors on our results of population-level was minimal. This is because the order in which participants underwent OCT scans was not based on any specific characteristics, which means that different measurements were conducted at random times without being influenced by individual characteristics, such as gender or age. The distribution of sample characteristics across each time point should be balanced. Regarding the interpretation of the regression model, it's important to note that the factor that evenly influences the independent variable only (e.g., timing of OCT scans) does not affect the association between the independent variable and dependent variables. Even if these variables were included, they would be excluded from the univariate feature selection process and thus would

not participate in the final multivariate model since they are unlikely to affect the dependent variable. However, the effect of circadian factors may need to be fully considered when using this indicator at the individual-level. This limitation has been added to the *Discussion* section (**Page 13, Line 358**).

To ensure that the changes in RPET were not secondary to abnormalities in cone function and/or changes in other retinal layers, we included measurements of other layers of the retina in the revised manuscript for additional analyses (**Supplementary Table S13**). Of overall average retinal thickness, average RPE thickness, average ELM-ISOS thickness, average GC-IPL thickness, and average RNFL thickness, only average RPE thickness demonstrated significantly association with the risk of future T2DM, after adjusted for age, sex, ethnicity, household income, Townsend deprivation index, education, smoking, drinking, body-mass index, and use of antihypertensive and lipid-lowering medication (all of which reached a threshold of $P < 0.1$ in the univariate models). These results suggest that the changes in RPET measurements that we observed are unlikely to be secondary to photoreceptor cell layers or other retinal layers, since these retinal layers did not show significant associations with incident diabetes.

The early changes of RPET in diabetes have been established in previous studies. In the experiment conducted by Hammoum et al.⁴, the outer retina, including RPE and photoreceptors, was significantly impaired at three months in the Meriones shawi diabetes model, while no significant differences were observed in ganglion cells compared to the control group. Enzsoly et al.⁵ also found early structural and functional damage to photoreceptors and RPE cells in streptozotocin-induced diabetic rats, even when overall retinal thickness appeared to be unchanged. Additionally, Lu et al.⁶ demonstrated a concomitant decrease in the number of photoreceptor cell nuclei layers and a shortening of RPE cell height in Long-Evans Tokushima fatty rats, which are spontaneously diabetic, not later than in the inner retina. These findings suggest that outer retinal changes could be an early indicator

of diabetes. Interestingly, our study also revealed that changes in RPET measurements may precede changes in the photoreceptor cell layer, as the ISOS layer, which represents the photoreceptors, did not show a significant association with incident T2DM in our study after adjusting for potential confounding factors. An extended discussion on this topic can be found in ***Supplementary Discussion: Associations between T2DM risk and other retinal layers.***

To obtain more accurate RPET measurements, we further employed SS-OCT with deeper penetration, faster acquisition, higher resolution in an ethnically diverse cohort⁷ to investigate the association of RPET with diabetic microvascular phenotypes. With the use of longer wavelengths, SS-OCT reduces attenuation from ocular opacities and provides deeper penetration and lower scattering from the RPE.⁸⁻⁹ Employing more accurate RPET measurements, we provide further evidence for the independent associations of this retinal biomarker with the risks of diabetic microvascular phenotypes (***Supplementary Table S9–S11***, as provided below for your convenience). Furthermore, we performed additional experiments using more sensitive LC/MS assay to systematically profile the metabolomics of participants and identified an expanded range of RPE-associated metabolites, particularly those at low concentrations.¹⁰ The incorporation of these metabolites into the clinical indicators-based model resulted in improvements in discriminative power and clinical utility, further confirming the role of RPET metabolic fingerprints in facilitating risk stratification for T2DM-related damages (***Supplementary Figure S3***, as provided below for your convenience).

While our primary results highlight the biological changes captured by altered RPET measurements as biomarkers for individualized stratification of T2DM, our study does not aim to emphasize the importance of lipid metabolites in the underpinnings of RPET with diabetes. We would also not propose that the biological alterations captured by NMR metabolomics can fully or predominantly elucidate the underpinnings of the association between RPET and diabetes. Instead, combining

other omics approaches such as MS-based metabolomics, proteomics, and transcriptomics could be even more effective in unraveling the underlying mechanisms of their association. With LC/MS assay in the Guangzhou Diabetic Eye Study (GDES) cohort, a number of low-molecular-weight metabolites were also identified, which further contributed to added predictability and clinical utility for diabetic-related damages, encompassing amino acids, FAs, benzene, nucleotides, organic acids, heterocyclic compounds, and their derivatives (**Supplementary Table S12**, as provided below for your convenience). NMR and LC/MS assays in this study provided complementary coverage of metabolic markers, exhibiting a more comprehensive RPET metabolic landscape. Our study establishes a framework and concept that suggests any biological changes that trigger or coincide with changes of RPET measurements could potentially be involved in the early pathogenesis of T2DM, surpassing the metabolic changes captured by NMR metabolomics, since these changes are detectable up to a decade before the onset of T2DM.

We have also put forth some speculations for the biological underpinnings between RPET and diabetes (**Discussion** section, **Page XX, Line XX**). Alterations in lipid metabolism are deemed at the core of diabetes phenotypes,¹¹⁻¹² and studies have indicated that insulin resistance can stimulate triglyceride synthesis and VLDL production, leading to an excessive accumulation of VLDL and LDL.¹³⁻¹⁵ Studies on NMR metabolomic profiling have also linked VLDL and LDL particle sizes and concentrations to an increased risk of T2DM.¹⁶⁻¹⁹ These findings are in keeping with the adverse RPET metabolic state contributing to the increased T2DM risk observed in this study. Moreover, studies on retina have established that DR and its severity are associated with multiple VLDL and LDL particles, and lipid-lowering therapy has been shown to have significant benefits in preventing and mitigating DR.^{20,21} Modified lipid can have toxic effects on retinal cells, including RPE cells, and their excessive accumulation may result in oxidative stress and endoplasmic reticulum stress,²² potentially leading to dysfunction and structural damages of the RPE. Consistently, *in vitro* studies have demonstrated that RPE cells cultured under

diabetic-like conditions produce elevated levels of reactive oxygen species compared to those incubated under non-diabetic conditions.²³ We speculate that a specific metabolic state that prioritizes damage to the retina before evident T2DM has already taken place, and while these subtle metabolic alterations may not have a noticeable effect on the body yet, they are already detectable through in vivo OCT scanning and metabolomic profiling.

Hopefully this addresses your concerns. Thank you again for all your valuable feedback and suggestions. We appreciate your dedication and time in helping us improve the quality of this manuscript.

Reference

1. Yang Q, Reisman CA, Wang Z, Fukuma Y, Hangai M, Yoshimura N, Tomidokoro A, Araie M, Raza AS, Hood DC, Chan K. Automated layer segmentation of macular OCT images using dual-scale gradient information. *Opt Express*. 2010; 18(20):21293-307.
2. Ko F, Foster PJ, Strouthidis NG, Shweikh Y, Yang Q, Reisman CA, Muthy ZA, Chakravarthy U, Lotery AJ, Keane PA, Tufail A, Grossi CM, Patel PJ; UK Biobank Eye & Vision Consortium. Associations with Retinal Pigment Epithelium Thickness Measures in a Large Cohort: Results from the UK Biobank. *Ophthalmology*. 2017 Jan;124(1):105-117. doi: 10.1016/j.ophtha.2016.07.033. Epub 2016 Oct 6. PMID: 27720551.
3. Fanjul-Moles ML, López-Riquelme GO. Relationship between Oxidative Stress, Circadian Rhythms, and AMD. *Oxid Med Cell Longev*. 2016; 2016:7420637. doi: 10.1155/2016/7420637. Epub 2015 Dec 28. PMID: 26885250; PMCID: PMC4738726.
4. Hammoum I, Benlarbi M, Dellaa A, Szabó K, Dékány B, Csaba D, Almási Z, Hajdú RI, Azaiz R, Charfeddine R, Lukáts Á, Ben Chaouacha-Chekir R. Study of retinal neurodegeneration and maculopathy in diabetic *Meriones shawi*: A particular animal model with human-like macula. *J Comp Neurol*. 2017 Sep

- 1;525(13):2890-2914. doi: 10.1002/cne.24245. Epub 2017 Jun 7. PMID: 28542922.
5. Énzsöly A, Szabó A, Kántor O, Dávid C, Szalay P, Szabó K, Szél Á, Németh J, Lukáts Á. Pathologic alterations of the outer retina in streptozotocin-induced diabetes. *Invest Ophthalmol Vis Sci.* 2014 May 20;55(6):3686-99. doi: 10.1167/iovs.13-13562. PMID: 24845643.
 6. Lu ZY, Bhutto IA, Amemiya T. Retinal changes in Otsuka long-evans Tokushima Fatty rats (spontaneously diabetic rat) -possibility of a new experimental model for diabetic retinopathy. *Jpn J Ophthalmol.* 2003 Jan-Feb;47(1):28-35. doi: 10.1016/s0021-5155(02)00631-7. PMID: 12586175.
 7. Zhang S, Chen Y, Wang L, Li Y, Tang X, Liang X, He M, Wenyong H, Wang W; GDES group. Design and Baseline Data of the Diabetes Registration Study: Guangzhou Diabetic Eye Study. *Curr Eye Res.* 2023 Feb 27:1-9. doi: 10.1080/02713683.2023.2182745. Epub ahead of print. PMID: 36803011.
 8. Láíns I, Wang JC, Cui Y, Katz R, Vingopoulos F, Staurenghi G, Vavvas DG, Miller JW, Miller JB. Retinal applications of swept source optical coherence tomography (OCT) and optical coherence tomography angiography (OCTA). *Prog Retin Eye Res.* 2021 Sep; 84:100951. doi: 10.1016/j.preteyeres.2021.100951. Epub 2021 Jan 28. PMID: 33516833.
 9. Jia Y, Bailey ST, Wilson DJ, Tan O, Klein ML, Flaxel CJ, Potsaid B, Liu JJ, Lu CD, Kraus MF, Fujimoto JG, Huang D. Quantitative optical coherence tomography angiography of choroidal neovascularization in age-related macular degeneration. *Ophthalmology.* 2014 Jul;121(7):1435-44. doi: 10.1016/j.ophtha.2014.01.034. Epub 2014 Mar 27. PMID: 24679442; PMCID: PMC4082740.
 10. Chen W, Gong L, Guo Z, Wang W, Zhang H, Liu X, Yu S, Xiong L, Luo J. A novel integrated method for large-scale detection, identification, and quantification of widely targeted metabolites: application in the study of rice metabolomics. *Mol Plant.* 2013 Nov;6(6):1769-80. doi: 10.1093/mp/sst080. Epub 2013 May 23. PMID: 23702596.

11. Kane JP, Pullinger CR, Goldfine ID, Malloy MJ. Dyslipidemia and diabetes mellitus: Role of lipoprotein species and interrelated pathways of lipid metabolism in diabetes mellitus. *Curr Opin Pharmacol*. 2021 Dec;61:21-27. doi: 10.1016/j.coph.2021.08.013. Epub 2021 Sep 22. PMID: 34562838.
12. Eid S, Sas KM, Abcouwer SF, Feldman EL, Gardner TW, Pennathur S, Fort PE. New insights into the mechanisms of diabetic complications: role of lipids and lipid metabolism. *Diabetologia*. 2019 Sep;62(9):1539-1549. doi: 10.1007/s00125-019-4959-1. Epub 2019 Jul 25. PMID: 31346658; PMCID: PMC6679814.
13. Vergès B. Abnormal hepatic apolipoprotein B metabolism in type 2 diabetes. *Atherosclerosis*. 2010 Aug;211(2):353-60. doi: 10.1016/j.atherosclerosis.2010.01.028. Epub 2010 Jan 29. PMID: 20189175.
14. Annuzzi G, De Natale C, Iovine C, Patti L, Di Marino L, Coppola S, Del Prato S, Riccardi G, Rivellese AA. Insulin resistance is independently associated with postprandial alterations of triglyceride-rich lipoproteins in type 2 diabetes mellitus. *Arterioscler Thromb Vasc Biol*. 2004 Dec;24(12):2397-402. doi: 10.1161/01.ATV.0000146267.71816.30. Epub 2004 Sep 30. PMID: 15458975.
15. Malmström R, Packard CJ, Caslake M, Bedford D, Stewart P, Yki-Järvinen H, Shepherd J, Taskinen MR. Defective regulation of triglyceride metabolism by insulin in the liver in NIDDM. *Diabetologia*. 1997 Apr;40(4):454-62. doi: 10.1007/s001250050700. PMID: 9112023.
16. Festa A, Williams K, Hanley AJ, Otvos JD, Goff DC, Wagenknecht LE, Haffner SM. Nuclear magnetic resonance lipoprotein abnormalities in prediabetic subjects in the Insulin Resistance Atherosclerosis Study. *Circulation*. 2005 Jun 28;111(25):3465-72. doi: 10.1161/CIRCULATIONAHA.104.512079. PMID: 15983261.
17. Mora S, Otvos JD, Rosenson RS, Pradhan A, Buring JE, Ridker PM. Lipoprotein particle size and concentration by nuclear magnetic resonance and incident type 2 diabetes in women. *Diabetes*. 2010 May;59(5):1153-60. doi: 10.2337/db09-1114. Epub 2010 Feb 25. PMID: 20185808; PMCID: PMC2857895.

18. Fizelova M, Miilunpohja M, Kangas AJ, Soininen P, Kuusisto J, Ala-Korpela M, Laakso M, Stančáková A. Associations of multiple lipoprotein and apolipoprotein measures with worsening of glycemia and incident type 2 diabetes in 6607 non-diabetic Finnish men. *Atherosclerosis*. 2015 May;240(1):272-7. doi: 10.1016/j.atherosclerosis.2015.03.034. Epub 2015 Mar 23. PMID: 25818853.
19. Ahola-Olli AV, Mustelin L, Kalimeri M, Kettunen J, Jokelainen J, Auvinen J, Puukka K, Havulinna AS, Lehtimäki T, Kähönen M, Juonala M, Keinänen-Kiukaanniemi S, Salomaa V, Perola M, Järvelin MR, Ala-Korpela M, Raitakari O, Würtz P. Circulating metabolites and the risk of type 2 diabetes: a prospective study of 11,896 young adults from four Finnish cohorts. *Diabetologia*. 2019 Dec;62(12):2298-2309. doi: 10.1007/s00125-019-05001-w. Epub 2019 Oct 4. PMID: 31584131; PMCID: PMC6861432.
20. Lyons TJ, Jenkins AJ, Zheng D, Lackland DT, McGee D, Garvey WT, Klein RL. Diabetic retinopathy and serum lipoprotein subclasses in the DCCT/EDIC cohort. *Invest Ophthalmol Vis Sci*. 2004 Mar;45(3):910-8. doi: 10.1167/iovs.02-0648. PMID: 14985310.
21. ACCORD Study Group; ACCORD Eye Study Group; Chew EY, Ambrosius WT, Davis MD, Danis RP, Gangaputra S, Greven CM, Hubbard L, Esser BA, Lovato JF, Perdue LH, Goff DC Jr, Cushman WC, Ginsberg HN, Elam MB, Genuth S, Gerstein HC, Schubart U, Fine LJ. Effects of medical therapies on retinopathy progression in type 2 diabetes. *N Engl J Med*. 2010 Jul 15;363(3):233-44. doi: 10.1056/NEJMoa1001288. Epub 2010 Jun 29. Erratum in: *N Engl J Med*. 2011 Jan 13;364(2):190. Erratum in: *N Engl J Med*. 2012 Dec 20;367(25):2458. PMID: 20587587; PMCID: PMC4026164.
22. Du M, Wu M, Fu D, Yang S, Chen J, Wilson K, Lyons TJ. Effects of modified LDL and HDL on retinal pigment epithelial cells: a role in diabetic retinopathy? *Diabetologia*. 2013 Oct;56(10):2318-28. doi: 10.1007/s00125-013-2986-x. Epub 2013 Jul 11. PMID: 23842729; PMCID: PMC4557884.
23. Tonade D, Kern TS. Photoreceptor cells and RPE contribute to the development of diabetic retinopathy. *Prog Retin Eye Res*. 2021 Jul;83:100919. doi:

[10.1016/j.preteyeres.2020.100919](https://doi.org/10.1016/j.preteyeres.2020.100919). Epub 2020 Nov 12. PMID: 33188897; PMCID: PMC8113320.

Supplementary Table S3

Sensitivity analyses on the association of average RPET with T2DM risk in UKB cohort.

Risk of T2DM *	HR	95% CI	P value †
Overall population-I			
Model 1 ‡	0.856	0.767-0.954	0.005
Model 2 §	0.868	0.779-0.968	0.011
Model 3 ¶	0.849	0.757-0.951	0.005
Model 4 #	0.867	0.775-0.970	0.013
Model 5 **	0.845	0.735-0.973	0.019
Stratified by PRS risk of AMD			
Quartile 2 to Quartile 4 (Participants with low-moderate AMD risk)			
Model 1 ‡	0.835	0.734-0.950	0.006
Model 2 §	0.847	0.744-0.963	0.011
Model 3 ¶	0.833	0.726-0.956	0.009
Model 4 #	0.833	0.727-0.954	0.009
Model 5 **	0.792	0.666-0.941	0.008
Quartile 1 (Participants with highest AMD risk)			
Model 1 ‡	0.900	0.731-1.107	0.319
Model 2 §	0.921	0.747-1.137	0.446
Model 3 ¶	0.907	0.723-1.138	0.400
Model 4 #	0.940	0.762-1.161	0.568
Model 5 **	0.969	0.757-1.241	0.802

* Per-SD change of RPET.

† Bold indicates significant.

‡ Adjusted for age, sex, ethnicity, Townsend deprivation index, household income, education, smoking, drinking, body mass index, and use of antihypertensive and lipid-lowering medication.

§ Further adjusted for more comprehensive ethnicity classifications with further subdivisions for South Asian, East Asian, Black, and Mixed races.

¶ Further adjusted for biological ages.¹

Further adjusted for frailty score.²

** Further excluded cancer, cardiovascular diseases, and renal diseases.

RPET= retinal pigment epithelium thickness; T2DM= type 2 diabetes mellitus; HR= hazard ratio; CI= confidence interval; SD= standard deviation.

Reference

1. Yang G, Cao X, Li X, Zhang J, Ma C, Zhang N, Lu Q, Crimmins EM, Gill TM, Chen X, Liu Z. Association of Unhealthy Lifestyle and Childhood Adversity With Acceleration of Aging Among UK Biobank Participants. *JAMA Netw Open*. 2022 Sep 1;5(9): e2230690. doi: 10.1001/jamanetworkopen.2022.30690. PMID: 36066889; PMCID: PMC9449787.
2. Fried LP, Tangen CM, Walston J, Newman AB, Hirsch C, Gottdiener J, Seeman T, Tracy R, Kop WJ, Burke G, McBurnie MA; Cardiovascular Health Study Collaborative Research Group. Frailty in older adults: evidence for a phenotype. *J Gerontol A Biol Sci Med Sci*. 2001 Mar;56(3):M146-56. doi: 10.1093/gerona/56.3.m146. PMID: 11253156.

Supplementary Table S13

Multivariable Cox models for evaluating associations between various retinal layer thickness and risk of incident T2DM in the UKB cohort.

Risk of T2DM *†	HR	95% CI	P ‡	
Overall retinal thickness	0.926	0.835	1.027	0.142
Outer retinal thickness				
RPE	0.856	0.768	0.954	0.005
ELM-ISOS	0.945	0.852	1.047	0.276
Inner retinal thickness				
RNFL	1.032	0.931	1.144	0.548
GC-IPL	0.978	0.884	1.082	0.668

* Adjusted for age, sex, ethnicity, household income, Townsend deprivation index, education, smoking, drinking, body-mass index, and use of antihypertensive and lipid-lowering medication.

† Per-SD change of various retinal thickness, measured by SD-OCT centered at the foveal.

‡ Bold indicates statistically significant.

T2DM = type 2 diabetes mellitus; HR = hazard ratio; CI = confidence interval; RPE = retinal pigment epithelium; ELM-ISOS = external limiting membrane-inner segment outer segment; RNFL = retinal nerve fiber layer; GC-IPL = ganglion cell-inner plexiform layer; SD-OCT = spectral-domain optical coherence tomography.

Supplementary Table S9

Multivariable logistic models for evaluating association between RPET across all ETDRS subfields and diabetic retinopathy.

RPET *†	Incident DR			DR progression ‡				
	OR	95% CI	P §	OR	95% CI	P §		
Total average	0.781	0.633	0.965	0.022	0.830	0.678	1.002	0.070
Outer average	0.786	0.628	0.983	0.035	0.846	0.685	1.045	0.121
Outer superior	0.767	0.600	0.981	0.035	0.763	0.600	0.969	0.026
Outer nasal	0.960	0.759	1.213	0.728	0.692	0.502	0.955	0.025
Outer inferior	0.890	0.663	1.195	0.437	0.783	0.537	1.141	0.203
Outer temporal	0.705	0.511	0.970	0.032	0.935	0.740	1.182	0.574
Inner average	0.736	0.588	0.922	0.008	0.738	0.592	0.920	0.007
Inner superior	0.832	0.658	1.051	0.124	0.714	0.560	0.909	0.006
Inner nasal	0.652	0.494	0.860	0.003	0.694	0.530	0.908	0.008
Inner inferior	0.887	0.682	1.151	0.370	0.783	0.573	1.070	0.125
Inner temporal	0.803	0.598	2.104	0.143	0.757	0.568	1.008	0.057
Central field	0.992	0.822	1.197	0.933	1.043	0.875	1.244	0.639

* Adjusted for age, sex, diabetes duration, HbA1c, body mass index, systolic blood pressure, smoking, drinking, and hyperlipidemia.

† Per-SD change of RPET, measured by SS-OCT using the ETDRS nine-pane grid.

‡ DR progression is defined as an increase in ETDRS grading score compared with baseline.

§ Bold indicates significant.

RPET = retinal pigment epithelium thickness; ETDRS = Early Treatment Diabetic Retinopathy Study; DR = diabetic retinopathy; CI = confidence interval, SS-OCT = swept-source optical coherence tomography.

Supplementary Table S10

Multivariable logistic models for evaluating association between RPET across all ETDRS subfields and fast progressor of renal function.

RPET *†	Fast eGFR decline ‡§			Halving of eGFR §				
	OR	95% CI	P §	OR	95% CI	P ¶		
Total average	0.839	0.703	0.999	0.049	0.271	0.111	0.665	0.004
Outer average	0.877	0.736	1.046	0.145	0.337	0.138	0.827	0.018
Outer superior	0.792	0.649	0.967	0.022	0.333	0.124	0.890	0.029
Outer nasal	0.739	0.557	0.978	0.035	0.648	0.189	2.217	0.489
Outer inferior	0.925	0.773	1.106	0.394	0.049	0.006	0.418	0.006
Outer temporal	0.936	0.797	1.100	0.422	0.152	0.039	0.586	0.006
Inner average	0.824	0.687	0.988	0.036	0.210	0.080	0.549	0.001
Inner superior	0.877	0.731	1.053	0.161	0.292	0.113	0.756	0.011
Inner nasal	0.767	0.619	0.952	0.016	0.281	0.096	0.824	0.021
Inner inferior	0.984	0.676	1.064	0.155	0.107	0.024	0.477	0.003
Inner temporal	0.909	0.753	1.099	0.325	0.095	0.026	0.353	<0.001
Central	0.992	0.829	1.186	0.927	0.127	0.039	0.414	0.001

* Adjusted for age, sex, diabetes duration, HbA1c, body mass index, systolic blood pressure, smoking, drinking, and hyperlipidemia.

† Per-SD change of RPET, measured by SS-OCT using the ETDRS nine-pane grid.

‡ GFR is estimated using the Chronic Kidney Disease-Epidemiology Collaboration (CKD-EPI) equation based on serum creatinine.¹

§ Fast eGFR decline is defined as a decrease of 20% or more.^{2,3} Halving of eGFR was defined as decline in eGFR by $\geq 50\%$ from baseline.^{4,5}

¶ Bold indicates significant.

RPET = retinal pigment epithelium thickness; ETDRS = Early Treatment Diabetic Retinopathy Study; eGFR = estimated glomerular filtration rate;

CI = confidence interval; SS-OCT = swept-source optical coherence tomography.

Reference:

1. Levey AS, Stevens LA. Estimating GFR using the CKD Epidemiology Collaboration (CKD-EPI) creatinine equation: more accurate GFR estimates, lower CKD prevalence estimates, and better risk predictions. *Am J Kidney Dis.* 2010 Apr;55(4):622-7. doi: 10.1053/j.ajkd.2010.02.337. PMID: 20338463; PMCID: PMC2846308.
2. Yang W, Xie D, Anderson AH, Joffe MM, Greene T, Teal V, Hsu CY, Fink JC, He J, Lash JP, Ojo A, Rahman M, Nessel L, Kusek JW, Feldman HI; CRIC Study Investigators. Association of kidney disease outcomes with risk factors for CKD: findings from the Chronic Renal Insufficiency Cohort (CRIC) study. *Am J Kidney Dis.* 2014 Feb;63(2):236-43. doi: 10.1053/j.ajkd.2013.08.028. Epub 2013 Oct 30. PMID: 24182662; PMCID: PMC3946885.
3. Heinzl A, Kammer M, Mayer G, Reindl-Schwaighofer R, Hu K, Perco P, Eder S, Rosivall L, Mark PB, Ju W, Kretzler M, Gilmour P, Wilson JM, Duffin KL, Abdalla M, McCarthy MI, Heinze G, Heerspink HL, Wiecek A, Gomez MF, Oberbauer R; BEAt-DKD Consortium. Validation of Plasma Biomarker Candidates for the Prediction of eGFR Decline in Patients With Type 2 Diabetes. *Diabetes Care.* 2018 Sep;41(9):1947-1954. doi: 10.2337/dc18-0532. Epub 2018 Jul 6. PMID: 29980527; PMCID: PMC6105325.
4. Wen D, Zheng Z, Surapaneni A, Yu B, Zhou L, Zhou W, Xie D, Shou H, Avila-Pacheco J, Kalim S, He J, Hsu CY, Parsa A, Rao P, Sondheimer J, Townsend R, Waikar SS, Rebholz CM, Denburg MR, Kimmel PL, Vasan RS, Clish CB, Coresh J, Feldman HI, Grams ME, Rhee EP; CKD Biomarkers Consortium and CRIC Study Investigators. Metabolite profiling of CKD progression in the chronic renal insufficiency cohort study.

JCI Insight. 2022 Oct 24;7(20):e161696. doi: 10.1172/jci.insight.161696. PMID: 36048534; PMCID: PMC9714776.

5. Anderson AH, Xie D, Wang X, Baudier RL, Orlandi P, Appel LJ, Dember LM, He J, Kusek JW, Lash JP, Navaneethan SD, Ojo A, Rahman M, Roy J, Scialla JJ, Sondheimer JH, Steigerwalt SP, Wilson FP, Wolf M, Feldman HI; CRIC Study Investigators. Novel Risk Factors for Progression of Diabetic and Nondiabetic CKD: Findings From the Chronic Renal Insufficiency Cohort (CRIC) Study. *Am J Kidney Dis.* 2021 Jan;77(1):56-73.e1. doi: 10.1053/j.ajkd.2020.07.011. Epub 2020 Aug 28. PMID: 32866540; PMCID: PMC7752839.

Supplementary Table S11

Multivariable logistic models for evaluating association between RPET across all ETDRS subfields and retinal capillary rarefaction.

RPET *†	Fast retinal capillary rarefaction in macula ‡				Fast retinal capillary rarefaction in ONH ‡			
	OR	95% CI		P §	OR	95% CI		P §
Total average	0.783	0.611	1.004	0.054	0.825	0.586	1.162	0.272
Outer average	0.739	0.564	0.969	0.029	0.856	0.589	1.244	0.414
Outer superior	1.142	0.922	1.415	0.221	0.896	0.592	1.357	0.603
Outer nasal	0.987	0.758	1.285	0.924	0.736	0.417	1.302	0.293
Outer inferior	0.872	0.637	1.192	0.391	0.820	0.444	1.516	0.528
Outer temporal	0.743	0.557	0.992	0.044	0.733	0.427	1.259	0.260
Inner average	0.891	0.689	1.155	0.385	0.758	0.520	1.104	0.150
Inner superior	0.934	0.744	1.174	0.557	0.624	0.410	0.949	0.028
Inner nasal	0.946	0.742	1.208	0.657	0.866	0.572	1.310	0.494
Inner inferior	0.760	0.533	1.084	0.130	0.780	0.450	1.355	0.379
Inner temporal	0.855	0.676	1.081	0.191	0.560	0.331	0.950	0.032
Central	0.828	0.622	1.102	0.196	0.934	0.666	1.310	0.692

* Adjusted for age, sex, diabetes duration, HbA1c, body mass index, systolic blood pressure, smoking, drinking, and history of hyperlipidemia.

† Per-SD change of RPET, measured by SS-OCT using the ETDRS nine-pane grid.

‡ Fast capillary rarefaction is defined as the longitudinal rates of vessel density decline of retinal deep capillary plexus in macular or optic nerve head regions measured by SS-OCT-angiography being in the first quartile, respectively.

§ Bold indicates significant. ONH = optic nerve head; RPET = retinal pigment epithelium thickness; ETDRS = Early Treatment Diabetic Retinopathy Study; CI = confidence interval; SS-OCT = swept-source optical coherence tomography.

Supplementary Figure S3

Added discriminative power and clinical utility of RPET metabolic fingerprints for stratifying T2DM adverse microvascular phenotypes in GDES.

Supplementary Table S12

RPET-associated metabolites identified using LC/MS assay in the Guangzhou Diabetic Eye Study (GDES) cohort after adjusting for other factors.

Metabolites *	Group	β	95%CI		P_{raw}	P_{FDR}
γ -Glu-Met	Amino acid and Its metabolites	2.855	2.147	3.563	$<1.0 \times 10^{-8}$	$<1.0 \times 10^{-8}$
L-Arginine	Amino acid and Its metabolites	2.607	1.882	3.332	$<1.0 \times 10^{-8}$	$<1.0 \times 10^{-8}$
Tryptamine	Tryptamines,Cholines,Pigments	2.327	1.590	3.065	$<1.0 \times 10^{-8}$	2.72×10^{-7}
Val-Ala	Amino acid and Its metabolites	2.247	1.525	2.969	$<1.0 \times 10^{-8}$	3.57×10^{-7}
Met-Glu	Amino acid and Its metabolites	2.270	1.535	3.006	$<1.0 \times 10^{-8}$	4.07×10^{-7}
Piperidine	Heterocyclic compounds	2.314	1.532	3.096	1.08×10^{-8}	1.38×10^{-6}
Azelaic Acid	Organic acid and Its derivatives	2.091	1.366	2.816	2.44×10^{-8}	2.38×10^{-6}
Glu-Met	Amino acid and Its metabolites	2.112	1.375	2.850	3.00×10^{-8}	2.67×10^{-6}
N-Propionylglycine	Amino acid and Its metabolites	2.204	1.434	2.974	3.11×10^{-8}	2.72×10^{-6}
2-(Formylamino)Benzoic Acid	Benzene and substituted derivatives	1.989	1.258	2.721	1.39×10^{-7}	1.07×10^{-5}
2'-O-methylcytidine	Nucleotide and Its metabolites	1.965	1.231	2.698	2.11×10^{-7}	1.51×10^{-5}
3-Hydroxy-L-phenylalanine	Amino acid and Its metabolites	1.854	1.120	2.588	9.72×10^{-7}	6.37×10^{-5}
DL-O-tyrosine	Amino acid and Its metabolites	1.777	1.032	2.521	3.63×10^{-6}	2.19×10^{-4}
D-Alloisoleucine	Amino acid and Its metabolites	1.747	0.959	2.534	1.62×10^{-5}	8.48×10^{-4}
5,6-Dimethylbenzimidazole	Heterocyclic compounds	1.654	0.903	2.405	1.85×10^{-5}	9.45×10^{-4}
L-Methionine	Amino acid and Its metabolites	1.653	0.891	2.416	2.47×10^{-5}	1.19×10^{-3}
Indoleacetaldehyde	Heterocyclic compounds	1.574	0.829	2.319	3.95×10^{-5}	1.66×10^{-3}
9,10-Epoxy-18-hydroxyoctadecanoic acid	FA	1.531	0.801	2.261	4.53×10^{-5}	1.81×10^{-3}
Trp-Gly	Amino acid and Its metabolites	1.603	0.837	2.368	4.64×10^{-5}	1.84×10^{-3}
Glycolithocholic acid	Bile acids	1.515	0.782	2.248	5.75×10^{-5}	2.09×10^{-3}
2'-O-methyluridine	Nucleotide and Its metabolites	1.561	0.803	2.320	6.21×10^{-5}	2.18×10^{-3}
Melatonin	Hormones and hormone related compounds	1.534	0.788	2.279	6.23×10^{-5}	2.19×10^{-3}

N-Acetyl-L-Leucine	Amino acid and Its metabolites	1.658	0.849	2.467	6.62×10 ⁻⁵	2.26×10 ⁻³
gamma-Glu-Phe	Amino acid and Its metabolites	1.593	0.796	2.390	1.00×10 ⁻⁴	3.10×10 ⁻³
12,13-DiHOME	FA	1.453	0.722	2.184	1.09×10 ⁻⁴	3.27×10 ⁻³
9,10-DiHOME	FA	1.449	0.718	2.180	1.15×10 ⁻⁴	3.40×10 ⁻³
N-acetyloronithine	Amino acid and Its metabolites	1.481	0.730	2.231	1.22×10 ⁻⁴	3.54×10 ⁻³
Gly-Phe	Amino acid and Its metabolites	1.437	0.699	2.174	1.49×10 ⁻⁴	4.14×10 ⁻³
Glu-Tyr	Amino acid and Its metabolites	1.462	0.705	2.220	1.70×10 ⁻⁴	4.58×10 ⁻³
Phe-Hyp	Amino acid and Its metabolites	1.410	0.673	2.146	1.94×10 ⁻⁴	5.04×10 ⁻³
Phe-Phe	Amino acid and Its metabolites	1.417	0.670	2.165	2.23×10 ⁻⁴	5.55×10 ⁻³
Pro-Asp	Amino acid and Its metabolites	1.398	0.658	2.138	2.33×10 ⁻⁴	5.71×10 ⁻³
Glu-Leu	Amino acid and Its metabolites	1.455	0.676	2.234	2.73×10 ⁻⁴	6.40×10 ⁻³
Phe-Gly	Amino acid and Its metabolites	1.385	0.641	2.129	2.86×10 ⁻⁴	6.60×10 ⁻³
L-Lysine	Amino acid and Its metabolites	1.360	0.624	2.096	3.20×10 ⁻⁴	7.16×10 ⁻³
L-Serine	Amino acid and Its metabolites	1.383	0.629	2.137	3.53×10 ⁻⁴	7.69×10 ⁻³
N-Amidino-L-Aspartate	Amino acid and Its metabolites	1.356	0.609	2.103	4.04×10 ⁻⁴	8.42×10 ⁻³
Multifidol	Benzene and substituted derivatives	1.334	0.594	2.073	4.40×10 ⁻⁴	8.90×10 ⁻³
Urocanic Acid	Organic acid and Its derivatives	1.338	0.595	2.080	4.46×10 ⁻⁴	8.98×10 ⁻³
PI (15:0/2:0)	GP	1.321	0.577	2.064	5.35×10 ⁻⁴	1.01×10 ⁻²
Caffeic Acid	Organic acid and Its derivatives	1.303	0.570	2.037	5.37×10 ⁻⁴	1.01×10 ⁻²
Hydroxyurea	Others	1.310	0.572	2.049	5.41×10 ⁻⁴	1.01×10 ⁻²
5'-Deoxy-5'-(Methylthio) Adenosine	Nucleotide and Its metabolites	1.318	0.556	2.080	7.45×10 ⁻⁴	1.32×10 ⁻²
Quinmerac	Benzene and substituted derivatives	-1.460	-2.306	-0.614	7.66×10 ⁻⁴	1.35×10 ⁻²
Proline betaine	Amino acid and Its metabolites	1.278	0.528	2.028	8.89×10 ⁻⁴	1.52×10 ⁻²
L-Ornithine	Amino acid and Its metabolites	1.256	0.519	1.994	8.96×10 ⁻⁴	1.53×10 ⁻²
L-Glutamine	Amino acid and Its metabolites	1.243	0.507	1.980	9.98×10 ⁻⁴	1.66×10 ⁻²
N, N-Bis(2-hydroxyethyl) dodecanamide	FA	1.238	0.496	1.979	1.14×10 ⁻³	1.84×10 ⁻²

Trolox	Benzene and substituted derivatives	1.235	0.491	1.978	1.20×10^{-3}	1.91×10^{-2}
Asp-Leu	Amino acid and Its metabolites	1.220	0.482	1.958	1.27×10^{-3}	2.00×10^{-2}
4-Acetylamino benzoic acid	Organic acid and Its derivatives	1.205	0.469	1.940	1.40×10^{-3}	2.15×10^{-2}
N-(2-hydroxyethyl)-3-pyridinecarboxamide	Heterocyclic compounds	1.307	0.502	2.112	1.53×10^{-3}	2.31×10^{-2}
Creatine phosphate	Nucleotide and Its metabolites	1.194	0.446	1.943	1.86×10^{-3}	2.71×10^{-2}
Phe-Val	Amino acid and Its metabolites	1.197	0.446	1.948	1.87×10^{-3}	2.72×10^{-2}
2-Aminophenol	Benzene and substituted derivatives	1.157	0.421	1.892	2.14×10^{-3}	3.06×10^{-2}
Hippuric Acid	Organic acid and Its derivatives	1.120	0.385	1.855	2.95×10^{-3}	4.01×10^{-2}
L-Asparagine Anhydrous	Amino acid and Its metabolites	1.162	0.394	1.929	3.14×10^{-3}	4.21×10^{-2}
Phenethylamine	Alcohol and amines	1.175	0.384	1.966	3.75×10^{-3}	4.85×10^{-2}

* Adjusted for age, sex duration of diabetes, HbA1c, body-mass index, systolic blood pressure, smoking, drinking, and hyperlipidemia.

Replies to Reviewer #2:

Comment #1. This is an interesting study. The reviewer has some comments for the authors to consider.

Response: We sincerely appreciate the recognition of our work and appreciate your kind words. We would also like to express our heartfelt gratitude for the time and effort you have dedicated to reviewing our manuscripts, which has undoubtedly enhanced the quality of our work. We have carefully considered all your comments and suggestions, and as a result, we have made significant revisions to the manuscript. We hope that these changes have addressed your concerns and improved the overall quality of our work.

Comment #2. The authors only considered the effect of age-related macular degeneration on the retinal pigment epithelium. To our knowledge, retinitis pigmentosa and Stargardts disease, among others, are retinal degenerative diseases that are accompanied by RPE dysfunction. I would like the authors to provide information on other diseases or to add the limitation in the discussion.

Response: We acknowledge and agree with your concerns that both retinitis pigmentosa and Stargardt's disease can affect the function and/or structure of the RPE. We apologize for the lack of detailed information about the eligibility criteria in our previous version of the manuscript. Accordingly, we have detailed the inclusion/exclusion criteria in the revised the manuscript to clarify that these two diseases (ICD-10 H35.5 and H35.8, respectively) were excluded in the exclusion criteria under "other retinal diseases". Moreover, we have also excluded other retinal disorders such as retinal detachments and breaks, retinal vascular occlusions, retinopathy of prematurity, degeneration of macula and posterior pole, peripheral retinal degeneration, other hereditary retinal dystrophy, retinal hemorrhage, and separation of retinal layers that may impact the retina. Additionally, a touchscreen questionnaire was conducted at baseline assessment where participants were asked "Has a doctor told you that you have any of the following problems with your eyes?". Participants answering, "Diabetes related eye diseases" and/or "Macular

degeneration" were also excluded based on their responses. Moreover, participants who reported that they had any retina-related disease at the verbal interview were also excluded. This information has been added to **Supplementary Methods: Eligibility criteria**. We hope that this addresses your concern.

Comment #3. Typically, adjustment variables are derived from screening processes such as univariate analysis or based on disease research experience. In the first phase of analysis, the investigators increased the number of adjustment variables based on model I. I note that the adjustment variables in both model I and model II contain population characteristics. What is the rationale for adjusting the models in this order?

Response: Thank you for your helpful comment on our manuscript. We sincerely apologize for any confusion caused in our previous manuscript regarding the model adjusting.

We have revised the manuscript by following your suggestions on model adjusting. Specifically, we performed a univariate analysis on the potential covariates that may affect the association between RPE thickness and incident diabetes. These covariates included age, sex, ethnicity, assessment center, household income, Townsend deprivation index, education, smoking, drinking, body-mass index, hypertension, hyperlipidemia, intra-ocular pressure, spherical equivalent, and use of antihypertensive and lipid-lowering medication. Variables that reached a threshold of $P < 0.1$ were adjusted in the multivariate models, which included age, sex, ethnicity, household income, Townsend deprivation index, education, smoking, drinking, body-mass index, and use of antihypertensive and lipid-lowering medication.

We have added this information to the **Methods** section (**Page 18, Line 501**) of the revised manuscript, read as: *"The covariates included age, sex, ethnicity, assessment center, household income, Townsend deprivation index, education, smoking, drinking, BMI, hypertension, hyperlipidemia, IOP, SE, and use of antihypertensive and*

lipid-lowering medication. Variables that reached a threshold of $P < 0.1$ were adjusted in the multivariate models.” Please note that the direction and significance of all results remained unchanged. We hope that this revised version of our manuscript addresses your concern.

Comment #4. In the third stage of the analysis, the authors randomly divided the participants from Population-II into training and validation groups in a 1:1 ratio, for which we only provide suggestions that trying to increase the grouping of the training set sample ratio may yield better model training results.

Response: Thank you for your kind suggestion. We agree that increasing the training set sample ratio may yield better model training results. Thus, we followed your advice and reallocated all participants in the ratio of 7:3 as training and validation samples, respectively (*Methods* section, **Page 19, Line 519**). This adjustment led to improved model training results, with a C-statistic from 0.809 (95% CI: 0.798, 0.819) to 0.837 (95% CI: 0.827, 0.847). Furthermore, we conducted additional decision curve analysis (*Methods* section, **Page 19, Line 533**) that confirmed the clinical utility of incorporating these metabolic fingerprints for T2DM stratification (**Figure 3**). The summary result figure is provided below for your convenience. Hopefully this addresses your concern.

Figure 3

Incremental (A) predictability and (B) clinical utility of RPET-associated metabolic biomarkers for stratifying T2DM. RPET = retinal pigment epithelium thickness.

Comment #5. The study title "Metabolic Fingerprinting on Retinal Pigment Epithelium Thickness for Individualized Risk Prediction of Type 2 Diabetes Mellitus" might be inaccurate. Individualized risk prediction requires clear prediction results and evaluation criteria. We believe that the results of this study can only show that metabolomic fingerprinting can help improve individualized risk stratification in T2DM.

Response: Thank you for your valuable suggestion regarding the title of our manuscript. We apologize for any confusion caused by the previous title. We have taken your advice and revised the title to: "*Metabolic Fingerprinting on Retinal Pigment Epithelium Thickness for Individualized Risk Stratification of Type 2 Diabetes Mellitus*". Thank you again for helping us to improve the quality and accuracy of our research.

Comment #6. The main results of the study include the identification of metabolic fingerprints associated with reduced RPET and metabolic fingerprints to improve risk stratification, and I suggest expanding the discussion section to include validation of the results of metabolic fingerprint-related features or previous studies.

Response: Thank you for your valuable suggestion. We have conducted new experiments in an ethnically diverse cohort and expanded the discussion section.

In the revised manuscript, we conducted additional experiments employing LC/MS assay¹ in an independent cohort² to further investigate the value of RPET metabolic fingerprints for stratifying diabetic adverse microvascular phenotypes (**Methods** section, **Page 17, Line 461; Supplementary Methods: Experimental procedures, and quality control processes of metabolomic profiling in the GDES cohort**). Within this approach, an expanded range of RPET-associated metabolites were identified, particularly those present at low concentrations (**Supplementary Table S12**, as provided below for your convenience). Of these, all RPET-associated amino acids identified using NMR metabolomics in the UKB cohort were validated in the LC/MS

metabolomics assay used in the GDES cohort, although tyrosine did not show significance in the GDES cohort. NMR and LC/MS assays offer complementary coverage of metabolic markers, providing a more comprehensive RPET metabolic landscape. The incorporation of these metabolites into the clinical indicators-based model resulted in further improvements in discriminative power and clinical utility, further confirming the role of RPET metabolic fingerprints in facilitating risk stratification for T2DM-related damages (**Supplementary Figure S3**, as provided below for your convenience). Future metabolomics studies with higher coverage were poised to improve our system-level understanding of the link underlying RPE with diabetes risks.

We have also extended the discussion section by reviewing previous studies and discussing on the main results of this study (**Discussion** section, **Page 9, Line 236**). Alterations in lipid metabolism are deemed at the core of diabetes phenotypes³⁻⁴ and studies have indicated that insulin resistance can stimulate triglyceride synthesis and VLDL production, leading to an excessive accumulation of VLDL and LDL.⁵⁻⁷ Studies on NMR metabolomic profiling have also linked VLDL and LDL particle sizes and concentrations to an increased risk of T2DM.⁸⁻¹¹ These findings are in keeping with the adverse RPET metabolic state contributing to the increased T2DM risk observed in this study. Moreover, studies on retina have established that DR and its severity are associated with multiple VLDL and LDL particles, and lipid-lowering therapy has been shown to have significant benefits in preventing and mitigating DR.^{12,13} Modified lipid can have toxic effects on retinal cells, including RPE cells, and their excessive accumulation may result in oxidative stress and endoplasmic reticulum stress,¹⁴ potentially leading to dysfunction and structural damages of the RPE. Consistently, *in vitro* studies have demonstrated that RPE cells cultured under diabetic-like conditions produce elevated levels of reactive oxygen species compared to those incubated under non-diabetic conditions.¹⁵ All these lines of evidence indicate that a specific metabolic state that prioritizes damage to the retina before evident T2DM has already taken place, and while these subtle metabolic alterations

may not have a noticeable effect on the body yet, they are already detectable through in vivo OCT scanning and metabolomic profiling.

Thank you again for all your valuable feedback and suggestions. We appreciate your dedication and time in helping us improve the quality of this manuscript. Hopefully we have made the discussion more balanced and reasonable.

Reference

1. Chen W, Gong L, Guo Z, Wang W, Zhang H, Liu X, Yu S, Xiong L, Luo J. A novel integrated method for large-scale detection, identification, and quantification of widely targeted metabolites: application in the study of rice metabolomics. *Mol Plant*. 2013 Nov;6(6):1769-80. doi: 10.1093/mp/sst080. Epub 2013 May 23. PMID: 23702596.
2. Zhang S, Chen Y, Wang L, Li Y, Tang X, Liang X, He M, Wenyong H, Wang W; GDES group. Design and Baseline Data of the Diabetes Registration Study: Guangzhou Diabetic Eye Study. *Curr Eye Res*. 2023 Feb 27:1-9. doi: 10.1080/02713683.2023.2182745. Epub ahead of print. PMID: 36803011.
3. Kane JP, Pullinger CR, Goldfine ID, Malloy MJ. Dyslipidemia and diabetes mellitus: Role of lipoprotein species and interrelated pathways of lipid metabolism in diabetes mellitus. *Curr Opin Pharmacol*. 2021 Dec; 61:21-27. doi: 10.1016/j.coph.2021.08.013. Epub 2021 Sep 22. PMID: 34562838.
4. Eid S, Sas KM, Abcouwer SF, Feldman EL, Gardner TW, Pennathur S, Fort PE. New insights into the mechanisms of diabetic complications: role of lipids and lipid metabolism. *Diabetologia*. 2019 Sep;62(9):1539-1549. doi: 10.1007/s00125-019-4959-1. Epub 2019 Jul 25. PMID: 31346658; PMCID: PMC6679814.
5. Vergès B. Abnormal hepatic apolipoprotein B metabolism in type 2 diabetes. *Atherosclerosis*. 2010 Aug;211(2):353-60. doi: 10.1016/j.atherosclerosis.2010.01.028. Epub 2010 Jan 29. PMID: 20189175.
6. Annuzzi G, De Natale C, Iovine C, Patti L, Di Marino L, Coppola S, Del Prato S,

- Riccardi G, Rivellese AA. Insulin resistance is independently associated with postprandial alterations of triglyceride-rich lipoproteins in type 2 diabetes mellitus. *Arterioscler Thromb Vasc Biol.* 2004 Dec;24(12):2397-402. doi: 10.1161/01.ATV.0000146267.71816.30. Epub 2004 Sep 30. PMID: 15458975.
7. Malmström R, Packard CJ, Caslake M, Bedford D, Stewart P, Yki-Järvinen H, Shepherd J, Taskinen MR. Defective regulation of triglyceride metabolism by insulin in the liver in NIDDM. *Diabetologia.* 1997 Apr;40(4):454-62. doi: 10.1007/s001250050700. PMID: 9112023.
 8. Festa A, Williams K, Hanley AJ, Otvos JD, Goff DC, Wagenknecht LE, Haffner SM. Nuclear magnetic resonance lipoprotein abnormalities in prediabetic subjects in the Insulin Resistance Atherosclerosis Study. *Circulation.* 2005 Jun 28;111(25):3465-72. doi: 10.1161/CIRCULATIONAHA.104.512079. PMID: 15983261.
 9. Mora S, Otvos JD, Rosenson RS, Pradhan A, Buring JE, Ridker PM. Lipoprotein particle size and concentration by nuclear magnetic resonance and incident type 2 diabetes in women. *Diabetes.* 2010 May;59(5):1153-60. doi: 10.2337/db09-1114. Epub 2010 Feb 25. PMID: 20185808; PMCID: PMC2857895.
 10. Fizeleva M, Miilunpohja M, Kangas AJ, Soininen P, Kuusisto J, Ala-Korpela M, Laakso M, Stančáková A. Associations of multiple lipoprotein and apolipoprotein measures with worsening of glycemia and incident type 2 diabetes in 6607 non-diabetic Finnish men. *Atherosclerosis.* 2015 May;240(1):272-7. doi: 10.1016/j.atherosclerosis.2015.03.034. Epub 2015 Mar 23. PMID: 25818853.
 11. Ahola-Olli AV, Mustelin L, Kalimeri M, Kettunen J, Jokelainen J, Auvinen J, Puukka K, Havulinna AS, Lehtimäki T, Kähönen M, Juonala M, Keinänen-Kiukaanniemi S, Salomaa V, Perola M, Järvelin MR, Ala-Korpela M, Raitakari O, Würtz P. Circulating metabolites and the risk of type 2 diabetes: a prospective study of 11,896 young adults from four Finnish cohorts. *Diabetologia.* 2019 Dec;62(12):2298-2309. doi: 10.1007/s00125-019-05001-w. Epub 2019 Oct 4. PMID: 31584131; PMCID: PMC6861432.
 12. Lyons TJ, Jenkins AJ, Zheng D, Lackland DT, McGee D, Garvey WT, Klein RL.

Diabetic retinopathy and serum lipoprotein subclasses in the DCCT/EDIC cohort. *Invest Ophthalmol Vis Sci.* 2004 Mar;45(3):910-8. doi: 10.1167/iovs.02-0648. PMID: 14985310.

13. ACCORD Study Group; ACCORD Eye Study Group; Chew EY, Ambrosius WT, Davis MD, Danis RP, Gangaputra S, Greven CM, Hubbard L, Esser BA, Lovato JF, Perdue LH, Goff DC Jr, Cushman WC, Ginsberg HN, Elam MB, Genuth S, Gerstein HC, Schubart U, Fine LJ. Effects of medical therapies on retinopathy progression in type 2 diabetes. *N Engl J Med.* 2010 Jul 15;363(3):233-44. doi: 10.1056/NEJMoa1001288. Epub 2010 Jun 29. Erratum in: *N Engl J Med.* 2011 Jan 13;364(2):190. Erratum in: *N Engl J Med.* 2012 Dec 20;367(25):2458. PMID: 20587587; PMCID: PMC4026164.
14. Du M, Wu M, Fu D, Yang S, Chen J, Wilson K, Lyons TJ. Effects of modified LDL and HDL on retinal pigment epithelial cells: a role in diabetic retinopathy? *Diabetologia.* 2013 Oct;56(10):2318-28. doi: 10.1007/s00125-013-2986-x. Epub 2013 Jul 11. PMID: 23842729; PMCID: PMC4557884.
15. Tonade D, Kern TS. Photoreceptor cells and RPE contribute to the development of diabetic retinopathy. *Prog Retin Eye Res.* 2021 Jul; 83:100919. doi: 10.1016/j.preteyeres.2020.100919. Epub 2020 Nov 12. PMID: 33188897; PMCID: PMC8113320.

Supplementary Table S12

RPET-associated metabolites identified using LC/MS assay in the Guangzhou Diabetic Eye Study (GDES) cohort after adjusting for other factors.

Metabolites *	Group	β	95%CI		P_{raw}	P_{FDR}
γ -Glu-Met	Amino acid and Its metabolites	2.855	2.147	3.563	$<1.0 \times 10^{-8}$	$<1.0 \times 10^{-8}$
L-Arginine	Amino acid and Its metabolites	2.607	1.882	3.332	$<1.0 \times 10^{-8}$	$<1.0 \times 10^{-8}$
Tryptamine	Tryptamines,Cholines,Pigments	2.327	1.590	3.065	$<1.0 \times 10^{-8}$	2.72×10^{-7}
Val-Ala	Amino acid and Its metabolites	2.247	1.525	2.969	$<1.0 \times 10^{-8}$	3.57×10^{-7}
Met-Glu	Amino acid and Its metabolites	2.270	1.535	3.006	$<1.0 \times 10^{-8}$	4.07×10^{-7}
Piperidine	Heterocyclic compounds	2.314	1.532	3.096	1.08×10^{-8}	1.38×10^{-6}
Azelaic Acid	Organic acid and Its derivatives	2.091	1.366	2.816	2.44×10^{-8}	2.38×10^{-6}
Glu-Met	Amino acid and Its metabolites	2.112	1.375	2.850	3.00×10^{-8}	2.67×10^{-6}
N-Propionylglycine	Amino acid and Its metabolites	2.204	1.434	2.974	3.11×10^{-8}	2.72×10^{-6}
2-(Formylamino)Benzoic Acid	Benzene and substituted derivatives	1.989	1.258	2.721	1.39×10^{-7}	1.07×10^{-5}
2'-O-methylcytidine	Nucleotide and Its metabolites	1.965	1.231	2.698	2.11×10^{-7}	1.51×10^{-5}
3-Hydroxy-L-phenylalanine	Amino acid and Its metabolites	1.854	1.120	2.588	9.72×10^{-7}	6.37×10^{-5}
DL-O-tyrosine	Amino acid and Its metabolites	1.777	1.032	2.521	3.63×10^{-6}	2.19×10^{-4}
D-Alloisoleucine	Amino acid and Its metabolites	1.747	0.959	2.534	1.62×10^{-5}	8.48×10^{-4}
5,6-Dimethylbenzimidazole	Heterocyclic compounds	1.654	0.903	2.405	1.85×10^{-5}	9.45×10^{-4}
L-Methionine	Amino acid and Its metabolites	1.653	0.891	2.416	2.47×10^{-5}	1.19×10^{-3}
Indoleacetaldehyde	Heterocyclic compounds	1.574	0.829	2.319	3.95×10^{-5}	1.66×10^{-3}
9,10-Epoxy-18-hydroxyoctadecanoic acid	FA	1.531	0.801	2.261	4.53×10^{-5}	1.81×10^{-3}
Trp-Gly	Amino acid and Its metabolites	1.603	0.837	2.368	4.64×10^{-5}	1.84×10^{-3}
Glycolithocholic acid	Bile acids	1.515	0.782	2.248	5.75×10^{-5}	2.09×10^{-3}
2'-O-methyluridine	Nucleotide and Its metabolites	1.561	0.803	2.320	6.21×10^{-5}	2.18×10^{-3}
Melatonin	Hormones and hormone related compounds	1.534	0.788	2.279	6.23×10^{-5}	2.19×10^{-3}

N-Acetyl-L-Leucine	Amino acid and Its metabolites	1.658	0.849	2.467	6.62×10 ⁻⁵	2.26×10 ⁻³
gamma-Glu-Phe	Amino acid and Its metabolites	1.593	0.796	2.390	1.00×10 ⁻⁴	3.10×10 ⁻³
12,13-DiHOME	FA	1.453	0.722	2.184	1.09×10 ⁻⁴	3.27×10 ⁻³
9,10-DiHOME	FA	1.449	0.718	2.180	1.15×10 ⁻⁴	3.40×10 ⁻³
N-acetylorithine	Amino acid and Its metabolites	1.481	0.730	2.231	1.22×10 ⁻⁴	3.54×10 ⁻³
Gly-Phe	Amino acid and Its metabolites	1.437	0.699	2.174	1.49×10 ⁻⁴	4.14×10 ⁻³
Glu-Tyr	Amino acid and Its metabolites	1.462	0.705	2.220	1.70×10 ⁻⁴	4.58×10 ⁻³
Phe-Hyp	Amino acid and Its metabolites	1.410	0.673	2.146	1.94×10 ⁻⁴	5.04×10 ⁻³
Phe-Phe	Amino acid and Its metabolites	1.417	0.670	2.165	2.23×10 ⁻⁴	5.55×10 ⁻³
Pro-Asp	Amino acid and Its metabolites	1.398	0.658	2.138	2.33×10 ⁻⁴	5.71×10 ⁻³
Glu-Leu	Amino acid and Its metabolites	1.455	0.676	2.234	2.73×10 ⁻⁴	6.40×10 ⁻³
Phe-Gly	Amino acid and Its metabolites	1.385	0.641	2.129	2.86×10 ⁻⁴	6.60×10 ⁻³
L-Lysine	Amino acid and Its metabolites	1.360	0.624	2.096	3.20×10 ⁻⁴	7.16×10 ⁻³
L-Serine	Amino acid and Its metabolites	1.383	0.629	2.137	3.53×10 ⁻⁴	7.69×10 ⁻³
N-Amidino-L-Aspartate	Amino acid and Its metabolites	1.356	0.609	2.103	4.04×10 ⁻⁴	8.42×10 ⁻³
Multifidol	Benzene and substituted derivatives	1.334	0.594	2.073	4.40×10 ⁻⁴	8.90×10 ⁻³
Urocanic Acid	Organic acid and Its derivatives	1.338	0.595	2.080	4.46×10 ⁻⁴	8.98×10 ⁻³
PI (15:0/2:0)	GP	1.321	0.577	2.064	5.35×10 ⁻⁴	1.01×10 ⁻²
Caffeic Acid	Organic acid and Its derivatives	1.303	0.570	2.037	5.37×10 ⁻⁴	1.01×10 ⁻²
Hydroxyurea	Others	1.310	0.572	2.049	5.41×10 ⁻⁴	1.01×10 ⁻²
5'-Deoxy-5'-(Methylthio) Adenosine	Nucleotide and Its metabolites	1.318	0.556	2.080	7.45×10 ⁻⁴	1.32×10 ⁻²
Quinmerac	Benzene and substituted derivatives	-1.460	-2.306	-0.614	7.66×10 ⁻⁴	1.35×10 ⁻²
Proline betaine	Amino acid and Its metabolites	1.278	0.528	2.028	8.89×10 ⁻⁴	1.52×10 ⁻²
L-Ornithine	Amino acid and Its metabolites	1.256	0.519	1.994	8.96×10 ⁻⁴	1.53×10 ⁻²
L-Glutamine	Amino acid and Its metabolites	1.243	0.507	1.980	9.98×10 ⁻⁴	1.66×10 ⁻²
N, N-Bis(2-hydroxyethyl) dodecanamide	FA	1.238	0.496	1.979	1.14×10 ⁻³	1.84×10 ⁻²

Trolox	Benzene and substituted derivatives	1.235	0.491	1.978	1.20×10^{-3}	1.91×10^{-2}
Asp-Leu	Amino acid and Its metabolites	1.220	0.482	1.958	1.27×10^{-3}	2.00×10^{-2}
4-Acetylamino benzoic acid	Organic acid and Its derivatives	1.205	0.469	1.940	1.40×10^{-3}	2.15×10^{-2}
N-(2-hydroxyethyl)-3-pyridinecarboxamide	Heterocyclic compounds	1.307	0.502	2.112	1.53×10^{-3}	2.31×10^{-2}
Creatine phosphate	Nucleotide and Its metabolites	1.194	0.446	1.943	1.86×10^{-3}	2.71×10^{-2}
Phe-Val	Amino acid and Its metabolites	1.197	0.446	1.948	1.87×10^{-3}	2.72×10^{-2}
2-Aminophenol	Benzene and substituted derivatives	1.157	0.421	1.892	2.14×10^{-3}	3.06×10^{-2}
Hippuric Acid	Organic acid and Its derivatives	1.120	0.385	1.855	2.95×10^{-3}	4.01×10^{-2}
L-Asparagine Anhydrous	Amino acid and Its metabolites	1.162	0.394	1.929	3.14×10^{-3}	4.21×10^{-2}
Phenethylamine	Alcohol and amines	1.175	0.384	1.966	3.75×10^{-3}	4.85×10^{-2}

* Adjusted for age, sex duration of diabetes, HbA1c, body-mass index, systolic blood pressure, smoking, drinking, and hyperlipidemia.

Supplementary Figure S3

Added discriminative power and clinical utility of RPET metabolic fingerprints for stratifying T2DM adverse microvascular phenotypes in GDES.

Replies to Reviewer #3:

Comment #1. This paper provide evidence that the reduced RPE thickness (RPET) is a predictor of type 2 diabetes development. In addition, the authors found that serum metabolic fingerprints (mainly lipid-related) associated with RPET were also predictors of future type 2 diabetes development beyond the traditional risk factors. This is a well-written and well-designed paper.

Response: Thank you for your positive feedback and the kind words you provided regarding our manuscript. We are thrilled to hear that you found our work to be well-written and well-designed. We would also like to express our gratitude for the time and effort you dedicated to reviewing our manuscript. Your suggestions and comments have been immensely helpful in improving the quality of our research. We are genuinely grateful for your contributions to our work.

Comment #2. Major point - There authors should give some explanation regarding the potential reasons why the metabolites (all lipid-related) detected in serum are related to the RPE thinning, as well as their role in the pathogenesis of type 2 diabetes.

Response: Thank you for providing valuable suggestions. We have expanded the *Discussion* section and included explanations regarding the potential reasons underlying our findings (**Page 9, Line 236**).

Alterations in lipid metabolism are deemed at the core of diabetes phenotypes¹⁻² and studies have indicated that insulin resistance can stimulate triglyceride synthesis and VLDL production, leading to an excessive accumulation of VLDL and LDL.³⁻⁵ Studies on NMR metabolomic profiling have also linked VLDL and LDL particle sizes and concentrations to an increased risk of T2DM.⁶⁻⁹ These findings are consistent with the notion that the adverse metabolic state associated with RPET contributes to the increased risk of type 2 diabetes mellitus observed in this study. Moreover, studies on retina have established that DR and its severity are associated with multiple VLDL and LDL particles, and lipid-lowering therapy has been shown to have significant

benefits in preventing and mitigating DR.^{10,11} Modified lipid can have toxic effects on retinal cells, including RPE cells, and their excessive accumulation may result in oxidative stress and endoplasmic reticulum stress,¹² potentially leading to dysfunction and structural damages of the RPE. Consistently, *in vitro* studies have demonstrated that RPE cells cultured under diabetic-like conditions produce elevated levels of reactive oxygen species compared to those incubated under non-diabetic conditions.¹³ All these lines of evidence indicate that a specific metabolic state that prioritizes damage to the retina before evident T2DM has already taken place, and while these subtle metabolic alterations may not have a noticeable effect on the body yet, they are already detectable through *in vivo* OCT scanning and metabolomic profiling.

Reference

1. Kane JP, Pullinger CR, Goldfine ID, Malloy MJ. Dyslipidemia and diabetes mellitus: Role of lipoprotein species and interrelated pathways of lipid metabolism in diabetes mellitus. *Curr Opin Pharmacol*. 2021 Dec; 61:21-27. doi: 10.1016/j.coph.2021.08.013. Epub 2021 Sep 22. PMID: 34562838.
2. Eid S, Sas KM, Abcouwer SF, Feldman EL, Gardner TW, Pennathur S, Fort PE. New insights into the mechanisms of diabetic complications: role of lipids and lipid metabolism. *Diabetologia*. 2019 Sep;62(9):1539-1549. doi: 10.1007/s00125-019-4959-1. Epub 2019 Jul 25. PMID: 31346658; PMCID: PMC6679814.
3. Vergès B. Abnormal hepatic apolipoprotein B metabolism in type 2 diabetes. *Atherosclerosis*. 2010 Aug;211(2):353-60. doi: 10.1016/j.atherosclerosis.2010.01.028. Epub 2010 Jan 29. PMID: 20189175.
4. Annuzzi G, De Natale C, Iovine C, Patti L, Di Marino L, Coppola S, Del Prato S, Riccardi G, Rivellese AA. Insulin resistance is independently associated with postprandial alterations of triglyceride-rich lipoproteins in type 2 diabetes mellitus. *Arterioscler Thromb Vasc Biol*. 2004 Dec;24(12):2397-402. doi: 10.1161/01.ATV.0000146267.71816.30. Epub 2004 Sep 30. PMID: 15458975.

5. Malmström R, Packard CJ, Caslake M, Bedford D, Stewart P, Yki-Järvinen H, Shepherd J, Taskinen MR. Defective regulation of triglyceride metabolism by insulin in the liver in NIDDM. *Diabetologia*. 1997 Apr;40(4):454-62. doi: 10.1007/s001250050700. PMID: 9112023.
6. Festa A, Williams K, Hanley AJ, Otvos JD, Goff DC, Wagenknecht LE, Haffner SM. Nuclear magnetic resonance lipoprotein abnormalities in prediabetic subjects in the Insulin Resistance Atherosclerosis Study. *Circulation*. 2005 Jun 28;111(25):3465-72. doi: 10.1161/CIRCULATIONAHA.104.512079. PMID: 15983261.
7. Mora S, Otvos JD, Rosenson RS, Pradhan A, Buring JE, Ridker PM. Lipoprotein particle size and concentration by nuclear magnetic resonance and incident type 2 diabetes in women. *Diabetes*. 2010 May;59(5):1153-60. doi: 10.2337/db09-1114. Epub 2010 Feb 25. PMID: 20185808; PMCID: PMC2857895.
8. Fizeleva M, Miilunpohja M, Kangas AJ, Soinen P, Kuusisto J, Ala-Korpela M, Laakso M, Stančáková A. Associations of multiple lipoprotein and apolipoprotein measures with worsening of glycemia and incident type 2 diabetes in 6607 non-diabetic Finnish men. *Atherosclerosis*. 2015 May;240(1):272-7. doi: 10.1016/j.atherosclerosis.2015.03.034. Epub 2015 Mar 23. PMID: 25818853.
9. Ahola-Olli AV, Mustelin L, Kalimeri M, Kettunen J, Jokelainen J, Auvinen J, Puukka K, Havulinna AS, Lehtimäki T, Kähönen M, Juonala M, Keinänen-Kiukaanniemi S, Salomaa V, Perola M, Järvelin MR, Ala-Korpela M, Raitakari O, Würtz P. Circulating metabolites and the risk of type 2 diabetes: a prospective study of 11,896 young adults from four Finnish cohorts. *Diabetologia*. 2019 Dec;62(12):2298-2309. doi: 10.1007/s00125-019-05001-w. Epub 2019 Oct 4. PMID: 31584131; PMCID: PMC6861432.
10. Lyons TJ, Jenkins AJ, Zheng D, Lackland DT, McGee D, Garvey WT, Klein RL. Diabetic retinopathy and serum lipoprotein subclasses in the DCCT/EDIC cohort. *Invest Ophthalmol Vis Sci*. 2004 Mar;45(3):910-8. doi: 10.1167/iovs.02-0648. PMID: 14985310.
11. ACCORD Study Group; ACCORD Eye Study Group; Chew EY, Ambrosius WT, Davis

MD, Danis RP, Gangaputra S, Greven CM, Hubbard L, Esser BA, Lovato JF, Perdue LH, Goff DC Jr, Cushman WC, Ginsberg HN, Elam MB, Genuth S, Gerstein HC, Schubart U, Fine LJ. Effects of medical therapies on retinopathy progression in type 2 diabetes. *N Engl J Med.* 2010 Jul 15;363(3):233-44. doi: 10.1056/NEJMoa1001288. Epub 2010 Jun 29. Erratum in: *N Engl J Med.* 2011 Jan 13;364(2):190. Erratum in: *N Engl J Med.* 2012 Dec 20;367(25):2458. PMID: 20587587; PMCID: PMC4026164.

12. Du M, Wu M, Fu D, Yang S, Chen J, Wilson K, Lyons TJ. Effects of modified LDL and HDL on retinal pigment epithelial cells: a role in diabetic retinopathy? *Diabetologia.* 2013 Oct;56(10):2318-28. doi: 10.1007/s00125-013-2986-x. Epub 2013 Jul 11. PMID: 23842729; PMCID: PMC4557884.
13. Tonade D, Kern TS. Photoreceptor cells and RPE contribute to the development of diabetic retinopathy. *Prog Retin Eye Res.* 2021 Jul;83:100919. doi: 10.1016/j.preteyeres.2020.100919. Epub 2020 Nov 12. PMID: 33188897; PMCID: PMC8113320.

Comment #3. Minor point - Please specify the number of patients in whom the metabolomic study was performed.

Response: Thank you for bringing to our attention the missing information in our previous manuscript. We apologize for any confusion caused by this oversight. In brief, a total of 110,730 UKB participants completed metabolomic profiling, and among them, 14,662 participants also underwent optical coherence tomography scanning.

After excluding eyes with missing thickness values (294 right eyes and 407 left eyes), low signal strength (1,272 right eyes and 1,067 left eyes), poor centration or segmentation (4,291 right eyes and 4,309 left eyes), and participants with certain conditions such as high refractive error (341 right eyes and 404 left eyes), visual impairment (2,054 right eyes and 2,014 left eyes), abnormal intraocular pressure

(701 right eyes and 752 left eyes), glaucoma (82 participants), other retinal disorders (162 participants), and neurodegenerative diseases (39 participants), a total of 7,824 participants were included in population-I.

In population-II, we excluded 18,032 participants due to lacking hospital records, and an additional 6,684 participants who were already included in population-I.

Participants with baseline T2DM were also excluded, resulting in a total of 84,224 participants in population-II.

We have added this information to the *Supplementary Methods: Eligibility criteria* section. We hope that this clarification will properly reflect our study and address your concern.

Thank you once again for providing valuable feedback and suggestions. We appreciate the dedication and time you have invested in helping us enhance the quality of this manuscript.

Replies to Reviewer #4:

Comment #1. Herein Yang et al. conducted a large population-based cohort study and found reduced retinal pigment epithelium thickness (RPET) drastically increased the risk of onset and development of type 2 diabetes mellitus (T2DM) after adjusting for covariates like age, sex, ethnicity, income, education, smoking and others. Then they identified several metabolite biomarkers for RPET via an NMR-based plasma metabolomics method. Furthermore, meta-RPET score calculated from RPET-related metabolomics changes improved T2DM predictability and achieved individualized risk stratification for T2DM. Overall, this study is an observational study, but lack of mechanisms. The causal association between lipids/lipoproteins and RPET remains obscure. Moreover, I am also worried about the quantitative analysis of so many lipoprotein parameters using NMR spectroscopy. Therefore, I do not recommend this

work to be published in Nature Communications. Major comments or suggestions as follows:

Response: We would like to express our sincere gratitude for your constructive feedback and valuable suggestions, which have significantly contributed to improving our manuscript. We have carefully considered all your specific comments and made substantial revisions accordingly. We hope that these changes have adequately addressed your concerns and enhanced the overall quality of our work.

Changes in the retina have long been regarded as a window to systemic health,¹⁻³ and our results suggest that alterations in the RPE may open up a new avenue for further unraveling diabetes. Our observational study revealed that reduced RPE thickness significantly increases the risk of future type 2 diabetes mellitus (T2DM). Furthermore, incorporating RPE metabolomic fingerprints improved risk stratification for T2DM. While we speculate that systemic metabolic alterations may contribute to altered RPE, we do not propose that the biological changes captured by NMR metabolomics can fully or predominantly elucidate the underpinnings of the association between RPE and diabetes. Rather, our study established a framework that suggests the possibility that RPE alterations, detectable up to a decade before the onset of T2DM, could potentially be involved in the early pathogenesis of T2DM, along with other biological changes that trigger or coincide with RPE alterations, beyond the metabolic changes captured by NMR metabolomics. Our findings on RPE metabolomics emphasize the significance of RPE-associated biological changes in the early pathogenesis of T2DM, which could potentially indicate T2DM risk that conventional risk factors fail to quantify. Importantly, our additional decision curve analyses confirm the clinical utility of these metabolic changes across a wide range of medical decision threshold scenarios.

Thank you for your comments on the limited sensitivity of NMR metabolomics assays and its ability to provide only a partial insight into RPE metabolic alterations, though it has been well-established as a robust and reliable tool for large-scale

population-based studies on diabetes.⁴ To address this issue, we conducted additional experiments using the more sensitive LC/MS metabolomic assay⁵ in an ethnically diverse cohort⁶ to further investigate the value of RPE metabolomics for stratifying diabetic microvascular phenotypes. NMR and LC/MS assays in this study offer complementary coverage of metabolic markers, providing a more comprehensive RPE metabolic landscape.⁷ The incorporation of these metabolites into the clinical indicator-based model led to further enhancements in discriminative power and clinical utility, providing additional evidence for the role of RPE metabolic fingerprints in facilitating risk stratification for T2DM-related outcomes.

We have conducted additional pathway analysis to gain a better understanding of the biological implications of these metabolite changes. Furthermore, we have expanded the discussion section and proposed some speculations on the biological underpinnings of the relationship between RPE and diabetes. Additionally, we have provided more detailed information regarding the methodology of experimental procedures and quality control processes for metabolomic profiling in the UKB cohort.⁸ To further confirm the reliability of the Nightingale NMR platform metabolite quantification, we have also compared the results with clinical chemistry and mass spectrometry.

Point-by-point responses to more specific comments are provided below. We appreciate your critical evaluation and have carefully reviewed the relevant literature to make our opinion on the issue more balanced in the revised manuscript. We have made significant revisions and hope that they meet your expectations.

Reference

1. Cheung CY, Ikram MK, Chen C, Wong TY. Imaging retina to study dementia and stroke. *Prog Retin Eye Res.* 2017 Mar; 57:89-107. doi: 10.1016/j.preteyeres.2017.01.001. Epub 2017 Jan 3. PMID: 28057562.
2. London A, Benhar I, Schwartz M. The retina as a window to the brain-from eye

- research to CNS disorders. *Nat Rev Neurol*. 2013 Jan;9(1):44-53. doi: 10.1038/nrneurol.2012.227. Epub 2012 Nov 20. PMID: 23165340.
3. Flammer J, Konieczka K, Bruno RM, Virdis A, Flammer AJ, Taddei S. The eye and the heart. *Eur Heart J*. 2013 May;34(17):1270-8. doi: 10.1093/eurheartj/ehs023. Epub 2013 Feb 10. PMID: 23401492; PMCID: PMC3640200.
 4. Jin Q, Ma RCW. Metabolomics in Diabetes and Diabetic Complications: Insights from Epidemiological Studies. *Cells*. 2021 Oct 21;10(11):2832. doi: 10.3390/cells10112832. PMID: 34831057; PMCID: PMC8616415.
 5. Chen W, Gong L, Guo Z, Wang W, Zhang H, Liu X, Yu S, Xiong L, Luo J. A novel integrated method for large-scale detection, identification, and quantification of widely targeted metabolites: application in the study of rice metabolomics. *Mol Plant*. 2013 Nov;6(6):1769-80. doi: 10.1093/mp/sst080. Epub 2013 May 23. PMID: 23702596.
 6. Zhang S, Chen Y, Wang L, Li Y, Tang X, Liang X, He M, Wenyong H, Wang W; GDES group. Design and Baseline Data of the Diabetes Registration Study: Guangzhou Diabetic Eye Study. *Curr Eye Res*. 2023 Feb 27:1-9. doi: 10.1080/02713683.2023.2182745. Epub ahead of print. PMID: 36803011.
 7. Soininen P, Kangas AJ, Würtz P, Suna T, Ala-Korpela M. Quantitative serum nuclear magnetic resonance metabolomics in cardiovascular epidemiology and genetics. *Circ Cardiovasc Genet*. 2015 Feb;8(1):192-206. doi: 10.1161/CIRCGENETICS.114.000216. PMID: 25691689.
 8. Julkunen H, Cichońska A, Tiainen M, Koskela H, Nybo K, Mäkelä V, Nokso-Koivisto J, Kristiansson K, Perola M, Salomaa V, Jousilahti P, Lundqvist A, Kangas AJ, Soininen P, Barrett JC, Würtz P. Atlas of plasma NMR biomarkers for health and disease in 118,461 individuals from the UK Biobank. *Nat Commun*. 2023 Feb 3;14(1):604. doi: 10.1038/s41467-023-36231-7. PMID: 36737450; PMCID: PMC9898515.

Comment #2. The absolute quantification of lipoprotein parameters is of great

importance in this study, but the detailed key information is insufficient. Although the authors cited the detailed protocol on quantitative serum NMR metabolomics, plasma sample was used in this study, so which type of anticoagulants was used? For example, EDTA seriously affects NMR metabolic signals. In addition, the main NMR parameters should be provided for users.

Response: Thank you for pointing out this important issue. We apologized for the insufficient information on NMR metabolomics in the UKB cohort. In response to your comment, we have now included detailed information on plasma sample preparation, NMR spectroscopy, quality control processes, technical and biological repeatability, and other plasma specific issues¹, in **Supplementary Methods**.

Absolute concentrations of 168 biomarkers and 81 biomarker ratios were quantified by NMR spectroscopy (Nightingale Health Plc.) from plasma samples (UK Biobank aliquot 3) of over 100,000 randomly selected UKB participants, as previously described in *Nature*, *Nature Medicine*, *Nature Communications*, *Lancet Regional Health*, *eLife*, and *BMC Medicine*.²⁻⁷ The UK Biobank laboratory in Stockport, UK prepared EDTA plasma samples from aliquot 3 in 96-well plates. TECAN freedom EVO 150 robotic liquid handlers were used to aliquot at least 90 μ L of plasma into each well, with coefficients of variation (CV) in pipetting volume at <0.75% across 8 tips. The 96-well plates containing plasma samples were shipped on dry ice in batches of 5,000-20,000 samples to Nightingale Health's laboratories in Finland. No selection criteria were applied to the sampling process. As a result, the participants underwent metabolomic profiling are a random subset of the full UKB cohort.

Upon arrival at Nightingale Health laboratories, the samples were immediately stored in a freezer at -80°C. Before preparation, the frozen samples were slowly thawed overnight at +4°C, and then gently mixed and centrifuged for 3 minutes at 3400'g and +4°C to remove any possible precipitate. Aliquots of each sample were transferred into 3-mm outer-diameter NMR tubes and mixed with a phosphate buffer (containing 75mM Na₂HPO₄ in 80%/20% H₂O/D₂O, pH 7.4, and including

0.08% sodium 3(trimethylsilyl) propionate-2,2,3,3-d₄ and 0.04% sodium azide) in a 1:1 ratio using an automated liquid handler (PerkinElmer Janus Automated Workstation).

The NMR spectrometers used to measure the plasma samples conducted measurements in a blinded manner, prior to linkage with the UK Biobank health outcomes. The prepared plasma samples in 96-well plates were loaded onto a cooled sample changer that maintained their temperature at +6°C while waiting to be measured. Two NMR spectra were recorded for each plasma sample using 500 MHz NMR spectrometers (Bruker AVANCE IIIHD). The first spectrum was a presaturated proton spectrum that mainly featured resonances arising from proteins and lipids within various lipoprotein particles. In contrast, the other spectrum was a Carr-Purcell-Meiboom-Gill T₂-relaxation-filtered spectrum designed to suppress most of the broad macromolecule and lipoprotein lipid signals, leading to enhanced detection of low-molecular-weight metabolites.

Nightingale Health's proprietary software (quantification library 2020) was used to quantify the biomarkers. This allowed for the simultaneous quantification of 249 metabolic measures per EDTA plasma sample, as broken down into 168 absolute level measures and 81 ratio measures. All biomarkers in this study have a known identity. The biomarker panel includes routine lipids, lipoprotein subclass profiling with lipid concentrations within 14 subclasses, fatty acid composition, and various low-molecular weight metabolites such as amino acids, ketone bodies, and glycolysis metabolites quantified in molar concentration units. For each of the 14 lipoprotein subclasses, lipid concentrations and composition are measured for triglycerides, phospholipids, total cholesterol, cholesterol esters, and free cholesterol, as well as the total lipid concentration within each subclass. Most of the biomarkers are measured in absolute concentration units (mmol/L). For clinical translation applications, 37 biomarkers in the panel have been certified for diagnostic use (CE-marked). The average detection rate across all plasma samples was greater than

99%.

Pre-specified metrics on the biomarker consistency were agreed between UK Biobank and Nightingale Health to ensure the quality of results throughout the project. To guarantee the accuracy and consistency of results across multiple spectrometers during the project, two internal control samples provided by Nightingale Health were included in each 96-well plate. Four sets of internal control samples, each with different biomarker concentration spans, were interleaved between the NMR instruments over the 1,352 96-well plates that were measured. This quality control process was conducted continuously over an extended duration of the project. An example of such continuous quality control is illustrated for leucine (**Appendix Figure 1**, as provided below for your convenience). Different colors show results for four control samples that are measured interleaved in NMR instruments during the project course. Dashed and full-blown lines indicate results from two different NMR instruments.

Each 96-well plate included two blind duplicate samples provided by the UKB, whose position information was only unlocked after interim results were delivered to the UKB. The distribution of coefficients of variation (CV) across the biomarker measures were measured, both for the UKB 's blind duplicates and Nightingale Health's internal control samples. For most biomarkers in both cases, the CVs are below 5%. These results satisfied the pre-specified CV targets across the biomarker measures for each set of approximately 20,000 consecutively measured samples. Prior studies on a smaller scale have reported representative CVs for NMR biomarkers' blind duplicate samples and repeat control samples.^{8,9}

The technical consistency of measurements across consecutive shipment batches and different NMR spectrometers is demonstrated for all NMR biomarkers in the UK Biobank data resource, which can be accessed at <https://biobank.ndph.ox.ac.uk/showcase/label.cgi?id=220>. This resource also

displays the correlation of blinded duplicate samples for each biomarker and the biological consistency in repeat-visit samples drawn from the same individuals four years apart. The technical and biological repeatability assessments are depicted for GlycA in **Appendix Figure 2**. The technical consistency over time is demonstrated through the distributions of consecutive batches of sample shipments (first panel) and different spectrometers (second panel). The technical repeatability is also displayed in the form of approximately 650 blind duplicate samples' consistency (third panel), resulting in a between-instrument CV of 3%. Furthermore, the biological repeatability for measurements from blood samples collected from the same individuals around 4 years apart is exhibited for roughly 1,300 samples (fourth panel).

The Nightingale Health NMR platform incorporates integrated quality procedures to identify signs of degradation and contamination issues in each plasma sample. These issues are reported as flags alongside the biomarker concentration result data. Issues that affect the entire sample are reported as sample-level flags, while issues that only affect certain biomarkers are reported as biomarker-level flags, provided as a separate data field for each biomarker. Generally, if a biomarker has a flag but the value is still provided, it indicates that the presence of the interfering substance is low and deemed not to interfere with the quantification of the biomarker (i.e., the value can be trusted). Thus, no biomarker values need to be removed a priori based on the flags.¹

Dilution issues may be present in the UKB blood samples, which are known to undergo unintended dilution during the initial sample storage process at the UKB facilities. Previous reports have suggested that samples taken from aliquot 3, used in NMR measurements, may experience 5-10% dilution.¹⁰ This dilution is thought to be caused by the mixing of participant samples with water due to seals that failed to maintain a system vacuum in the automated liquid handling systems. While this issue is likely to impact some of the absolute biomarker concentration values, it is not

expected to have a significant effect on epidemiological analyses.¹ Biomarker values that substantially affected by interfering substances have been removed during the quality control procedures.

An independent study analyzed sources of technical variation in the UKB NMR biomarker data and performed post-measurement quality control.¹¹ The study found that the spectrometer used and time from plasma preparation to measurement were the only notable sources of variation. For most biomarkers, each factor explained 1-3% of variation, with only amino acids histidine and alanine showing substantially higher levels of technical variation. Regressing out or adjusting for spectrometer as a factor in epidemiological analyses may provide slight power gains for genome-wide association analyses, but the impact on biomarker-disease associations is minor.¹ This study¹¹ also identified a small number of outlier plates with deviating concentrations across many biomarkers, which was deemed to arise from the UKB 's sample plating process. A median of nine outlier plates were identified across the biomarkers, with a maximum of 20 for albumin. However, since only ~1% of the samples were affected, the impact on epidemiological associations is modest.¹

This information has been added to the ***Supplementary Methods: Experimental procedures, and quality control processes of metabolomic profiling in the UKB cohort.*** Hopefully this addresses your concerns.

Reference

1. Julkunen H, Cichońska A, Tiainen M, Koskela H, Nybo K, Mäkelä V, Nokso-Koivisto J, Kristiansson K, Perola M, Salomaa V, Jousilahti P, Lundqvist A, Kangas AJ, Soininen P, Barrett JC, Würtz P. Atlas of plasma NMR biomarkers for health and disease in 118,461 individuals from the UK Biobank. *Nat Commun.* 2023 Feb 3;14(1):604. doi: 10.1038/s41467-023-36231-7. PMID: 36737450; PMCID: PMC9898515.
2. Xu Y, Ritchie SC, Liang Y, Timmers PRHJ, Pietzner M, Lannelongue L, Lambert SA,

Tahir UA, May-Wilson S, Foguet C, Johansson Å, Surendran P, Nath AP, Persyn E, Peters JE, Oliver-Williams C, Deng S, Prins B, Luan J, Bombá L, Soranzo N, Di Angelantonio E, Pirastu N, Tai ES, van Dam RM, Parkinson H, Davenport EE, Paul DS, Yau C, Gerszten RE, Mälarstig A, Danesh J, Sim X, Langenberg C, Wilson JF, Butterworth AS, Inouye M. An atlas of genetic scores to predict multi-omic traits. *Nature*. 2023 Apr;616(7955):123-131. doi: 10.1038/s41586-023-05844-9. Epub 2023 Mar 29. PMID: 36991119.

3. Buergel T, Steinfeldt J, Ruyoga G, Pietzner M, Bizzarri D, Vojinovic D, Upmeier Zu Belzen J, Look L, Kittner P, Christmann L, Hollmann N, Strangalies H, Braunger JM, Wild B, Chiesa ST, Spranger J, Klostermann F, van den Akker EB, Trompet S, Mooijaart SP, Sattar N, Jukema JW, Lavrijssen B, Kavousi M, Ghanbari M, Ikram MA, Slagboom E, Kivimaki M, Langenberg C, Deanfield J, Eils R, Landmesser U. Metabolomic profiles predict individual multidisease outcomes. *Nat Med*. 2022 Nov;28(11):2309-2320. doi: 10.1038/s41591-022-01980-3. Epub 2022 Sep 22. PMID: 36138150; PMCID: PMC9671812.
4. Bell JA, Richardson TG, Wang Q, Sanderson E, Palmer T, Walker V, O'Keeffe LM, Timpson NJ, Cichonska A, Julkunen H, Würtz P, Holmes MV, Davey Smith G. Effects of general and central adiposity on circulating lipoprotein, lipid, and metabolite levels in UK Biobank: A multivariable Mendelian randomization study. *Lancet Reg Health Eur*. 2022 Jul 6; 21:100457. doi: 10.1016/j.lanepe.2022.100457. PMID: 35832062; PMCID: PMC9272390.
5. Smith CJ, Sinnott-Armstrong N, Cichońska A, Julkunen H, Fauman EB, Würtz P, Pritchard JK. Integrative analysis of metabolite GWAS illuminates the molecular basis of pleiotropy and genetic correlation. *Elife*. 2022 Sep 8;11: e79348. doi: 10.7554/eLife.79348. PMID: 36073519; PMCID: PMC9536840.
6. Julkunen H, Cichońska A, Slagboom PE, Würtz P; Nightingale Health UK Biobank Initiative. Metabolic biomarker profiling for identification of susceptibility to severe pneumonia and COVID-19 in the general population. *Elife*. 2021 May 4;10: e63033. doi: 10.7554/eLife.63033. PMID: 33942721; PMCID: PMC8172246.
7. Zhang X, Hu W, Wang Y, Wang W, Liao H, Zhang X, Kiburg KV, Shang X, Bulloch G,

Huang Y, Zhang X, Tang S, Hu Y, Yu H, Yang X, He M, Zhu Z. Plasma metabolomic profiles of dementia: a prospective study of 110,655 participants in the UK Biobank. *BMC Med.* 2022 Aug 15;20(1):252. doi: 10.1186/s12916-022-02449-3. PMID: 35965319; PMCID: PMC9377110.

8. Holmes MV, Millwood IY, Kartsonaki C, Hill MR, Bennett DA, Boxall R, Guo Y, Xu X, Bian Z, Hu R, Walters RG, Chen J, Ala-Korpela M, Parish S, Clarke RJ, Peto R, Collins R, Li L, Chen Z; China Kadoorie Biobank Collaborative Group. Lipids, Lipoproteins, and Metabolites and Risk of Myocardial Infarction and Stroke. *J Am Coll Cardiol.* 2018 Feb 13;71(6):620-632. doi: 10.1016/j.jacc.2017.12.006. PMID: 29420958; PMCID: PMC5811927.
9. Kettunen J, Demirkan A, Würtz P, Draisma HH, Haller T, Rawal R, Vaarhorst A, Kangas AJ, Lyytikäinen LP, Pirinen M, Pool R, Sarin AP, Soininen P, Tukiainen T, Wang Q, Tiainen M, Tynkkynen T, Amin N, Zeller T, Beekman M, Deelen J, van Dijk KW, Esko T, Hottenga JJ, van Leeuwen EM, Lehtimäki T, Mihailov E, Rose RJ, de Craen AJ, Gieger C, Kähönen M, Perola M, Blankenberg S, Savolainen MJ, Verhoeven A, Viikari J, Willemsen G, Boomsma DI, van Duijn CM, Eriksson J, Jula A, Järvelin MR, Kaprio J, Metspalu A, Raitakari O, Salomaa V, Slagboom PE, Waldenberger M, Ripatti S, Ala-Korpela M. Genome-wide study for circulating metabolites identifies 62 loci and reveals novel systemic effects of LPA. *Nat Commun.* 2016 Mar 23; 7:11122. doi: 10.1038/ncomms11122. PMID: 27005778; PMCID: PMC4814583.
10. Allen NE, Arnold M, Parish S, Hill M, Sheard S, Callen H, Fry D, Moffat S, Gordon M, Welsh S, Elliott P, Collins R. Approaches to minimising the epidemiological impact of sources of systematic and random variation that may affect biochemistry assay data in UK Biobank. *Wellcome Open Res.* 2021 Jan 4; 5:222. doi: 10.12688/wellcomeopenres.16171.2. PMID: 33364437; PMCID: PMC7739095.
11. Ritchie SC, Surendran P, Karthikeyan S, Lambert SA, Bolton T, Pennells L, Danesh J, Di Angelantonio E, Butterworth AS, Inouye M. Quality control and removal of technical variation of NMR metabolic biomarker data in ~120,000 UK Biobank

participants. Sci Data. 2023 Jan 31;10(1):64. doi: 10.1038/s41597-023-01949-y.
PMID: 36720882; PMCID: PMC9887579.

Appendix Figure 1. An example of the consistency of the biomarker quantification in the control samples for leucine. Different colors show results for four control samples that are measured interleaved in NMR instruments during the project course. Dashed and full-blown lines indicate results from two different NMR instruments.

Appendix Figure 2. Technical and biological repeatability for glycoprotein acetyls (GlycA). The technical consistency over time is demonstrated through the distributions of consecutive batches of sample shipments (first panel) and different spectrometers (second panel). The technical repeatability is also displayed in the form of approximately 650 blind duplicate samples' consistency (third panel), resulting in a between-instrument CV of 3%. The biological repeatability for measurements from blood samples collected from the same individuals around 4 years apart is exhibited in the fourth panel.

Comment #3. The authors reported that metabolic metrics including 168 metabolites in absolute levels and 81 as ratio values were identified in plasma sample, which is probably a little bit difficult using 500 MHz NMR spectrometer. Several macromolecules such as lipids, proteins or lipoproteins in plasma will overlay small molecular metabolites and make their quantifications impossible. The authors must show the characteristic NMR spectrum of plasma and provide the detailed information of metabolite assignments.

Response: Thank you for bringing up this important issue. To clarify, the Nightingale Health's 500 MHz NMR metabolomics platform utilizes two distinct types of NMR spectra to detect both macromolecules and low-molecular-weight metabolites.¹ In addition, it is feasible to determine over 200 metabolic indicators using Nightingale Health's 500 MHz NMR metabolomics platform.²⁻⁹ In fact, The Nightingale NMR platform has received various regulatory approvals, including CE-mark, and 37 biomarkers in the panel have been certified for diagnostics use, which include most of the low-molecular-weight metabolites being quantified.¹ To further validate the reliability of the platform's metabolite quantification, analyses were conducted to compare quantification results of the metabolites with clinical chemistry and mass spectrometry.

The two spectra enable the detection of both macromolecules and low-molecular-weight metabolites. The first spectrum was a presaturated proton spectrum that mainly featured resonances arising from proteins and lipids within various lipoprotein particles. In contrast, the other spectrum was a Carr-Purcell-Meiboom-Gill T2-relaxation-filtered spectrum designed to suppress most of the broad macromolecule and lipoprotein lipid signals, leading to enhanced detection of low-molecular-weight metabolites. By using these two different types of spectra, Nightingale Health's 500 MHz NMR platform can detect both large and small molecule metabolites and provide comprehensive biomarker information. **Appendix Figure 3** shows an example of key signal assignments obtained using 500 MHz NMR spectrometer (Bruker AVANCE III), as provided below for your convenience. The first

window is dominated by broad signals arising from macromolecules, mainly lipoprotein lipids and albumin. Despite the broad overall characteristics and heavy overlap of the resonances, appropriate data analyses provided abundant information on lipoprotein particles as indicated by the inset illustrating the lipoprotein subclasses.¹⁰ In the low-molecular-weight metabolite window, data are acquired with such spectrometer settings (using a T2-relaxation-filtered pulse sequence) that suppress most of the broad macromolecule and lipoprotein lipid signals and in that way enhance the detection of small molecular metabolites.¹⁰ However, the exact NMR spectral data in the UKB cohort are not available as they are outside of the scope of the Nightingale-UKB initiative (<https://www.ukbiobank.ac.uk>)¹.

In addition, it is feasible to determine over 200 metabolic indicators using Nightingale Health's 500 MHz NMR metabolomics platform.²⁻⁹ In fact, 500 MHz NMR spectroscopy is a highly sensitive metabolomics assay that can detect the signal of many metabolites in plasma and provides precise quantification and structural identification of these metabolites. Numerous published studies have used this 500 MHz NMR metabolomics platform to identify over 200 metabolic markers, including amino acids, glycolysis-related metabolites, fatty acids and detailed lipoprotein lipid profiles, covering triacylglycerol, total cholesterol, non-esterified cholesterol, esterified cholesterol and phospholipids within various subclasses, as in the current study.²⁻⁹

The platform has received various regulatory approvals, including the CE-mark, and 37 biomarkers in the panel have been certified for diagnostic use, which include most of the low-molecular-weight metabolites being quantified.¹ This confirms the feasibility of detect both large and small molecule metabolites at the same time and further confirms the reliability of the metabolite quantification. Biomarkers that have been certified for diagnostics use include various amino acids, covering alanine, glycine, histidine, phenylalanine, tyrosine, isoleucine, leucine, valine, total branched-chain amino acids; glycolysis-related metabolites, covering glucose and

lactate; inflammation-related metabolite (glycoprotein acetyls [GlycA]); fluid balance-related metabolite, covering creatinine and albumin; fatty acids, covering total fatty acids, omega-3 fatty acids, omega-6 fatty acids, polyunsaturated fatty acids (PUFA), monounsaturated fatty acids (MUFA), saturated fatty acids, docosahexaenoic acid (DHA); fatty acid ratios, covering omega-3 fatty acids ratio to total fatty acids, omega-6 fatty acids ratio to total fatty acids, PUFA ratio to total fatty acids, MUFA ratio to total fatty acids, saturated fatty acids ratio to total fatty acids, DHA ratio to total fatty acids, PUFA to MUFA ratio, omega-6 fatty acids to omega-3 fatty acids ratio; cholesterol, covering total cholesterol, VLDL cholesterol, clinical LDL cholesterol, HDL cholesterol; triglycerides (total triglycerides); apolipoproteins, covering apolipoprotein B, apolipoprotein A1, apolipoprotein B to apolipoprotein A1 ratio.

To further validate the reliability of the Nightingale NMR platform metabolite quantification, the quantification results of the metabolites were also compared with those quantified by clinical chemistry and mass spectrometry. It should be noted that when comparing the clinical biochemistry biomarkers in the UKB, these biomarkers were primarily measured from serum samples, specifically from aliquot 1. In contrast, NMR-based biomarkers were measured from EDTA plasma samples from aliquot 3, which may have been affected by different degrees of dilution.¹⁰ **Appendix Figure 4** displays scatterplots and correlation coefficients between lipids, apolipoproteins, creatinine, albumin, and glucose that were measured both by clinical chemistry assays and NMR in the UKB cohort. The corresponding plots that demonstrate biomarker consistency in the FinHealth 2017 study, a population-based cohort under the Finnish Institute for Health and Welfare (THL) Biobank comprising approximately 6,000 participants¹¹, are also presented. In that cohort, clinical chemistry assays were measured in a central laboratory from frozen serum samples shortly after the cohort survey. NMR biomarkers were measured 1-2 years later from frozen serum samples taken during 2018-2019, and aliquots of 350 μ L serum were used for the NMR measurements.

The correlation coefficients in both cohorts were high, but the overall consistency was weaker in the UK Biobank when compared to the FinHealth 2017 cohort¹¹. Additionally, there were more pronounced offset and slope deviations from the diagonal in the UK Biobank dataset. This is likely due to the use of different aliquots for the measurements and the differences between serum and EDTA plasma, as well as variations in sample storage time. Notably, the correlation coefficient for albumin was weak in the UK Biobank dataset. However, it is worth highlighting that the associations with disease outcomes were broadly similar for albumin for both assays.

The consistency between NMR biomarker measurements and clinical biochemistry is also consistent with earlier studies that have found correlation coefficients above 0.9.¹² A recent study also reported similar correlations for these NMR biomarkers with clinical chemistry for FINRISK cohorts under the THL Biobank.¹³ Specifically, correlations were approximately $r \sim 0.95$ for the most recent sample collection (FINRISK 2012) and $r \sim 0.90$ for the oldest sample collections (FINRISK 1997).

According to Würtz et al.¹², the biomarker coverage provided by Nightingale Health's NMR platform is largely unique from those obtained through mass-spectrometry based assays. Only a small number of the 249 biomarkers are quantified by mass-spectrometry metabolomics vendors, predominantly in the cases of amino acids and glycolysis metabolites. This is due to the inability of mass spectrometry platforms to accurately quantify detailed lipoprotein measures captured by the Nightingale Health NMR platform, as the physiological nature of lipoprotein particles are lost during mass spectrometry. Moreover, essential analytes, such as the GlycA composite-protein biomarker and aggregate fatty acid measures (e.g., omega-3%), which are relevant for dietary studies and supplementation trials and are often more readily understood than molecule-specific fatty acids, are not commonly analyzed by mass spectrometry.

Appendix Figure 5 displays scatter plots of absolute and relative fatty acids measured

using Nightingale Health NMR platform compared to gas chromatography (Vitas Analytical Services, Oslo, Norway) for a cohort of N=144 individuals with familial hypercholesterolemia.¹⁴ The correlation was especially high for absolute fatty acid measures ($r=0.89-0.98$) and slightly lower for fatty acid ratios relative to the total fatty acids ($r=0.80-0.92$). These results align with earlier comparisons of NMR versus gas chromatography fatty acids using a previous version of the Nightingale Health biomarker platform that entailed lipid extraction.¹²

Appendix Figure 6 displays scatter plots of amino acids compared to the Biocrates p180 mass spectrometry platform (Innsbruck, Austria) for the ADNI1 cohort (N=749). The highest correlations were observed for branched-chain and aromatic amino acids as well as alanine and glycine ($r=0.78-0.90$), with lower correlations observed for glutamine ($r=0.65$) and histidine ($r=0.54$). Correlation coefficients (R) represent linear Pearson's correlations. The regression line and the corresponding equation represent the slope and offset from ordinary least squares linear regression fit.

Appendix Figure 6 also shows the scatter plot for the ketone body 3-hydroxybutyrate compared to measurements taken using cyclic enzymatic method (Wako Chemicals GmbH, Neuss, Germany) for an Italian cohort¹⁵, with the correlation between NMR-based measure and cyclic enzymatic method found to be $r=0.98$. From the same study, triglyceride-rich lipoprotein cholesterol measured using Nightingale Health's NMR platform was previously found to correlate well with ultra-centrifugation ($r=0.90$).¹⁶

Lastly, Pearson's correlations of amino acids and glycolysis-related metabolites measured with Nightingale Health's platform and the Metabolon HD4 mass-spectrometry platform (Morrisville, North Carolina, US) from the same samples were reported as follows in the Qatar Metabolomics Study on Diabetes cohort (QMDiab): leucine $r=0.86$; valine $r=0.82$; phenylalanine $r=0.67$; tyrosine $r=0.90$; glutamine $r=0.75$; histidine $r=0.62$; alanine $r=0.75$; glucose $r=0.86$; lactate $r=0.93$;

citrate $r=0.81$ (results courtesy of Karsten Suhre, Weill Cornell Medicine Qatar). These results align with other studies reporting moderate to high consistency ($r=0.42-0.85$) between the few overlapping biomarkers measured by Nightingale Health's NMR and mass-spectrometry data from Metabolon and Biocrates.^{17,18}

Based on the presented evidence, we assert that the quantification of metabolic biomarkers using Nightingale Health's 500 MHz NMR metabolomics platform in the current study is reliable. We hope this addresses your concerns.

Reference

1. Julkunen H, Cichońska A, Tiainen M, Koskela H, Nybo K, Mäkelä V, Nokso-Koivisto J, Kristiansson K, Perola M, Salomaa V, Jousilahti P, Lundqvist A, Kangas AJ, Soininen P, Barrett JC, Würtz P. Atlas of plasma NMR biomarkers for health and disease in 118,461 individuals from the UK Biobank. *Nat Commun.* 2023 Feb 3;14(1):604. doi: 10.1038/s41467-023-36231-7. PMID: 36737450; PMCID: PMC9898515.
2. Ahola-Olli AV, Mustelin L, Kalimeri M, Kettunen J, Jokelainen J, Auvinen J, Puukka K, Havulinna AS, Lehtimäki T, Kähönen M, Juonala M, Keinänen-Kiukaanniemi S, Salomaa V, Perola M, Järvelin MR, Ala-Korpela M, Raitakari O, Würtz P. Circulating metabolites and the risk of type 2 diabetes: a prospective study of 11,896 young adults from four Finnish cohorts. *Diabetologia.* 2019 Dec;62(12):2298-2309. doi: 10.1007/s00125-019-05001-w. Epub 2019 Oct 4. PMID: 31584131; PMCID: PMC6861432.
3. Tikkanen E, Jägerroos V, Holmes MV, Sattar N, Ala-Korpela M, Jousilahti P, Lundqvist A, Perola M, Salomaa V, Würtz P. Metabolic Biomarker Discovery for Risk of Peripheral Artery Disease Compared with Coronary Artery Disease: Lipoprotein and Metabolite Profiling of 31 657 Individuals From 5 Prospective Cohorts. *J Am Heart Assoc.* 2021 Dec 7;10(23): e021995. doi: 10.1161/JAHA.121.021995. Epub 2021 Nov 30. PMID: 34845932; PMCID: PMC9075369.

4. Xu Y, Ritchie SC, Liang Y, Timmers PRHJ, Pietzner M, Lannelongue L, Lambert SA, Tahir UA, May-Wilson S, Foguet C, Johansson Å, Surendran P, Nath AP, Persyn E, Peters JE, Oliver-Williams C, Deng S, Prins B, Luan J, Bombá L, Soranzo N, Di Angelantonio E, Pirastu N, Tai ES, van Dam RM, Parkinson H, Davenport EE, Paul DS, Yau C, Gerszten RE, Mälärstig A, Danesh J, Sim X, Langenberg C, Wilson JF, Butterworth AS, Inouye M. An atlas of genetic scores to predict multi-omic traits. *Nature*. 2023 Apr;616(7955):123-131. doi: 10.1038/s41586-023-05844-9. Epub 2023 Mar 29. PMID: 36991119.
5. Buergel T, Steinfeldt J, Ruyoga G, Pietzner M, Bizzarri D, Vojinovic D, Upmeier Zu Belzen J, Look L, Kittner P, Christmann L, Hollmann N, Strangalies H, Braunger JM, Wild B, Chiesa ST, Spranger J, Klostermann F, van den Akker EB, Trompet S, Mooijaart SP, Sattar N, Jukema JW, Lavrijssen B, Kavousi M, Ghanbari M, Ikram MA, Slagboom E, Kivimäki M, Langenberg C, Deanfield J, Eils R, Landmesser U. Metabolomic profiles predict individual multidisease outcomes. *Nat Med*. 2022 Nov;28(11):2309-2320. doi: 10.1038/s41591-022-01980-3. Epub 2022 Sep 22. PMID: 36138150; PMCID: PMC9671812.
6. Bell JA, Richardson TG, Wang Q, Sanderson E, Palmer T, Walker V, O'Keefe LM, Timpson NJ, Cichonska A, Julkunen H, Würtz P, Holmes MV, Davey Smith G. Effects of general and central adiposity on circulating lipoprotein, lipid, and metabolite levels in UK Biobank: A multivariable Mendelian randomization study. *Lancet Reg Health Eur*. 2022 Jul 6; 21:100457. doi: 10.1016/j.lanepe.2022.100457. PMID: 35832062; PMCID: PMC9272390.
7. Smith CJ, Sinnott-Armstrong N, Cichońska A, Julkunen H, Fauman EB, Würtz P, Pritchard JK. Integrative analysis of metabolite GWAS illuminates the molecular basis of pleiotropy and genetic correlation. *Elife*. 2022 Sep 8;11: e79348. doi: 10.7554/eLife.79348. PMID: 36073519; PMCID: PMC9536840.
8. Julkunen H, Cichońska A, Slagboom PE, Würtz P; Nightingale Health UK Biobank Initiative. Metabolic biomarker profiling for identification of susceptibility to severe pneumonia and COVID-19 in the general population. *Elife*. 2021 May 4;10: e63033. doi: 10.7554/eLife.63033. PMID: 33942721; PMCID: PMC8172246.

9. Zhang X, Hu W, Wang Y, Wang W, Liao H, Zhang X, Kiburg KV, Shang X, Bulloch G, Huang Y, Zhang X, Tang S, Hu Y, Yu H, Yang X, He M, Zhu Z. Plasma metabolomic profiles of dementia: a prospective study of 110,655 participants in the UK Biobank. *BMC Med.* 2022 Aug 15;20(1):252. doi: 10.1186/s12916-022-02449-3. PMID: 35965319; PMCID: PMC9377110.
10. Soininen P, Kangas AJ, Würtz P, Tukiainen T, Tynkkynen T, Laatikainen R, Järvelin MR, Kähönen M, Lehtimäki T, Viikari J, Raitakari OT, Savolainen MJ, Ala-Korpela M. High-throughput serum NMR metabonomics for cost-effective holistic studies on systemic metabolism. *Analyst.* 2009 Sep;134(9):1781-5. doi: 10.1039/b910205a. Epub 2009 Jul 30. PMID: 19684899.
11. Borodulin K, Tolonen H, Jousilahti P, Jula A, Juolevi A, Koskinen S, Kuulasmaa K, Laatikainen T, Männistö S, Peltonen M, Perola M, Puska P, Salomaa V, Sundvall J, Virtanen SM, Vartiainen E. Cohort Profile: The National FINRISK Study. *Int J Epidemiol.* 2018 Jun 1;47(3):696-696i. doi: 10.1093/ije/dyx239. PMID: 29165699.
12. Würtz P, Kangas AJ, Soininen P, Lawlor DA, Davey Smith G, Ala-Korpela M. Quantitative Serum Nuclear Magnetic Resonance Metabolomics in Large-Scale Epidemiology: A Primer on -Omic Technologies. *Am J Epidemiol.* 2017 Nov 1;186(9):1084-1096. doi: 10.1093/aje/kwx016. PMID: 29106475; PMCID: PMC5860146.
13. Tikkanen E, Jägerroos V, Holmes MV, Sattar N, Ala-Korpela M, Jousilahti P, Lundqvist A, Perola M, Salomaa V, Würtz P. Metabolic Biomarker Discovery for Risk of Peripheral Artery Disease Compared with Coronary Artery Disease: Lipoprotein and Metabolite Profiling of 31 657 Individuals From 5 Prospective Cohorts. *J Am Heart Assoc.* 2021 Dec 7;10(23): e021995. doi: 10.1161/JAHA.121.021995. Epub 2021 Nov 30. PMID: 34845932; PMCID: PMC9075369.
14. Øyri LKL, Hansson P, Bogsrud MP, Narverud I, Florholmen G, Leder L, Byfuglien MG, Veierød MB, Ulven SM, Holven KB. Delayed postprandial TAG peak after intake of SFA compared with PUFA in subjects with and without familial

hypercholesterolaemia: a randomised controlled trial. *Br J Nutr.* 2018 May;119(10):1142-1150. doi: 10.1017/S0007114518000673. Erratum in: *Br J Nutr.* 2018 Sep;120(5):597. PMID: 29759104.

15. Tikkanen E, Minicocci I, Hällfors J, Di Costanzo A, D'Erasmus L, Poggiogalle E, Donini LM, Würtz P, Jauhiainen M, Olkkonen VM, Arca M. Metabolomic Signature of Angiopoietin-Like Protein 3 Deficiency in Fasting and Postprandial State. *Arterioscler Thromb Vasc Biol.* 2019 Apr;39(4):665-674. doi: 10.1161/ATVBAHA.118.312021. PMID: 30816800.
16. Würtz P, Soininen P. Reply to: "Methodological issues regarding: "A third of nonfasting plasma cholesterol is in remnant lipoproteins: Lipoprotein subclass profiling in 9293 individuals"". *Atherosclerosis.* 2020 Jun; 302:59-61. doi: 10.1016/j.atherosclerosis.2020.03.028. Epub 2020 Apr 20. PMID: 32359769.
17. Deelen J, Kettunen J, Fischer K, van der Spek A, Trompet S, Kastenmüller G, Boyd A, Zierer J, van den Akker EB, Ala-Korpela M, Amin N, Demirkan A, Ghanbari M, van Heemst D, Ikram MA, van Klinken JB, Mooijaart SP, Peters A, Salomaa V, Sattar N, Spector TD, Tiemeier H, Verhoeven A, Waldenberger M, Würtz P, Davey Smith G, Metspalu A, Perola M, Menni C, Geleijnse JM, Drenos F, Beekman M, Jukema JW, van Duijn CM, Slagboom PE. A metabolic profile of all-cause mortality risk identified in an observational study of 44,168 individuals. *Nat Commun.* 2019 Aug 20;10(1):3346. doi: 10.1038/s41467-019-11311-9. PMID: 31431621; PMCID: PMC6702196.
18. Schmidt JA, Fensom GK, Rinaldi S, Scalbert A, Gunter MJ, Holmes MV, Key TJ, Travis RC. NMR Metabolite Profiles in Male Meat-Eaters, Fish-Eaters, Vegetarians and Vegans, and Comparison with MS Metabolite Profiles. *Metabolites.* 2021 Feb 20;11(2):121. doi: 10.3390/metabo11020121. PMID: 33672542; PMCID: PMC7923783.

Appendix Figure 3. An example of key signal assignments of metabolic biomarkers of the two molecular windows obtained using 500 MHz NMR spectrometer.

Appendix Figure 4. Scatter plots of lipids and other routine biomarkers quantified with NMR platform and clinical chemistry analyzers for the UK Biobank and the FinHealth 2017 cohort.

Appendix Figure 5. Scatter plots of fatty acids quantified with NMR and gas chromatography.

Appendix Figure 6. Scatter plots of amino acids quantified with Nightingale Health NMR platform in comparison to Biocrates p180 mass spectrometry in the ADNI-1 cohort.

Comment #4. Lipoproteins-associated metabolites were found to be more related to RPET, but what is the causal relationship between them? Additional in vivo and in vitro studies need to explore the potential mechanisms.

Response: Thank you for bringing up this important issue. While our primary results focus on the use of altered RPET measurements as biomarkers for individualized stratification of T2DM, which are largely associated with lipoproteins as you kindly pointed out, our study does not intend to emphasize the importance of these specific lipid metabolites in the underlying mechanisms of RPET and diabetes association. We would also like to note that NMR metabolomics alone may not fully or predominantly elucidate the underpinnings of the RPET-diabetes association. Instead, we suggest that the combination of other omics approaches such as MS-based metabolomics, proteomics, and transcriptomics would provide a more comprehensive understanding of the biological landscape of RPET and be more effective in unraveling the underlying mechanisms of their association.

According to your suggestions, we have conducted additional experiment using LC/MS assay¹ in an ethnically diverse cohort² to further investigate the value of RPET metabolic fingerprints for stratifying diabetic adverse microvascular phenotypes (**Methods** section, **Page 17, Line 477**; **Supplementary Methods: Experimental procedures, and quality control processes of metabolomic profiling in the GDES cohort**). With LC/MS assay, a variety of additional low-molecular-weight metabolites were also identified to be RPE metabolic fingerprints, encompassing amino acids, FAs, benzene, nucleotides, organic acids, heterocyclic compounds, and their derivatives (**Supplementary Table S12**, as provided below for your convenience). We have also performed additional pathway analysis for understanding the biological implications of these metabolite changes (**Supplementary Figure S5**, as provided below). The incorporation of these metabolites into the clinical indicators-based model resulted in further improvements in discriminative power and clinical utility, further confirming the role of RPET metabolic fingerprints in facilitating risk stratification for T2DM-related damages (**Supplementary Figure S3**, as provided below for your

convenience). NMR and LC/MS assays in this study provided complementary coverage of metabolic markers, exhibiting a more comprehensive RPET metabolic landscape. Our study here establishes a framework that suggests any biological changes that trigger or coincide with changes of RPET measurements could potentially be involved in the early pathogenesis of T2DM, surpassing the metabolic changes captured by NMR metabolomics, since these changes are detectable up to a decade before the onset of T2DM. Future studies aimed at investigating the in-depth mechanisms of these RPET-associated biological changes are poised to improve our system-level understanding of the underlying link between RPE and diabetes risks.

Regarding the biological underpinnings linking RPET and diabetes, we have expanded the **Discussion** section and put forth some speculations in the revised manuscript (**Page 9, Line 236**). Alterations in the retina have long been considered a window to systemic health,³⁻⁵ and our results indicated that changes in the RPE may suggest a new direction to unravel diabetes even further. Alterations in lipid metabolism are deemed at the core of diabetes phenotypes,⁶⁻⁷ and studies have indicated that insulin resistance can stimulate triglyceride synthesis and VLDL production, leading to an excessive accumulation of VLDL and LDL.⁸⁻¹⁰ Studies on NMR metabolomic profiling have also linked VLDL and LDL particle sizes and concentrations to an increased risk of T2DM.¹¹⁻¹⁴ These findings are in keeping with the adverse RPET metabolic state contributing to the increased T2DM risk observed in this study. Moreover, studies on retina have established that DR and its severity are associated with multiple VLDL and LDL particles, and lipid-lowering therapy has been shown to have significant benefits in preventing and mitigating DR.^{15,16} Modified lipid can have toxic effects on retinal cells, including RPE cells, and their excessive accumulation may result in oxidative stress and endoplasmic reticulum stress,¹⁷ potentially leading to dysfunction and structural damages of the RPE. Consistently, *in vitro* studies have demonstrated that RPE cells cultured under diabetic-like conditions produce elevated levels of reactive oxygen species compared to those incubated under non-diabetic conditions.¹⁸ We speculate that a specific metabolic state that prioritizes damage to

the retina before evident T2DM has already taken place, and while these subtle metabolic alterations may not have a noticeable effect on the body yet, they are already detectable through in vivo OCT scanning and metabolomic profiling.

Thank you again for all your valuable feedback and constructive suggestions. We appreciate the time and dedication you have taken to help us improve the quality of this manuscript.

Reference

1. Chen W, Gong L, Guo Z, Wang W, Zhang H, Liu X, Yu S, Xiong L, Luo J. A novel integrated method for large-scale detection, identification, and quantification of widely targeted metabolites: application in the study of rice metabolomics. *Mol Plant*. 2013 Nov;6(6):1769-80. doi: 10.1093/mp/sst080. Epub 2013 May 23. PMID: 23702596.
2. Zhang S, Chen Y, Wang L, Li Y, Tang X, Liang X, He M, Wenyong H, Wang W; GDES group. Design and Baseline Data of the Diabetes Registration Study: Guangzhou Diabetic Eye Study. *Curr Eye Res*. 2023 Feb 27;1-9. doi: 10.1080/02713683.2023.2182745. Epub ahead of print. PMID: 36803011.
3. Cheung CY, Ikram MK, Chen C, Wong TY. Imaging retina to study dementia and stroke. *Prog Retin Eye Res*. 2017 Mar; 57:89-107. doi: 10.1016/j.preteyeres.2017.01.001. Epub 2017 Jan 3. PMID: 28057562.
4. London A, Benhar I, Schwartz M. The retina as a window to the brain-from eye research to CNS disorders. *Nat Rev Neurol*. 2013 Jan;9(1):44-53. doi: 10.1038/nrneurol.2012.227. Epub 2012 Nov 20. PMID: 23165340.
5. Flammer J, Konieczka K, Bruno RM, Viridis A, Flammer AJ, Taddei S. The eye and the heart. *Eur Heart J*. 2013 May;34(17):1270-8. doi: 10.1093/eurheartj/ehs023. Epub 2013 Feb 10. PMID: 23401492; PMCID: PMC3640200.
6. Kane JP, Pullinger CR, Goldfine ID, Malloy MJ. Dyslipidemia and diabetes mellitus: Role of lipoprotein species and interrelated pathways of lipid metabolism in diabetes mellitus. *Curr Opin Pharmacol*. 2021 Dec;61:21-27. doi:

- 10.1016/j.coph.2021.08.013. Epub 2021 Sep 22. PMID: 34562838.
7. Eid S, Sas KM, Abcouwer SF, Feldman EL, Gardner TW, Pennathur S, Fort PE. New insights into the mechanisms of diabetic complications: role of lipids and lipid metabolism. *Diabetologia*. 2019 Sep;62(9):1539-1549. doi: 10.1007/s00125-019-4959-1. Epub 2019 Jul 25. PMID: 31346658; PMCID: PMC6679814.
 8. Vergès B. Abnormal hepatic apolipoprotein B metabolism in type 2 diabetes. *Atherosclerosis*. 2010 Aug;211(2):353-60. doi: 10.1016/j.atherosclerosis.2010.01.028. Epub 2010 Jan 29. PMID: 20189175.
 9. Annuzzi G, De Natale C, Iovine C, Patti L, Di Marino L, Coppola S, Del Prato S, Riccardi G, Rivellese AA. Insulin resistance is independently associated with postprandial alterations of triglyceride-rich lipoproteins in type 2 diabetes mellitus. *Arterioscler Thromb Vasc Biol*. 2004 Dec;24(12):2397-402. doi: 10.1161/01.ATV.0000146267.71816.30. Epub 2004 Sep 30. PMID: 15458975.
 10. Malmström R, Packard CJ, Caslake M, Bedford D, Stewart P, Yki-Järvinen H, Shepherd J, Taskinen MR. Defective regulation of triglyceride metabolism by insulin in the liver in NIDDM. *Diabetologia*. 1997 Apr;40(4):454-62. doi: 10.1007/s001250050700. PMID: 9112023.
 11. Festa A, Williams K, Hanley AJ, Otvos JD, Goff DC, Wagenknecht LE, Haffner SM. Nuclear magnetic resonance lipoprotein abnormalities in prediabetic subjects in the Insulin Resistance Atherosclerosis Study. *Circulation*. 2005 Jun 28;111(25):3465-72. doi: 10.1161/CIRCULATIONAHA.104.512079. PMID: 15983261.
 12. Mora S, Otvos JD, Rosenson RS, Pradhan A, Buring JE, Ridker PM. Lipoprotein particle size and concentration by nuclear magnetic resonance and incident type 2 diabetes in women. *Diabetes*. 2010 May;59(5):1153-60. doi: 10.2337/db09-1114. Epub 2010 Feb 25. PMID: 20185808; PMCID: PMC2857895.
 13. Fizekova M, Miilunpohja M, Kangas AJ, Sojinen P, Kuusisto J, Ala-Korpela M, Laakso M, Stančáková A. Associations of multiple lipoprotein and apolipoprotein measures with worsening of glycemia and incident type 2 diabetes in 6607

- non-diabetic Finnish men. *Atherosclerosis*. 2015 May;240(1):272-7. doi: 10.1016/j.atherosclerosis.2015.03.034. Epub 2015 Mar 23. PMID: 25818853.
14. Ahola-Olli AV, Mustelin L, Kalimeri M, Kettunen J, Jokelainen J, Auvinen J, Puukka K, Havulinna AS, Lehtimäki T, Kähönen M, Juonala M, Keinänen-Kiukaanniemi S, Salomaa V, Perola M, Järvelin MR, Ala-Korpela M, Raitakari O, Würtz P. Circulating metabolites and the risk of type 2 diabetes: a prospective study of 11,896 young adults from four Finnish cohorts. *Diabetologia*. 2019 Dec;62(12):2298-2309. doi: 10.1007/s00125-019-05001-w. Epub 2019 Oct 4. PMID: 31584131; PMCID: PMC6861432.
 15. Lyons TJ, Jenkins AJ, Zheng D, Lackland DT, McGee D, Garvey WT, Klein RL. Diabetic retinopathy and serum lipoprotein subclasses in the DCCT/EDIC cohort. *Invest Ophthalmol Vis Sci*. 2004 Mar;45(3):910-8. doi: 10.1167/iovs.02-0648. PMID: 14985310.
 16. ACCORD Study Group; ACCORD Eye Study Group; Chew EY, Ambrosius WT, Davis MD, Danis RP, Gangaputra S, Greven CM, Hubbard L, Esser BA, Lovato JF, Perdue LH, Goff DC Jr, Cushman WC, Ginsberg HN, Elam MB, Genuth S, Gerstein HC, Schubart U, Fine LJ. Effects of medical therapies on retinopathy progression in type 2 diabetes. *N Engl J Med*. 2010 Jul 15;363(3):233-44. doi: 10.1056/NEJMoa1001288. Epub 2010 Jun 29. Erratum in: *N Engl J Med*. 2011 Jan 13;364(2):190. Erratum in: *N Engl J Med*. 2012 Dec 20;367(25):2458. PMID: 20587587; PMCID: PMC4026164.
 17. Du M, Wu M, Fu D, Yang S, Chen J, Wilson K, Lyons TJ. Effects of modified LDL and HDL on retinal pigment epithelial cells: a role in diabetic retinopathy? *Diabetologia*. 2013 Oct;56(10):2318-28. doi: 10.1007/s00125-013-2986-x. Epub 2013 Jul 11. PMID: 23842729; PMCID: PMC4557884.
 18. Tonade D, Kern TS. Photoreceptor cells and RPE contribute to the development of diabetic retinopathy. *Prog Retin Eye Res*. 2021 Jul;83:100919. doi: 10.1016/j.preteyeres.2020.100919. Epub 2020 Nov 12. PMID: 33188897; PMCID: PMC8113320.

Supplementary Table S12

RPET-associated metabolites identified using LC/MS assay in the Guangzhou Diabetic Eye Study (GDES) cohort after adjusting for other factors.

Metabolites *	Group	β	95%CI		P_{raw}	P_{FDR}
γ -Glu-Met	Amino acid and Its metabolites	2.855	2.147	3.563	$<1.0 \times 10^{-8}$	$<1.0 \times 10^{-8}$
L-Arginine	Amino acid and Its metabolites	2.607	1.882	3.332	$<1.0 \times 10^{-8}$	$<1.0 \times 10^{-8}$
Tryptamine	Tryptamines,Cholines,Pigments	2.327	1.590	3.065	$<1.0 \times 10^{-8}$	2.72×10^{-7}
Val-Ala	Amino acid and Its metabolites	2.247	1.525	2.969	$<1.0 \times 10^{-8}$	3.57×10^{-7}
Met-Glu	Amino acid and Its metabolites	2.270	1.535	3.006	$<1.0 \times 10^{-8}$	4.07×10^{-7}
Piperidine	Heterocyclic compounds	2.314	1.532	3.096	1.08×10^{-8}	1.38×10^{-6}
Azelaic Acid	Organic acid and Its derivatives	2.091	1.366	2.816	2.44×10^{-8}	2.38×10^{-6}
Glu-Met	Amino acid and Its metabolites	2.112	1.375	2.850	3.00×10^{-8}	2.67×10^{-6}
N-Propionylglycine	Amino acid and Its metabolites	2.204	1.434	2.974	3.11×10^{-8}	2.72×10^{-6}
2-(Formylamino)Benzoic Acid	Benzene and substituted derivatives	1.989	1.258	2.721	1.39×10^{-7}	1.07×10^{-5}
2'-O-methylcytidine	Nucleotide and Its metabolites	1.965	1.231	2.698	2.11×10^{-7}	1.51×10^{-5}
3-Hydroxy-L-phenylalanine	Amino acid and Its metabolites	1.854	1.120	2.588	9.72×10^{-7}	6.37×10^{-5}
DL-O-tyrosine	Amino acid and Its metabolites	1.777	1.032	2.521	3.63×10^{-6}	2.19×10^{-4}
D-Alloisoleucine	Amino acid and Its metabolites	1.747	0.959	2.534	1.62×10^{-5}	8.48×10^{-4}
5,6-Dimethylbenzimidazole	Heterocyclic compounds	1.654	0.903	2.405	1.85×10^{-5}	9.45×10^{-4}
L-Methionine	Amino acid and Its metabolites	1.653	0.891	2.416	2.47×10^{-5}	1.19×10^{-3}
Indoleacetaldehyde	Heterocyclic compounds	1.574	0.829	2.319	3.95×10^{-5}	1.66×10^{-3}
9,10-Epoxy-18-hydroxyoctadecanoic acid	FA	1.531	0.801	2.261	4.53×10^{-5}	1.81×10^{-3}
Trp-Gly	Amino acid and Its metabolites	1.603	0.837	2.368	4.64×10^{-5}	1.84×10^{-3}
Glycolithocholic acid	Bile acids	1.515	0.782	2.248	5.75×10^{-5}	2.09×10^{-3}
2'-O-methyluridine	Nucleotide and Its metabolites	1.561	0.803	2.320	6.21×10^{-5}	2.18×10^{-3}
Melatonin	Hormones and hormone related compounds	1.534	0.788	2.279	6.23×10^{-5}	2.19×10^{-3}

N-Acetyl-L-Leucine	Amino acid and Its metabolites	1.658	0.849	2.467	6.62×10^{-5}	2.26×10^{-3}
gamma-Glu-Phe	Amino acid and Its metabolites	1.593	0.796	2.390	1.00×10^{-4}	3.10×10^{-3}
12,13-DiHOME	FA	1.453	0.722	2.184	1.09×10^{-4}	3.27×10^{-3}
9,10-DiHOME	FA	1.449	0.718	2.180	1.15×10^{-4}	3.40×10^{-3}
N-acetylorithine	Amino acid and Its metabolites	1.481	0.730	2.231	1.22×10^{-4}	3.54×10^{-3}
Gly-Phe	Amino acid and Its metabolites	1.437	0.699	2.174	1.49×10^{-4}	4.14×10^{-3}
Glu-Tyr	Amino acid and Its metabolites	1.462	0.705	2.220	1.70×10^{-4}	4.58×10^{-3}
Phe-Hyp	Amino acid and Its metabolites	1.410	0.673	2.146	1.94×10^{-4}	5.04×10^{-3}
Phe-Phe	Amino acid and Its metabolites	1.417	0.670	2.165	2.23×10^{-4}	5.55×10^{-3}
Pro-Asp	Amino acid and Its metabolites	1.398	0.658	2.138	2.33×10^{-4}	5.71×10^{-3}
Glu-Leu	Amino acid and Its metabolites	1.455	0.676	2.234	2.73×10^{-4}	6.40×10^{-3}
Phe-Gly	Amino acid and Its metabolites	1.385	0.641	2.129	2.86×10^{-4}	6.60×10^{-3}
L-Lysine	Amino acid and Its metabolites	1.360	0.624	2.096	3.20×10^{-4}	7.16×10^{-3}
L-Serine	Amino acid and Its metabolites	1.383	0.629	2.137	3.53×10^{-4}	7.69×10^{-3}
N-Amidino-L-Aspartate	Amino acid and Its metabolites	1.356	0.609	2.103	4.04×10^{-4}	8.42×10^{-3}
Multifidol	Benzene and substituted derivatives	1.334	0.594	2.073	4.40×10^{-4}	8.90×10^{-3}
Urocanic Acid	Organic acid and Its derivatives	1.338	0.595	2.080	4.46×10^{-4}	8.98×10^{-3}
PI (15:0/2:0)	GP	1.321	0.577	2.064	5.35×10^{-4}	1.01×10^{-2}
Caffeic Acid	Organic acid and Its derivatives	1.303	0.570	2.037	5.37×10^{-4}	1.01×10^{-2}
Hydroxyurea	Others	1.310	0.572	2.049	5.41×10^{-4}	1.01×10^{-2}
5'-Deoxy-5'-(Methylthio) Adenosine	Nucleotide and Its metabolites	1.318	0.556	2.080	7.45×10^{-4}	1.32×10^{-2}
Quinmerac	Benzene and substituted derivatives	-1.460	-2.306	-0.614	7.66×10^{-4}	1.35×10^{-2}
Proline betaine	Amino acid and Its metabolites	1.278	0.528	2.028	8.89×10^{-4}	1.52×10^{-2}
L-Ornithine	Amino acid and Its metabolites	1.256	0.519	1.994	8.96×10^{-4}	1.53×10^{-2}
L-Glutamine	Amino acid and Its metabolites	1.243	0.507	1.980	9.98×10^{-4}	1.66×10^{-2}
N, N-Bis(2-hydroxyethyl) dodecanamide	FA	1.238	0.496	1.979	1.14×10^{-3}	1.84×10^{-2}

Trolox	Benzene and substituted derivatives	1.235	0.491	1.978	1.20×10^{-3}	1.91×10^{-2}
Asp-Leu	Amino acid and Its metabolites	1.220	0.482	1.958	1.27×10^{-3}	2.00×10^{-2}
4-Acetylamino benzoic acid	Organic acid and Its derivatives	1.205	0.469	1.940	1.40×10^{-3}	2.15×10^{-2}
N-(2-hydroxyethyl)-3-pyridinecarboxamide	Heterocyclic compounds	1.307	0.502	2.112	1.53×10^{-3}	2.31×10^{-2}
Creatine phosphate	Nucleotide and Its metabolites	1.194	0.446	1.943	1.86×10^{-3}	2.71×10^{-2}
Phe-Val	Amino acid and Its metabolites	1.197	0.446	1.948	1.87×10^{-3}	2.72×10^{-2}
2-Aminophenol	Benzene and substituted derivatives	1.157	0.421	1.892	2.14×10^{-3}	3.06×10^{-2}
Hippuric Acid	Organic acid and Its derivatives	1.120	0.385	1.855	2.95×10^{-3}	4.01×10^{-2}
L-Asparagine Anhydrous	Amino acid and Its metabolites	1.162	0.394	1.929	3.14×10^{-3}	4.21×10^{-2}
Phenethylamine	Alcohol and amines	1.175	0.384	1.966	3.75×10^{-3}	4.85×10^{-2}

* Adjusted for age, sex duration of diabetes, HbA1c, body-mass index, systolic blood pressure, smoking, drinking, and hyperlipidemia.

Supplementary Figure S5.

Enrichment analysis probing the biological implications of RPET-associated metabolite changes in the GDES cohort.

Supplementary Figure S3

Added discriminative power and clinical utility of RPET metabolic fingerprints for stratifying T2DM adverse microvascular phenotypes in GDES.

REVIEWER COMMENTS

Reviewer #1 (Remarks to the Author):

The authors have provided more details on the methods and they have added new results from the early study as well as results from an additional study.

A more sensitive metabolic methods has confirmed the first results.

The authors have added new citations.

They have made an important effort to respond to all reviewers concerns.

Although the exact mechanisms linking RPE layer thinning on OCT to lipid dysmetabolism is unclear, the results have been obtained from large population and seem reproducible.

I consider the authors have addressed all concerns.

Reviewer #2 (Remarks to the Author):

The authors have addressed my comments and I have no additional comments at current stage.

Reviewer #3 (Remarks to the Author):

My concerns have been correctly answered.

Reviewer #4 (Remarks to the Author):

The authors have made a significant revision and provided more data to support the conclusions. Although the quality of the manuscript has been improved, there are still several suggestions to be considered as follows:

- (1) Figure legends are too simple to understand these figures. Please rewrite them!
- (2) In Figure 3, it seems there was no significant improvement for stratifying T2DM using an integrated method of clinical indicators and RPET metabolic state?
- (3) Please consider to present more meaningful figures or tables in the main text.
- (4) I still ask for the characteristic NMR spectrum of plasma and provide the detailed information of metabolite assignments.

Responds to the reviewer's comments:

Replies to Reviewer #1

Comment #1. The authors have provided more details on the methods and they have added new results from the early study as well as results from an additional study. A more sensitive metabolic methods has confirmed the first results. The authors have added new citations. They have made an important effort to respond to all reviewers concerns. Although the exact mechanisms linking RPE layer thinning on OCT to lipid dysmetabolism is unclear, the results have been obtained from large population and seem reproducible. I consider the authors have addressed all concerns.

Response: We sincerely thank you for reviewing our manuscript and providing valuable feedback. We deeply appreciate the time and effort you devoted to thoroughly examining our work. It brings us great pleasure to confirm that we have successfully addressed all your concerns. Your support and constructive input have played a crucial role in elevating the quality of our manuscript. We express our heartfelt gratitude for your guidance throughout the review process.

Replies to Reviewer #2

Comment #1. The authors have addressed my comments and I have no additional comments at current stage.

Response: We would like to express our sincere gratitude once again for your invaluable feedback and constructive suggestions. We appreciate the time and dedication you have taken to help us improve the quality of this manuscript. Your insights have been instrumental in refining our work, and we are genuinely thankful for your contribution.

Replies to Reviewer #3:

Comment #1. My concerns have been correctly answered.

Response: We sincerely thank you for your invaluable contribution to our manuscript. Your meticulous review and positive feedback have inspired us to refine

our research further. Your thoughtful engagement and meaningful input have greatly enhanced the quality of our work. Your support has been pivotal to our scholarly endeavor, and we are truly grateful for your contribution.

Replies to Reviewer #4:

Comment #1. Figure legends are too simple to understand these figures. Please rewrite them!

Response: We express our gratitude for your valuable suggestions. We have accepted your recommendations and redesigned all figures, along with thoroughly rewriting all figure legends accordingly.

Comment #2. In Figure 3, it seems there was no significant improvement for stratifying T2DM using an integrated method of clinical indicators and RPET metabolic state?

Response: We sincerely appreciate your bringing this matter to our attention and apologize for any confusion arising from our previous description. The purpose of the two curves in **Figure 4c** is to demonstrate the impact of incorporating RPET metabolomic information into traditional clinical indicators. In the **Figure 4c**, the orange curve represents the traditional clinical indicators, while the blue one represents the combination of the traditional clinical index with RPET metabolomic state. Upon closer examination, it becomes evident that the addition of the RPE index to the model of traditional clinical indicators results in a significant improvement in the C-index (from 0.809 to 0.837, $P < 1.0 \times 10^{-8}$). Moreover, the performance of the solely PRET metabolic state is comparable to that of the traditional clinical indicators. To enhance readability and eliminate potential confusion, we have removed the green curve representing solely RPET metabolomic state from **Figure 4c**. We hope that these efforts demonstrate our commitment to addressing your concerns and alleviating any doubts you may have had.

Comment #3. Please consider to present more meaningful figures or tables in the

main text.

Response: We sincerely appreciate your valuable suggestions and wholeheartedly accept them. Accordingly, we have presented new figures and diligently reorganized and rearranged all figures in the main text, ensuring the core findings of this study are presented in a logical sequence and to their fullest extent. We are hopeful that these revisions effectively address your concerns and demonstrate our commitment to satisfying your expectations.

Comment #4. I still ask for the characteristic NMR spectrum of plasma and provide the detailed information of metabolite assignments.

Response: We sincerely appreciate your raising this important issue. Accordingly, we have conducted a meticulous examination and thoughtful deliberation on the matter. In the subsequent paragraphs, we provide a comprehensive and detailed explanation that addresses the relevant considerations and offers answers aimed at effectively resolving your concerns and dispelling any remaining doubts.

We totally agree with you and acknowledge that the characteristic NMR spectrum would be informative. However, it is essential to clarify that the NMR metabolomics data utilized in this manuscript were obtained directly from a publicly available application to the reputable UK Biobank, a widely recognized and esteemed biomedical database (<https://www.ukbiobank.ac.uk/>). The UK Biobank is one of the world's largest cohorts, comprising samples from 500,000 individuals recruited between 2006 and 2010, and linked to diverse electronic health records. This extensive dataset enables unprecedented studies on health conditions, genetics, environment, and lifestyle factors, significantly advancing our understanding of disease etiology, prevention, and treatment (<https://www.ukbiobank.ac.uk/learn-more-about-uk-biobank/our-impact>). Esteemed journals such as *Nature*¹⁻²⁴, *Science*²⁵⁻²⁷, *Cell*²⁸⁻³³, *Lancet*³⁴⁻⁴⁴, and *JAMA*⁴⁵⁻⁵⁷ have published numerous findings resulting from its contributions. Collaborating with Nightingale Health, the Finnish pioneer in blood biomarker technology, the meticulous assessment of metabolomic biomarkers

were conducted in UK Biobank, adhering to a rigorous and transparent process that has been duly validated.⁵⁸ Around 120,000 participants have metabolic biomarker data from EDTA plasma samples, measured with Nightingale Health Ltd's NMR-based platform.⁵⁸ These well-recognized metabolomics data have been featured in numerous distinguished publications, such as *Nature*¹⁶, *Nature Medicine*⁵⁹, *Nature Communications*⁵⁸, *Lancet Regional Health*⁶⁰, *eLife*⁶¹⁻⁶², and *BMC Medicine*⁶³.

As stated by the UK Biobank team, the NMR spectral data are currently not accessible to researchers, as they fall outside the scope of the Nightingale-UK Biobank initiative.⁵⁸

The authors did not participate in the experimentation or quality control processes of NMR metabolomics conducted in the UK Biobank. However, in response to the concerns raised by the reviewers, we have made several efforts to address the questions. Additionally, we have reached out to the manufacturer of the Bruker AVANCE NMR spectrometer, which was utilized for the NMR metabolomics assay in the UK Biobank study. This correspondence has provided us with more extensive information regarding the methodology for metabolite quantification and metabolite assignment on this platform. We hope that these additional details will address the concerns raised.

In response to the concerns raised by the reviewers, we have provided additional information regarding the NMR metabolomics conducted at the UK Biobank. A comprehensive overview of the metabolomics process, including a brief introduction, experimental procedures, and quality control assurance conducted by Nightingale Health, can be found at the following links: <https://www.ukbiobank.ac.uk/learn-more-about-uk-biobank/news/uk-biobank-adds-the-first-tranche-of-data-from-a-study-into-circulating-metabolomic-biomarkers-to-its-biomedical-database> & <https://biobank.ctsu.ox.ac.uk/crystal/label.cgi?id=220>.

For additional useful information, please refer to the following resources:

(1) To gain insight into the distribution of biomarker values per sample batch

processed, refer to: <https://biobank.ctsu.ox.ac.uk/crystal/refer.cgi?id=3001>.

(2) For an examination of the correlation between measures for blinded duplicates used as controls, refer to: <https://biobank.ctsu.ox.ac.uk/crystal/refer.cgi?id=3002>.

(3) To understand the meaning of QC flags and the frequency of their expected occurrence, refer to: <https://biobank.ctsu.ox.ac.uk/crystal/refer.cgi?id=3004>.

(4) For a comprehensive view of the distribution of biomarker values per spectrometer used for measurement, refer to:

<https://biobank.ctsu.ox.ac.uk/crystal/refer.cgi?id=3005>.

Additionally, we have contacted the manufacturer of the Bruker AVANCE NMR spectrometer and verified that all metabolite signals are captured simultaneously in a single scan, eliminating the need for intricate pre-processing techniques such as separation. This feature significantly improves the reproducibility and stability of the obtained results. In terms of addressing the potential overshadowing of small metabolites by large molecules, the following approach is employed:

1. Metabolites with distinct and distinguishable peaks, such as glucose at 5.24 ppm (double peak), and alanine at 1.48 ppm (double peak), allow for direct calculation of the peak areas.
2. For metabolites that partially overlap with other peaks, such as lactic acid at 1.33 ppm (double peak), preset functions are utilized to fit the peaks and separate them from the overlapping signals.
3. Metabolites that completely overlap with other peaks are excluded from the analysis to ensure accurate quantification.

To provide a visual representation of the signal assignments, we have included graphical examples from the Bruker AVANCE in **Appendix 1**.

Specifically, the subsequent paragraphs provide a concise overview of the methodology and algorithm utilized for quantifications.

1. To quantify small molecules, a specialized algorithm based on a simplex approach was developed in-house and implemented in MatLab R2017a

(Mathworks Inc.). The 1D ¹H general NMR profile experiment served as the basis for quantifying low-molecular-weight molecules. To ensure the accuracy and reliability of the signal assignments and quantification results, additional quality control steps were performed using 2D Jres and CPMG spectra. During signal fitting, a Gauss-Lorentz mixed line shape was employed as the principal line shape for each metabolite signal. In the case of multiplets, a sum of Gauss-Lorentz lines was utilized to reconstruct the measured line shape. The concentration ratio of the signal was calculated for each metabolite and served as a constraint in the minimization process. Baseline contributions were approximated by locally optimized first-order polynomials, incorporating offset and slope as fit parameters, which were added to the multiplet line shapes. All signals were used for the quantification of the metabolites unless they were affected by signal overlap or exhibited high multiplicity, resulting in a detection limit that was too high for a particular signal. The algorithm incorporates up to 11 constraints for the concentration calculation such as molecular mass, number of protons, relaxation time, multiplicity, range of signal detection, fit range, chemical shift, line width, coupling constant, percentage of Lorentzian and Gaussian used in the fitting, baseline offset, slope, and curvature. These comprehensive constraints are considered in the algorithm to ensure accurate and robust concentration calculations of the metabolites of interest.

2. For lipoprotein quantification, the spectra underwent normalization to a standardized quantitative scale using Bruker's QuantRef manager within TopSpin software, which is based on the PULCON method.⁶⁴ This normalization process ensures that the spectral intensity is adjusted to reflect proton concentration in millimoles per liter (mmol/L). The data tables were meancentered using the model data mean. The chemical shift was initially calibrated to the methyl signal of trimethylsilylpropanoic acid TSP using Topspin 3.5 and subsequently calibrated to the alanine doublet at 1.48 ppm. These calibrations ensure accurate and consistent chemical shift referencing across the spectra. The LPD method involves integrating the signals corresponding to the –CH₃ and –CH₂–

groups from lipoproteins, which appear in the 1D ¹H general NMR profile spectrum at chemical shifts of 0.8 and 1.25 ppm, respectively. These integrated signals are fitted using lipoprotein and lipoprotein subclass-related parameters through a PLS-2 regression model. To construct the PLS-2 regression model, spectra from independent blood collections in two cohorts of 100 donors were combined. The first cohort comprised healthy volunteers, while the second one cohort with some kind of lipid metabolism impairment. Each sample in these cohorts underwent analysis using ultracentrifugation, and lipoprotein parameters were quantified using this method. The fractions obtained from ultracentrifugation were then subjected to NMR spectroscopy to generate the reference spectrum used in the PLS-2 model. Both the NMR data and centrifugation data were utilized as input for the PLS-2 modeling. The PLS-2 model was constructed using bucketing parameters (such as size, number, and exclusions), as described by Okazaki et al.⁶⁵ The number of components in the model was determined by minimizing the prediction error through a Monte Carlo embedded cross-validation approach as part of the modeling procedure.

3. As for the 81 ratio measurements, the formulas for calculating them are provided in **Appendix 2**.

We extend our deepest gratitude for your invaluable feedback and constructive suggestions. Your unwavering dedication and meticulous attention have greatly enhanced the quality of this manuscript. While we acknowledge the inherent imperfections of clinical study, we persistently pursue excellence, emphasizing the robustness, reliability, and generalizability of our findings. Hopefully our efforts have addressed your concerns. Once again, we sincerely appreciate your invaluable contribution, and we are genuinely grateful for your guidance and expertise.

Appendix 1. Examples of signal assignments of low-molecular-weight molecules.

Appendix 2. Formulae for calculating NMR metabolomics ratios.

Biomarker_name	Unit	Numerator_ Field_ID	Denominator _Field_ID
Ratio of triglycerides to phosphoglycerides	ratio	23407	23434
Ratio of apolipoprotein B to apolipoprotein A1	ratio	23439	23440
Ratio of omega-3 fatty acids to total fatty acids	%	23444	23442
Ratio of omega-6 fatty acids to total fatty acids	%	23445	23442
Ratio of polyunsaturated fatty acids to total fatty acids	%	23446	23442
Ratio of monounsaturated fatty acids to total fatty acids	%	23447	23442
Ratio of saturated fatty acids to total fatty acids	%	23448	23442
Ratio of linoleic acid to total fatty acids	%	23449	23442
Ratio of docosahexaenoic acid to total fatty acids	%	23450	23442
Ratio of polyunsaturated fatty acids to monounsaturated fatty acids	ratio	23446	23447
Ratio of omega-6 fatty acids to omega-3 fatty acids	ratio	23445	23444
Phospholipids to total lipids ratio in chylomicrons and extremely large VLDL	%	23483	23482
Cholesterol to total lipids ratio in chylomicrons and extremely large VLDL	%	23484	23482
Cholesteryl esters to total lipids ratio in chylomicrons and extremely large VLDL	%	23485	23482
Free cholesterol to total lipids ratio in chylomicrons and extremely large VLDL	%	23486	23482
Triglycerides to total lipids ratio in chylomicrons and extremely large VLDL	%	23487	23482
Phospholipids to total lipids ratio in very large VLDL	%	23490	23489
Cholesterol to total lipids ratio in very large VLDL	%	23491	23489
Cholesteryl esters to total lipids ratio in very large VLDL	%	23492	23489
Free cholesterol to total lipids ratio in very large VLDL	%	23493	23489
Triglycerides to total lipids ratio in very large VLDL	%	23494	23489
Phospholipids to total lipids ratio in large VLDL	%	23497	23496
Cholesterol to total lipids ratio in large VLDL	%	23498	23496
Cholesteryl esters to total lipids ratio in large VLDL	%	23499	23496
Free cholesterol to total lipids ratio in large VLDL	%	23500	23496
Triglycerides to total lipids ratio in large VLDL	%	23501	23496
Phospholipids to total lipids ratio in medium VLDL	%	23504	23503
Cholesterol to total lipids ratio in medium VLDL	%	23505	23503
Cholesteryl esters to total lipids ratio in medium VLDL	%	23506	23503
Free cholesterol to total lipids ratio in medium VLDL	%	23507	23503
Triglycerides to total lipids ratio in medium VLDL	%	23508	23503
Phospholipids to total lipids ratio in small VLDL	%	23511	23510
Cholesterol to total lipids ratio in small VLDL	%	23512	23510
Cholesteryl esters to total lipids ratio in small VLDL	%	23513	23510

Free cholesterol to total lipids ratio in small VLDL	%	23514	23510
Triglycerides to total lipids ratio in small VLDL	%	23515	23510
Phospholipids to total lipids ratio in very small VLDL	%	23518	23517
Cholesterol to total lipids ratio in very small VLDL	%	23519	23517
Cholesteryl esters to total lipids ratio in very small VLDL	%	23520	23517
Free cholesterol to total lipids ratio in very small VLDL	%	23521	23517
Triglycerides to total lipids ratio in very small VLDL	%	23522	23517
Phospholipids to total lipids ratio in IDL	%	23525	23524
Cholesterol to total lipids ratio in IDL	%	23526	23524
Cholesteryl esters to total lipids ratio in IDL	%	23527	23524
Free cholesterol to total lipids ratio in IDL	%	23528	23524
Triglycerides to total lipids ratio in IDL	%	23529	23524
Phospholipids to total lipids ratio in large LDL	%	23532	23531
Cholesterol to total lipids ratio in large LDL	%	23533	23531
Cholesteryl esters to total lipids ratio in large LDL	%	23534	23531
Free cholesterol to total lipids ratio in large LDL	%	23535	23531
Triglycerides to total lipids ratio in large LDL	%	23536	23531
Phospholipids to total lipids ratio in medium LDL	%	23539	23538
Cholesterol to total lipids ratio in medium LDL	%	23540	23538
Cholesteryl esters to total lipids ratio in medium LDL	%	23541	23538
Free cholesterol to total lipids ratio in medium LDL	%	23542	23538
Triglycerides to total lipids ratio in medium LDL	%	23543	23538
Phospholipids to total lipids ratio in small LDL	%	23546	23545
Cholesterol to total lipids ratio in small LDL	%	23547	23545
Cholesteryl esters to total lipids ratio in small LDL	%	23548	23545
Free cholesterol to total lipids ratio in small LDL	%	23549	23545
Triglycerides to total lipids ratio in small LDL	%	23550	23545
Phospholipids to total lipids ratio in very large HDL	%	23553	23552
Cholesterol to total lipids ratio in very large HDL	%	23554	23552
Cholesteryl esters to total lipids ratio in very large HDL	%	23555	23552
Free cholesterol to total lipids ratio in very large HDL	%	23556	23552
Triglycerides to total lipids ratio in very large HDL	%	23557	23552
Phospholipids to total lipids ratio in large HDL	%	23560	23559
Cholesterol to total lipids ratio in large HDL	%	23561	23559
Cholesteryl esters to total lipids ratio in large HDL	%	23562	23559
Free cholesterol to total lipids ratio in large HDL	%	23563	23559
Triglycerides to total lipids ratio in large HDL	%	23564	23559
Phospholipids to total lipids ratio in medium HDL	%	23567	23566
Cholesterol to total lipids ratio in medium HDL	%	23568	23566
Cholesteryl esters to total lipids ratio in medium HDL	%	23569	23566
Free cholesterol to total lipids ratio in medium HDL	%	23570	23566
Triglycerides to total lipids ratio in medium HDL	%	23571	23566
Phospholipids to total lipids ratio in small HDL	%	23574	23573

Cholesterol to total lipids ratio in small HDL	%	23575	23573
Cholesteryl esters to total lipids ratio in small HDL	%	23576	23573
Free cholesterol to total lipids ratio in small HDL	%	23577	23573
Triglycerides to total lipids ratio in small HDL	%	23578	23573

Reference:

1. Backman JD, Li AH, Marcketta A, et al. Exome sequencing and analysis of 454,787 UK Biobank participants. *Nature*. 2021;599(7886):628-634. doi: 10.1038/s41586-021-04103-z
2. Bycroft C, Freeman C, Petkova D, et al. The UK Biobank resource with deep phenotyping and genomic data. *Nature*. 2018;562(7726):203-209. doi: 10.1038/s41586-018-0579-z
3. Douaud G, Lee S, Alfaro-Almagro F, et al. SARS-CoV-2 is associated with changes in brain structure in UK Biobank. *Nature*. 2022;604(7907):697-707. doi: 10.1038/s41586-022-04569-5
4. Van Hout CV, Tachmazidou I, Backman JD, et al. Exome sequencing and characterization of 49,960 individuals in the UK Biobank. *Nature*. 2020;586(7831):749-756. doi: 10.1038/s41586-020-2853-0
5. Turro E, Astle WJ, Megy K, et al. Whole-genome sequencing of patients with rare diseases in a national health system. *Nature*. 2020;583(7814):96-102. doi: 10.1038/s41586-020-2434-2
6. Kessler MD, Damask A, O'Keeffe S, et al. Common and rare variant associations with clonal haematopoiesis phenotypes [published correction appears in *Nature*. 2023 Mar;615(7950):E3]. *Nature*. 2022;612(7939):301-309. doi: 10.1038/s41586-022-05448-9
7. Halldorsson BV, Eggertsson HP, Moore KHS, et al. The sequences of 150,119 genomes in the UK Biobank. *Nature*. 2022;607(7920):732-740. doi: 10.1038/s41586-022-04965-x
8. Elliott LT, Sharp K, Alfaro-Almagro F, et al. Genome-wide association studies of brain imaging phenotypes in UK Biobank. *Nature*. 2018;562(7726):210-216. doi: 10.1038/s41586-018-0571-7
9. Wang Q, Dhindsa RS, Carss K, et al. Rare variant contribution to human disease in 281,104 UK Biobank exomes. *Nature*. 2021;597(7877):527-532. doi: 10.1038/s41586-021-03855-y

10. Sun BB, Kurki MI, Foley CN, et al. Genetic associations of protein-coding variants in human disease. *Nature*. 2022;603(7899):95-102. doi: 10.1038/s41586-022-04394-w
11. Thompson DJ, Genovese G, Halvardson J, et al. Genetic predisposition to mosaic Y chromosome loss in blood. *Nature*. 2019;575(7784):652-657. doi: 10.1038/s41586-019-1765-3
12. Ostendorf BN, Patel MA, Bilanovic J, et al. Common human genetic variants of APOE impact murine COVID-19 mortality. *Nature*. 2022;611(7935):346-351. doi: 10.1038/s41586-022-05344-2
13. Evershed RP, Davey Smith G, Roffet-Salque M, et al. Dairying, diseases and the evolution of lactase persistence in Europe [published correction appears in *Nature*. 2022 Sep;609(7927):E9]. *Nature*. 2022;608(7922):336-345. doi: 10.1038/s41586-022-05010-7
14. Okbay A, Beauchamp JP, Fontana MA, et al. Genome-wide association study identifies 74 loci associated with educational attainment. *Nature*. 2016;533(7604):539-542. doi: 10.1038/nature17671
15. Wong WJ, Emdin C, Bick AG, et al. Clonal haematopoiesis and risk of chronic liver disease. *Nature*. 2023;616(7958):747-754. doi: 10.1038/s41586-023-05857-4
16. Xu Y, Ritchie SC, Liang Y, et al. An atlas of genetic scores to predict multi-omic traits. *Nature*. 2023;616(7955):123-131. doi:10.1038/s41586-023-05844-9
17. Whole-genome sequencing of the UK Biobank [published online ahead of print, 2022 Jul 20]. *Nature*. 2022;10.1038/d41586-022-01984-6. doi: 10.1038/d41586-022-01984-6
18. UK Biobank data on 500,000 people paves way to precision medicine. *Nature*. 2018;562(7726):163-164. doi:10.1038/d41586-018-06950-9
19. Meyer HV, Dawes TJW, Serrani M, et al. Genetic and functional insights into the fractal structure of the heart. *Nature*. 2020;584(7822):589-594. doi: 10.1038/s41586-020-2635-8

20. Weiner DJ, Nadig A, Jagadeesh KA, et al. Polygenic architecture of rare coding variation across 394,783 exomes. *Nature*. 2023;614(7948):492-499. doi: 10.1038/s41586-022-05684-z
21. Loh PR, Genovese G, McCarroll SA. Monogenic and polygenic inheritance become instruments for clonal selection. *Nature*. 2020;584(7819):136-141. doi: 10.1038/s41586-020-2430-6
22. Loh PR, Genovese G, Handsaker RE, et al. Insights into clonal haematopoiesis from 8,342 mosaic chromosomal alterations. *Nature*. 2018;559(7714):350-355. doi: 10.1038/s41586-018-0321-x
23. Ding Y, Hou K, Xu Z, et al. Polygenic scoring accuracy varies across the genetic ancestry continuum [published online ahead of print, 2023 May 17]. *Nature*. 2023;10.1038/s41586-023-06079-4. doi: 10.1038/s41586-023-06079-4
24. Saevarsdottir S, Olafsdottir TA, Ivarsdottir EV, et al. FLT3 stop mutation increases FLT3 ligand level and risk of autoimmune thyroid disease. *Nature*. 2020;584(7822):619-623. doi: 10.1038/s41586-020-2436-0
25. Palmer DS, Zhou W, Abbott L, et al. Analysis of genetic dominance in the UK Biobank. *Science*. 2023;379(6639):1341-1348. doi: 10.1126/science.abn8455
26. Ferraro NM, Strober BJ, Einson J, et al. Transcriptomic signatures across human tissues identify functional rare genetic variation. *Science*. 2020;369(6509):eaaz5900. doi: 10.1126/science.aaz5900
27. Mukamel RE, Handsaker RE, Sherman MA, et al. Protein-coding repeat polymorphisms strongly shape diverse human phenotypes. *Science*. 2021;373(6562):1499-1505. doi: 10.1126/science.abg8289
28. Wei TT, Chandy M, Nishiga M, et al. Cannabinoid receptor 1 antagonist genistein attenuates marijuana-induced vascular inflammation [published correction appears in *Cell*. 2022 Jun 23;185(13):2387-2389]. *Cell*. 2022;185(10):1676-1693.e23. doi: 10.1016/j.cell.2022.04.005
29. Vuckovic D, Bao EL, Akbari P, et al. The Polygenic and Monogenic Basis of Blood Traits and Diseases. *Cell*. 2020;182(5):1214-1231.e11. doi: 10.1016/j.cell.2020.08.008

30. Astle WJ, Elding H, Jiang T, et al. The Allelic Landscape of Human Blood Cell Trait Variation and Links to Common Complex Disease. *Cell*. 2016;167(5):1415-1429.e19. doi: 10.1016/j.cell.2016.10.042
31. Khera AV, Chaffin M, Wade KH, et al. Polygenic Prediction of Weight and Obesity Trajectories from Birth to Adulthood. *Cell*. 2019;177(3):587-596.e9. doi: 10.1016/j.cell.2019.03.028
32. Lotta LA, Mokrosiński J, Mendes de Oliveira E, et al. Human Gain-of-Function MC4R Variants Show Signaling Bias and Protect against Obesity. *Cell*. 2019;177(3):597-607.e9. doi: 10.1016/j.cell.2019.03.044
33. Hujoel MLA, Sherman MA, Barton AR, et al. Influences of rare copy-number variation on human complex traits. *Cell*. 2022;185(22):4233-4248.e27. doi:10.1016/j.cell.2022.09.028
34. Collins R. What makes UK Biobank special? *Lancet*. 2012;379(9822):1173-1174. doi:10.1016/S0140-6736(12)60404-8
35. Wood AM, Kaptoge S, Butterworth AS, et al. Risk thresholds for alcohol consumption: combined analysis of individual-participant data for 599 912 current drinkers in 83 prospective studies [published correction appears in *Lancet*. 2018 Jun 2;391(10136):2212]. *Lancet*. 2018;391(10129):1513-1523. doi:10.1016/S0140-6736(18)30134-X
36. Keyes KM, Westreich D. UK Biobank, big data, and the consequences of non-representativeness. *Lancet*. 2019;393(10178):1297. doi:10.1016/S0140-6736(18)33067-8
37. Forrest IS, Petrazzini BO, Duffy Á, et al. Machine learning-based marker for coronary artery disease: derivation and validation in two longitudinal cohorts. *Lancet*. 2023;401(10372):215-225. doi:10.1016/S0140-6736(22)02079-7
38. Swanson JM. The UK Biobank and selection bias. *Lancet*. 2012;380(9837):110. doi:10.1016/S0140-6736(12)61179-9
39. An afternoon at UK Biobank. *Lancet*. 2009;373(9670):1146. doi:10.1016/S0140-6736(09)60664-4

40. Palmer LJ. UK Biobank: bank on it. *Lancet*. 2007;369(9578):1980-1982.
doi:10.1016/S0140-6736(07)60924-6
41. Laurie G. Role of the UK Biobank Ethics and Governance Council. *Lancet*. 2009;374(9702):1676. doi:10.1016/S0140-6736(09)61989-9
42. Thompson SG, Willeit P. UK Biobank comes of age. *Lancet*. 2015;386(9993):509-510. doi:10.1016/S0140-6736(15)60578-5
43. Ganna A, Ingelsson E. 5-year mortality predictors in 498,103 UK Biobank participants: a prospective population-based study. *Lancet*. 2015;386(9993):533-540. doi:10.1016/S0140-6736(15)60175-1
44. Barbour V. UK Biobank: a project in search of a protocol? *Lancet*. 2003;361(9370):1734-1738. doi:10.1016/S0140-6736(03)13377-6
45. Harshfield EL, Pennells L, Schwartz JE, et al. Association Between Depressive Symptoms and Incident Cardiovascular Diseases. *JAMA*. 2020;324(23):2396-2405. doi:10.1001/jama.2020.23068
46. Emerging Risk Factors Collaboration, Di Angelantonio E, Kaptoge S, et al. Association of Cardiometabolic Multimorbidity with Mortality [published correction appears in *JAMA*. 2015 Sep 15;314(11):1179. Leening, Maarten [corrected to Leening, Maarten J G]]. *JAMA*. 2015;314(1):52-60. doi:10.1001/jama.2015.7008
47. Lourida I, Hannon E, Littlejohns TJ, et al. Association of Lifestyle and Genetic Risk With Incidence of Dementia. *JAMA*. 2019;322(5):430-437. doi:10.1001/jama.2019.9879
48. Honigberg MC, Zekavat SM, Aragam K, et al. Association of Premature Natural and Surgical Menopause with Incident Cardiovascular Disease. *JAMA*. 2019;322(24):2411-2421. doi:10.1001/jama.2019.19191
49. Elliott J, Bodinier B, Bond TA, et al. Predictive Accuracy of a Polygenic Risk Score-Enhanced Prediction Model vs a Clinical Risk Score for Coronary Artery Disease. *JAMA*. 2020;323(7):636-645. doi:10.1001/jama.2019.22241

50. Choi SH, Weng LC, Roselli C, et al. Association Between Titin Loss-of-Function Variants and Early-Onset Atrial Fibrillation. *JAMA*. 2018;320(22):2354-2364. doi:10.1001/jama.2018.18179
51. Emdin CA, Khera AV, Natarajan P, et al. Genetic Association of Waist-to-Hip Ratio With Cardiometabolic Traits, Type 2 Diabetes, and Coronary Heart Disease. *JAMA*. 2017;317(6):626-634. doi:10.1001/jama.2016.21042
52. Forrest IS, Chaudhary K, Vy HMT, et al. Population-Based Penetrance of Deleterious Clinical Variants. *JAMA*. 2022;327(4):350-359. doi:10.1001/jama.2021.23686
53. Pirruccello JP, Lin H, Khurshid S, et al. Development of a Prediction Model for Ascending Aortic Diameter Among Asymptomatic Individual. *JAMA*. 2022;328(19):1935-1944. doi:10.1001/jama.2022.19701
54. Berry ASF, Finucane BM, Myers SM, et al. Association of Supernumerary Sex Chromosome Aneuploidies with Venous Thromboembolism. *JAMA*. 2023;329(3):235-243. doi:10.1001/jama.2022.23897
55. Atkins JL, Pilling LC, Masoli JAH, et al. Association of Hemochromatosis HFE p.C282Y Homozygosity with Hepatic Malignancy. *JAMA*. 2020;324(20):2048-2057. doi:10.1001/jama.2020.21566
56. Lotta LA, Wittemans LBL, Zuber V, et al. Association of Genetic Variants Related to Gluteofemoral vs Abdominal Fat Distribution with Type 2 Diabetes, Coronary Disease, and Cardiovascular Risk Factors. *JAMA*. 2018;320(24):2553-2563. doi:10.1001/jama.2018.19329
57. Ference BA, Bhatt DL, Catapano AL, et al. Association of Genetic Variants Related to Combined Exposure to Lower Low-Density Lipoproteins and Lower Systolic Blood Pressure with Lifetime Risk of Cardiovascular Disease. *JAMA*. 2019;322(14):1381-1391. doi:10.1001/jama.2019.14120
58. Julkunen H, Cichońska A, Tiainen M, et al. Atlas of plasma NMR biomarkers for health and disease in 118,461 individuals from the UK Biobank. *Nat Commun*. 2023;14(1):604. Published 2023 Feb 3. doi:10.1038/s41467-023-36231-7

59. Buergel T, Steinfeldt J, Ruyoga G, et al. Metabolomic profiles predict individual multidisease outcomes. *Nat Med.* 2022;28(11):2309-2320. doi:10.1038/s41591-022-01980-3
60. Bell JA, Richardson TG, Wang Q, et al. Effects of general and central adiposity on circulating lipoprotein, lipid, and metabolite levels in UK Biobank: A multivariable Mendelian randomization study. *Lancet Reg Health Eur.* 2022; 21:100457. Published 2022 Jul 6. doi: 10.1016/j.lanep.2022.100457
61. Smith CJ, Sinnott-Armstrong N, Cichońska A, et al. Integrative analysis of metabolite GWAS illuminates the molecular basis of pleiotropy and genetic correlation. *Elife.* 2022;11: e79348. Published 2022 Sep 8. doi:10.7554/eLife.79348
62. Julkunen H, Cichońska A, Slagboom PE, Würtz P; Nightingale Health UK Biobank Initiative. Metabolic biomarker profiling for identification of susceptibility to severe pneumonia and COVID-19 in the general population. *Elife.* 2021;10: e63033. Published 2021 May 4. doi:10.7554/eLife.63033
63. Zhang X, Hu W, Wang Y, et al. Plasma metabolomic profiles of dementia: a prospective study of 110,655 participants in the UK Biobank. *BMC Med.* 2022;20(1):252. Published 2022 Aug 15. doi:10.1186/s12916-022-02449-3
64. Wider G, Dreier L. Measuring protein concentrations by NMR spectroscopy. *J Am Chem Soc.* 2006;128(8):2571-2576. doi:10.1021/ja055336t
65. Okazaki M, Usui S, Ishigami M, et al. Identification of unique lipoprotein subclasses for visceral obesity by component analysis of cholesterol profile in high-performance liquid chromatography. *Arterioscler Thromb Vasc Biol.* 2005;25(3):578-584. doi: 10.1161/01.ATV.0000155017.60171.88

REVIEWERS' COMMENTS

Reviewer #1 (Remarks to the Author):

The authors have made the best of their effort to address the reviewer's questions

Reviewer #4 (Remarks to the Author):

I have carefully read this revised manuscript and the authors have addressed my concerns, so I have no additional comments at current version.

Responds to the reviewer's comments:

Replies to Reviewer #1

Comment #1. The authors have made the best of their effort to address the reviewer's questions.

Response: We sincerely appreciate your thorough review of our manuscript and the valuable feedback you provided. Your efforts have significantly contributed to the improvement of our work. We are pleased to confirm that we have addressed all your concerns. Your support and constructive input have been invaluable to us throughout the review process. Once again, we extend our heartfelt gratitude for your time and expertise in evaluating our submission.

Replies to Reviewer #4:

Comment #1. I have carefully read this revised manuscript and the authors have addressed my concerns, so I have no additional comments at current version.

Response: We would like to extend our heartfelt gratitude for your meticulous review of our revised manuscript. Your constructive feedback and insightful comments have been invaluable in enhancing the quality of our work. We are sincerely appreciative of the time and effort you dedicated to this review process. Your expertise has played a pivotal role in refining our manuscript, and we are genuinely thankful for your invaluable contribution.